# Perception of motion salience shapes the emergence of collective motions

Yandong Xiao [1] ✉, Xiaokang Lei [2], Zhicheng Zheng[3], Yalun Xiang[3], Yang-Yu Liu [4,5] & Xingguang Peng [3] ✉

Despite the profound implications of self-organization in animal groups for collective behaviors, understanding the fundamental principles and applying them to swarm robotics remains incomplete. Here we propose a heuristic measure of perception of motion salience (MS) to quantify relative motion changes of neighbors from first-person view. Leveraging three large bird-flocking datasets, we explore how this perception of MS relates to the structure of leader-follower (LF) relations, and further perform an individual-level correlation analysis between past perception of MS and future change rate of velocity consensus. We observe prevalence of the positive correlations in real flocks, which demonstrates that individuals will accelerate the convergence of velocity with neighbors who have higher MS. This empirical finding motivates us to introduce the concept of adaptive MS-based (AMS) interaction in swarm model. Finally, we implement AMS in a swarm of ~$10^2$ miniature robots. Swarm experiments show the significant advantage of AMS in enhancing self-organization of the swarm for smooth evacuations from confined environments.

The collective motion of organisms, e.g., flocks of starlings, colonies of army ants, schools of barracudas milling, and herds of zebra, is one of the most pervasive and spectacular manifestations of coordinated behavior. Collective motion has been studied for decades from biological, physical, and engineering perspectives[1-8]. The bird flocking is one of the most extensively studied examples of collective motion. For example, the studies on starlings[9-11], pigeons[12], jackdaws[13-15], and chimney swifts[16] proposed the topological[10], metric[16], or plastic[15] interactions in the flocking models to explain different motion patterns of collective behaviors.

While the proposed topological, metric, or plastic interactions have been successful in explaining macro-level behaviors, they are largely phenomenological and primarily rooted in computational theories rather than biological evidence. These local interactions originate from the physics-oriented Vicsek-like model[17], a simplified rule

of averaging-alignment. It is apparent that this description fails to comprehensively encapsulate the underlying principles that govern the collective motions observed in animal flocks within their natural environments. For example, starling flocks display the so-called wave of agitation or shimmering waves to reduce risk of predation[18], and fish slow down to avoid collisions whereas birds fly at low variability of speed and lose altitude during turning[19].

In order to go beyond this averaging-type interaction, a natural thought is to dissect bio-inspired mechanisms from real animal flocks. It is particularly important to initiate the modeling from the perspective of individual first-person perception during this exploration. At the individual level, the macro-phenomena emerging from group-level, such as hierarchical leading relations in pigeon flocks[12], stem from individuals adjusting their movements based on information perceived from the surrounding environment and neighbors'

[1]College of System Engineering, National University of Defense Technology, Changsha, Hunan, China. [2]College of Information and Control Engineering, Xi'an University of Architecture and Technology, Xi'an, Shaanxi, China. [3]School of Marine Science and Technology, Northwestern Polytechnical University, Xi'an, Shaanxi, China. [4]Channing Division of Network Medicine, Department of Medicine, Brigham and Women's Hospital and Harvard Medical School, Boston, MA, USA. [5]Center for Artificial Intelligence and Modeling, The Carl R. Woese Institute for Genomic Biology, University of Illinois at Urbana-Champaign, Champaign, IL, USA. ✉e-mail: xiaoyandong08@gmail.com; pxg@nwpu.edu.cn

movements. Even though this idea is widely accepted that individual first-person perception shapes the self-organization of collective behaviors[20,21], the majority of swarm models bypass this aspect at the initial stages of modeling. Instead, they tend to favor averaging-type interactions and idealized perceptions. As an example, they do not explicitly incorporate the perception of neighbors' motions but rather directly feed neighbors' absolute coordinates and velocities to synthesize reactive actions. It is not biologically plausible and suffers from redundant inputs, effective information dilution[22], and sensory overload[23].

Moreover, transcending the confines of mathematical modeling pertaining to real animal flocks and the validation of their mechanisms through numerical simulations, physical swarm robotics provides a versatile test-bed that amalgamates physical authenticity with convenient analyzability[24]. This facilitates the investigation of group-level organization of collective behaviors that emerges from individual-level attributes like individual perception, decision-making, and kinematics within a tangible real-world setting. As an illustration, a research endeavor integrated the computational and robotic approaches to investigate the interaction strategies among individuals within schooling fish[25]. Nevertheless, the reality remains that existing local interactions have not adequately supported the implementation of bio-inspired mechanisms for the practical utilization of swarm robotics[26,27]. The majority of swarm models integrated into robots commonly employ the position-based and averaging-type interactions to engender the observable phenomena of alignment, attraction, and repulsion[17,28–30]. Nevertheless, practical applications necessitate a higher degree of complexity and a broader range of group adaptability across diverse contexts, such as collective anti-predators[15], collective turn[31], cooperative transportation[32] or excavation[33], collective chase or escape[34–36], and more. Clearly, the progression of swarm robotics with the capacity to tackle various collective tasks is intricately linked to the elucidation of the fundamental mechanisms governing the different collective patterns, dynamics, and functionalities observed in natural animal flocks.

Despite extensive research conducted across the fields of biology, physics, and robotics, a definitive pathway to bridge bio-inspired mechanisms with the applications of swarm robotics has yet to be firmly established. Once this promising route is well-defined, it has the potential not only to validate the effectiveness of self-organization underlying collective behaviors, but also to offer substantial guidance and improvements in practical design and distributed control strategies for swarm robots. In this study, by leveraging three large bird-flocking datasets, we try to integrate a comprehensive research chain, starting with investigating the interactions with neighbors in real bird flocks, translating bio-inspired mechanisms into swarm model with explicit empirical evidence, and eventually applying in swarm robotics to examine the efficacy of self-organization in collective tasks.

Initially, we endeavor to propose a heuristic metric to measure the relative motion changes between a pair of individuals from the focal one's first-person view. This serves as an initial step in elucidating the connection between the perception of neighbors' movements and the inherent leader-follower dynamics, as leadership emerges from the adaptation of one's movements upon perceiving the motions of neighbors. In addition, through research on motion perception and the utilization of three large bird-flocking datasets, our objective is to answer the question of what specific motion characteristic an individual possesses to lead the flock. Subsequently, we aspire to translate these findings into bio-inspired mechanisms supported by explicit empirical evidence. Thus, we model leaders' characteristic motions as an adaptive interaction rule about how the focal individual interacts with its neighbors after perceiving their motions. Finally, we adopt the adaptive interaction rule learned from real flocks for swarm robotics consisting of ~$10^2$ two-wheel differential miniature robots. Extensive swarm experiments focusing on collective evacuation and collective following

provide compelling evidence that bio-inspired mechanisms augment the self-organization of the swarm across various collective tasks.

## Results

We utilize the high-resolution movement data from bird flocks exhibiting various motion patterns, including mobbing[15], circling[16], and transit[15]. The mobbing and transit datasets were derived from video recording capturing the 3D movements of all individuals within flocks of wild jackdaws (*Corvus monedul*) in Cornwall, UK[13–15]. The circling dataset comprised 3D tracks reconstructed from video recordings of a flock of 1800 chimney swifts (*Chaetura pelagica*) as they entered an overnight roost in Raleigh, USA[16]. The mobbing flocks recorded the collective anti-predator events during which individuals gathered together to inspect and repel a predator (Fig. 1a and Supplementary Movie 1). The circling flocks with hundreds of chimney swifts displayed the circling approach pattern from surrounding areas near a roost site (Fig. 1b and Supplementary Movie 2). The transit dataset showed the highly ordered and smooth movement of jackdaw flocks flying towards their winter roosts (Fig. 1c and Supplementary Movie 3). See Supplementary Note 1, 2 and Supplementary Fig. 1 for detailed information about the data collection and processing of three datasets. In this study, we totally get 140, 94, and 1483 tracks of mobbing, circling, and transit flocks, respectively. See Supplementary Figs. 2–10 for the overview of flocking trajectories. Supplementary Fig. 11 counts the flock size and flocking time of all flocks analyzed in this work. Besides, we use four metrics, e.g., group order, trajectory curvature, group density and instability of neighbor[37] to systematically measure the properties of collective motions (see Supplementary Note 3 for the detailed definitions and results). Based on the aforementioned metrics of collective motion, three datasets could be classified into two distinct motion patterns: the mobbing and circling datasets demonstrate maneuver motion, characterized by collective sharp turns aimed at repelling predators or by engaging in collective circling behavior near a roost site. In contrast, the transit dataset exhibits a notably smoother and highly ordered motion, indicative of coordinated movement towards winter roosts.

### Flocks with higher maneuver motions display stronger nested leader-follower relations

To unveil the interactions embedded in flocks exhibiting different patterns of collective motions, we first focus on quantitatively assessing how leader-follower (LF) relations[12] are structured within the flocks, as the interactions among neighbors could be viewed as the influences exerted by leaders on followers. The LF relationship, as defined in this study, pertains to the emergent leadership resulting from the consensus of velocity among pairs of individuals within the flock, distinct from the underlying social structure that some individuals are more influential than others. For a pair of birds $i$ and $j$ during the time period $[t_s, t_e]$, their normalized temporal flying directions are denoted as $\hat{\mathbf{v}}_i(t)$ and $\hat{\mathbf{v}}_j(t)$, respectively. For time stamp $t$ and a time interval $\tau$, the degree of motion alignment of this pair is defined as $C_{ij}(t,\tau) = \hat{\mathbf{v}}_i(t) \cdot \hat{\mathbf{v}}_j(t+\tau)$. Here the sign of $\tau$ means the lag ($\tau < 0$) or advance ($\tau > 0$) time of focal $i$ from the view of neighbor-$j$. For a given $\tau$, we average $C_{ij}(t,\tau)$ over all the time stamps $t$ from the period $[t_s, t_e]$, yielding $\langle C_{ij}\rangle(\tau) = \langle C_{ij}(t,\tau)\rangle$, which indicates the LF relation between birds $i$ and $j$ in the flock within $[t_s, t_e]$ (see "Methods").

The value of $\tau_{ij}^{LF}$ that makes $\langle C_{ij}\rangle(\tau)$ reach the maximal value is referred to as the leading or lag time of this pair's LF relation. $\tau_{ij}^{LF} < 0$ (or $> 0$) means that the flying direction of individual-$i$ lags $|\tau_{ij}^{LF}|$ seconds behind (or advances $|\tau_{ij}^{LF}|$ seconds before) that of individual-$j$ to keep the maximal consensus. It could be interpreted as a case of individual-$i$ following (or leading) individual-$j$. For example, in a mobbing flock shown in Fig. 1a, due to $\tau_{61}^{LF} = -0.87 < 0$, bird-1 is the leader of bird-6, because bird-6 falls 0.87 s behind to maintain the maximal alignment with bird-1 (Fig. 1d and Supplementary Fig. 16).

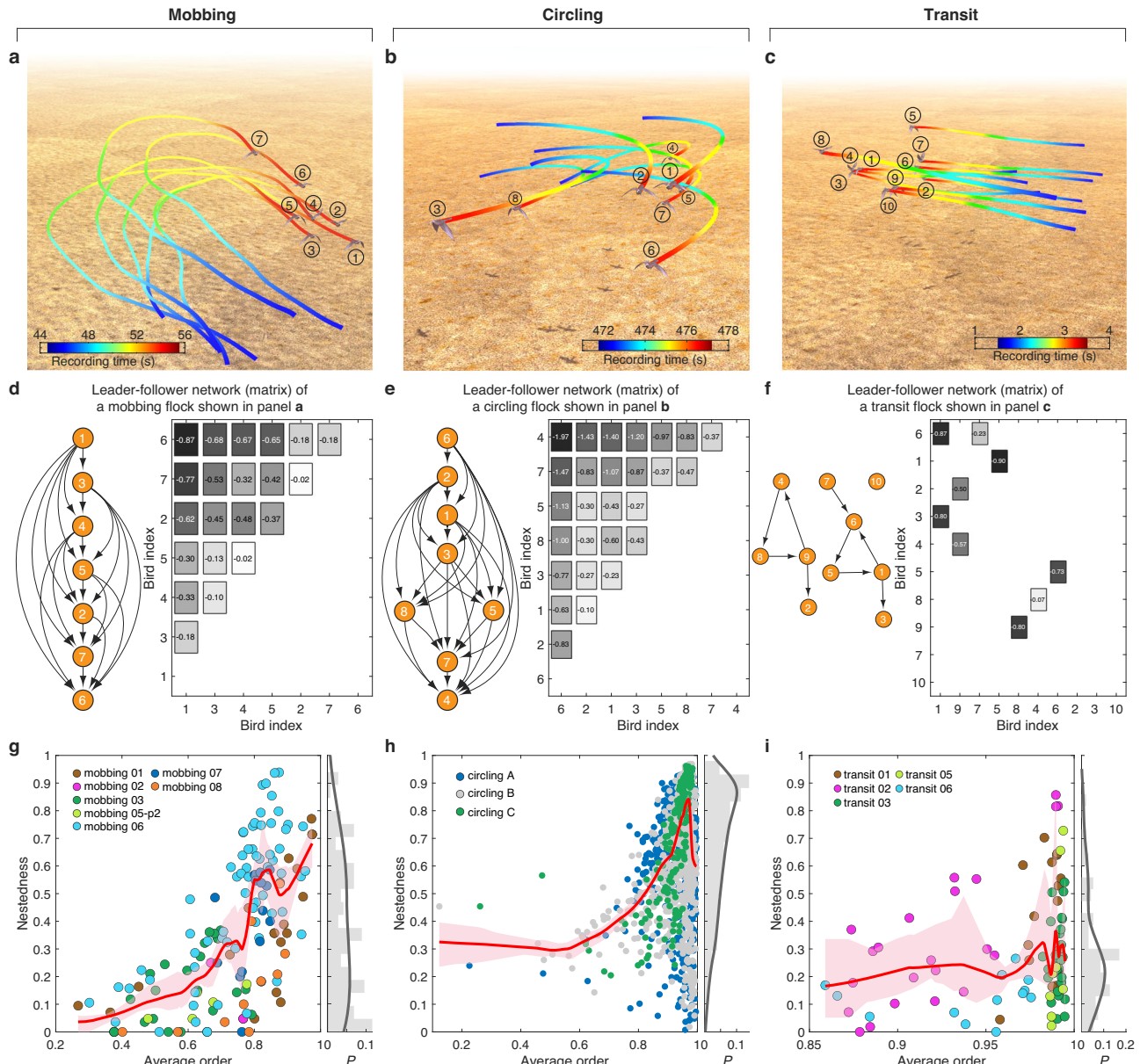

**Fig. 1 | The leader-follower relations of different patterns of collective motions.** The flocking trajectories are from three datasets: mobbing (**a**), circling (**b**), and transit (**c**), which are categorized into maneuver (mobbing and circling) and smooth (transit) modes of collective motions. In (**a**–**c**), the gradient color from blue to red corresponds to the recording time from beginning to end. **d**–**f** The LF relation matrix and corresponding LF network of three flocks shown in panels **a**–**c**, respectively. The negative values ($\tau_{ij}^{LF} < 0$) in LF relation matrix represent that individual-$j$ leads individual-$i$, and equals the directed edges pointing from leader-$j$ to follower-$i$ in the corresponding LF networks. The box colors from black to white correspond to the descending order of absolute value of $\tau_{ij}^{LF}$. The axis in (**d**–**f**) is ordered by the nestedness of their corresponding LF networks. The nestedness of

LF networks as a function of average order for three flocking datasets of mobbing (**g**), circling (**h**), and transit (**i**). The right marginal plots show the nestedness distribution of LF networks. The order parameter is a temporal measurement evaluating the motion polarization of a flock at time $t$, i.e., $\sum_{i=1}^{N} \hat{v}_i(t)/N$, and the average order is the mean of polarization over the whole time. In (**d**–**i**), the LF relation matrices are calculated from the whole period of each flock, and the nestedness is measured by NODF. In (**g**–**i**), each point represents a flock from three datasets, and the nonparametric regression and bootstrap sampling are performed to calculate the trend (red curve) and its 94% confidence interval (red shadow) between nestedness and average order.

For a flock within $[t_s, t_e]$, we construct an LF relation matrix $T_{LF}(t_s, t_e)$ by assigning its $(i,j)$-entry as $\tau_{ij}^{LF}$ if $\tau_{ij}^{LF} < 0$; and 0 otherwise (see matrices in Fig. 1d–f). An LF relation matrix can also be represented as a directed network, where the directed edge points from the leader $j$ to its follower $i$ if there exists $\tau_{ij}^{LF} < 0$ (see LF networks in Fig. 1d–f).

Interestingly, we find that the LF network of the mobbing (or circling) flock displays a highly nested structure[38–41] and is completely hierarchical[42] (Fig. 1d, e). This finding observed in the jackdaw and swift

flocks is consistent with previous findings in pigeon flocks[12]. Here, the nested structure means that for those birds with few leaders (i.e., they have a high position in the hierarchy), their leaders also tend to be the leaders of those birds who have many leaders and a low position in the hierarchy. A completely hierarchical structure means that birds with lower positions will never be leaders of birds with higher positions in the hierarchy. However, for the transit flock shown in Fig. 1c, its LF relation matrix doesn't display neither a nested nor a hierarchical structure (Fig. 1f).

To systematically study the hierarchy in the flocks with different motion patterns, we calculate the nestedness of LF networks for the three flocking datasets (see Supplementary Note 4 for discussion of nestedness metric). Note that we calculate the nestedness by binarizing the LF matrix, meaning that we set non-zero elements as 1 and all other elements as 0. The distributions of nestedness values displayed in marginal plots of Fig. 1g–i demonstrate that the nestedness of transit flocks is significantly lower than that of mobbing and circling flocks (p-value = 0.017 and 5.54e−52, respectively, Mann–Whitney U-test). Furthermore, we plot the nestedness of the LF network of a flock as a function of its average order for three flocking datasets. Here, the order of a flock at time $t$ is a temporal measurement evaluating its motion polarization, and the average order of a flock is the average of its order over the entire period of measurement. We find that both mobbing and circling flocks exhibit a general positive correlation between their average order and the nestedness of their LF networks (Fig. 1g, h). However, this positive correlation is not evident in transit flocks (Fig. 1i). Furthermore, these findings are consistently supported across separate flock records (a collection of some tracks, e.g., mobbing-01), as well as different group sizes and flocking durations (Supplementary Figs. 20–22).

## Perception of motion salience measures the relative motion changes

Overall, the comparison across three datasets indicates that the mobbing and circling flocks tend to exhibit a more distinct hierarchy of LF relations compared to transit flocks. In the case of mobbing or circling flocks, we could infer that the clear LF structure and hierarchy contribute to high group order. Similarly, the hierarchy in LF networks emerges when the flocks with maneuver motion display high group order. Regrettably, the relation is not clearly evident in transit flocks. These findings lead us to pose a fundamental question: what are the interaction rules responsible for the emergence of nested and hierarchical LF relations? To address this question, we investigate how individuals perceive changes in the motion of their neighbors. The emergence of leading relationships occurs as individuals adjust their movements based on the perceived motions of their neighbors.

In cognitive science, a seminal model known as the salience model, along with its variants has documented human shifts of visual attention in both spatial and temporal dimensions[43,44]. Coincidentally, empirical evidence in fish schooling has revealed the selective attention phenomena, wherein individual directional decisions are influenced by the relative strength of visual features among neighbors[45–47]. Furthermore, a large-scale motion-capture system has unveiled the visual attention of freely-behaving pigeons[48]. Drawing inspiration from these findings, we recognize that individual perception should incorporate attention information from the first-person perspective, particularly regarding movement changes of neighbors over a time period.

Therefore, we propose a heuristic measure called individual perception of motion salience (MS) to quantify the relative movement changes of neighbor-$j$ from the focal individual-$i$'s first-person view within the period $[t − τ, t]$ in the form of

$$M_{ij}(t,\tau) = \frac{\angle(\hat{\mathbf{x}}_{ij}(t),\hat{\mathbf{x}}_{ij}(t-\tau))}{\tau} \left(\frac{1+\hat{\mathbf{v}}_i(t)\cdot\hat{\mathbf{x}}_{ij}(t)}{2}\right)^{\alpha} \left(\frac{1+\hat{\mathbf{v}}_i(t-\tau)\cdot\hat{\mathbf{x}}_{ij}(t-\tau)}{2}\right)^{\alpha}$$
$$(i \neq j, j \in \mathcal{S}_i, 0 < \tau < t). \quad (1)$$

$\hat{\mathbf{x}}_{ij}(t) = (\mathbf{x}_j(t) − \mathbf{x}_i(t))/||\mathbf{x}_j(t) − \mathbf{x}_i(t)||$, $\mathbf{x}_j(t)$ is individual-$j$'s position vector at time $t$, $\angle$ means the angle between two vectors in 3D Cartesian coordination, and $\mathcal{S}_i$ indicates the collection of neighbors of focal-$i$. Note that due to the limitation of recording area in the original observation studies, we roughly assume that, for a bird flock from three dataset, $\mathcal{S}_i$ represents all individuals appeared in the flock when calculating $M_{ij}(t,\tau)$. Here $τ$, being $0 < τ < t$, means the lag time for individuals $i$ and $j$ at time $t$, and $τ$ is also referred to as the perceiving

time. In line with a few flock models[37,49], $τ$ shares the similar view that the sensory information during an interval is calculated and then updated. We underscore that the definition of MS represents a heuristic endeavor aimed at quantitatively capturing the relative changes in motions between a pair of birds, rather than constituting a computational visual model of a bird's actual visual perceptual capabilities. This formulation enables the discrete-data analysis of real flocks and its direct applicability in swarm robotics.

The diagram of definition of motion salience is shown in Fig. 2a. $M_{ij}(t,\tau)$ comprehensively quantifies the perception of neighbor-$j$'s motion changes relative to the focal individual-$i$ during the period $[t − τ, t]$. The larger $M_{ij}(t,\tau)$, the more pronounced movement of individual-$j$ felt by individual-$i$ during $[t − τ, t]$. Note that while MS equation does not directly account for variations of velocity and speed, we observe that it effectively encompasses these variations by taking into account the relative position changes between the initial and final time stamps (see detailed analysis in Supplementary Note 5).

In order to replicate the anisotropic effect of motion perception in real birds, the last two components on the right-hand of MS equation in Eq. (1) embody the forward-oriented preference in perceiving the motion of neighbor-$j$ from the first-person view of focal individual-$i$ (Fig. 2b). It simulates that, for the focal individual-$i$, its perception capability diminishes as the first-person view transitions from the front to the back. Here the first-person view of individual-$i$ aligns with itself heading at time $t$. For example, if $\angle(\hat{\mathbf{v}}_i, \hat{\mathbf{x}}_{ij}) = 0$, it means that neighbor-$j$ locates directly ahead of the focal-$i$'s movement direction, and the focal individual could fully perceive the neighbors because $(1 + \hat{\mathbf{v}}_i \cdot \hat{\mathbf{x}}_{ij})/2$ is 1. Once $\angle(\hat{\mathbf{v}}_i, \hat{\mathbf{x}}_{ij})$ deviates from 0 to $π$ or $−π$, it represents that the neighbor-$j$ gradually moves from the front to back of the focal-$i$, and assumes the ability of motion perception could increasingly diminish since $(1 + \hat{\mathbf{v}}_i \cdot \hat{\mathbf{x}}_{ij})/2$ decreases from 1 to 0.

We incorporate the anisotropic factor $\alpha \geq 0$ in MS to control the fact of forward-oriented preference in biological perception[16,50,51] (Fig. 2c). If $\alpha = 0$, $M_{ij}(t,\tau)$ ignores the blind area of perception because the last two components in Eq. (1) always equal to 1 regardless of the relative positions of neighbors. Increasing $\alpha$ could make Eq. (1) amplify the anisotropic effect of motion perception, that is, the ability of individual perception gradually becomes more restricted towards the frontal field of vision. For instance, $\alpha = 10$ controls $M_{ij}(t,\tau) \approx 0$ when the neighbor-$j$'s relative position is beyond the horizontal sight $(−π/2, π/2)$ of the focal individual-$i$ (Fig. 2c).

## Individuals with higher motion salience tend to lead the group in the maneuver motions

According to Eq. (1), we could construct MS matrix $M(t,\tau) = [M_{ij}(t,\tau)]_{N \times N}$ and derive the average MS of each individual $M_i(t,\tau)$ by averaging each column of $M(t,\tau)$. Moreover, besides the nestedness value capturing the hierarchy of LF relations at the group level, we also calculate the local reaching centrality[42] to quantify the leading tier of each individual in an LF network derived from a flock within the time period, labeled as $L_i(t,\tau) \in [0,1]$ (see "Methods"). The larger the $L_i$, the higher the leading role of individual-$i$. As illustrated in Fig. 1d, bird-1 positioned at the top of LF network guides all the downstream individuals, resulting in the largest $L_1 = 1$. Conversely, Bird-6, situated at the bottommost tier of the LF network, exhibits the lowest $L_6 = 0$.

An in-depth study of the correlation between leader-follower (LF) and motion salience (MS) enables us to disclose the role of individual motion characters during the group formation to answer a fundamental question: what kind of motion characteristics does an individual possess to lead the flock? Leader's motion character could be interpreted as the interaction rule about how the focal aligns with neighbors after perceiving their motions. For example, according to the mobbing flock shown in Fig. 1a, we calculate the LF network and MS matrix ($\alpha = 0$) from a short period $[t − τ, t]$, and then derive each

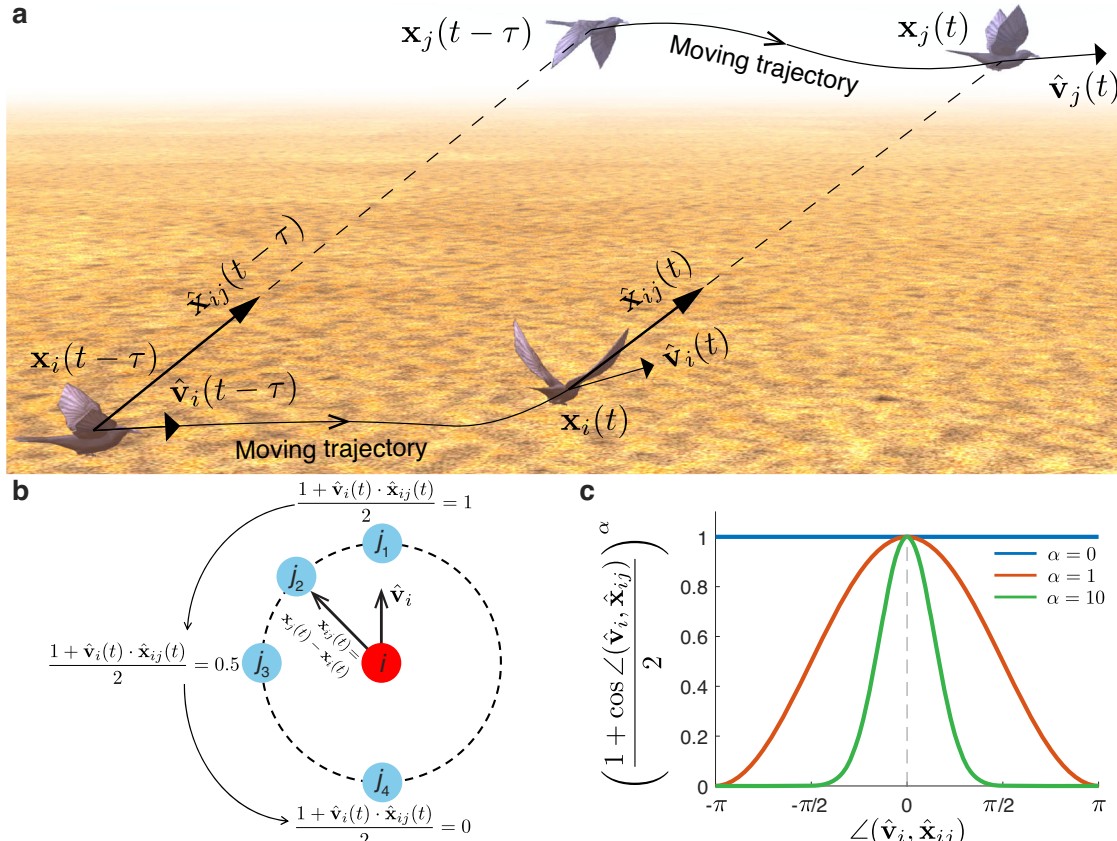

**Fig. 2 | Perception of motion salience in collective motions. a** Diagram of MS to quantify the relative movement changes of neighbor-$j$ from the focal individual-$i$'s first-person view. **b** The anisotropic effect of motion perception in Eq. (1) mimics the idea that the perception capability diminishes as the focal individual's first-person sight extends from the front to the back. **c** The anisotropic factor $\alpha \geq 0$ in MS controls the fact of forward-oriented preference in biological perception. The $x$-axis indicates the heading between two vectors $\hat{\mathbf{v}}_i$ and $\hat{\mathbf{x}}_{ij}$, and $y$-axis corresponds to the second or third term of right-hand of Eq. (1). If $\alpha = 0$, we ignore the blind area of motion perception. With increasing $\alpha > 0$, we assume that the ability of individual perceiving movements of around neighbors gradually narrows to the front vision.

individual's leading tier ($L_i(t,\tau)$) and average MS ($M_i(t,\tau)$) (Fig. 3a). Interestingly, the two vectors composed of $L_i(t,\tau)$ and $M_i(t,\tau)$ show a strong positive correlation. Finally, for the mobbing flock within $[t - \tau, t]$, we compute the Spearman correlation coefficient ($\rho$) between the two vectors composed of $L_i(t,\tau)$ or $M_i(t,\tau)$ over different combinations of $t$ and $\tau$ (see Fig. 3b for $\alpha = 0$, Fig. 3e for $\alpha = 1$ and Supplementary Movie 4). Compared with the mobbing flock (shown in Fig. 1a), we perform the same correlation analysis for circling and transit flocks shown in Fig. 1b, c (see Fig. 3b–g). An intriguing observation from our study is that, when considering the forward-oriented preference of visual perception ($\alpha = 1$), both mobbing and circling flocks, renowned for their highly maneuverable motions, demonstrate a predominance of positive correlations between LF and MS across different combinations of $t$ and $\tau$ (Fig. 3e, f). This suggests that individuals with higher MS tend to play the higher-tier leading role for the majority of a flock's duration. By contrast, the transit flock displays well-mixed positive and negative correlations between LF and MS at different combinations of $t$ and $\tau$ (Fig. 3g).

Next we extend the correlation analysis to three flocking datasets (Fig. 3h–j). When $\alpha = 0$, three flocking datasets display quite similar $\rho$ distribution centered around 0. However, if we consider the forward-oriented sight of birds[16,50,51] ($\alpha > 0$), the two flocking datasets with maneuver motions display the distribution of $\rho$ dominated by positive values, while the transit flock dataset still display the distribution of $\rho$ centered around 0. The domination of positive $\rho$ occurred in the situation of forward-oriented preference of perception ($\alpha > 0$) accords with the spatial structure of the leading position, which aims to investigate the relations between the leading tier and fraction of

neighbors present in the front view (Supplementary Note 6). The findings illustrate that in mobbing and circling flocks, the higher leading tier an individual occupies, the more likely it will be at the front of flock to lead the group. In contrast, the transit flocks do not show any preference between the leading tier and neighbors' relative position (Supplementary Fig. 25). Furthermore, even when controlling for the average speed of a flock (e.g., categorizing three flocking datasets by average speed in 1 m/s intervals or average angular speed in 0.1 rad/ s intervals), we consistently observe predominantly positive correlations between LF and MS across the majority of speed ranges in both mobbing and circling datasets. This trend persists irrespective of whether the categorization is founded on average speed or average angular speed. However, such a pattern is not evident in the transit dataset (Supplementary Figs. 26 and 27).

Given the well-known finding that faster individuals tend to assume leadership positions in animal groups[52,53], two natural concerns arise regarding the correlation of MS-Speed and comparison of correlation between LF-MS and LF-Speed. Here the Speed of each individual are three types: average speed, average radial speed and average angular speed. See Supplementary Note 7 for detailed introduction and results. Interestingly, regardless of three flocking datasets and $\alpha$ values in MS, correlation between MS and three types of Speed do not demonstrate a significant positive or negative dominance. Moreover, we perform the correlation analysis between LF and Speed, and then compare the correlation of LF-MS and LF-Speed in three datasets. We find that the flocks (or sub-flocks) with maneuver motions, such as mobbing and circling flocks, show a notably stronger positive correlation of LF-MS than that of LF-Speed (Supplementary Fig. 31).

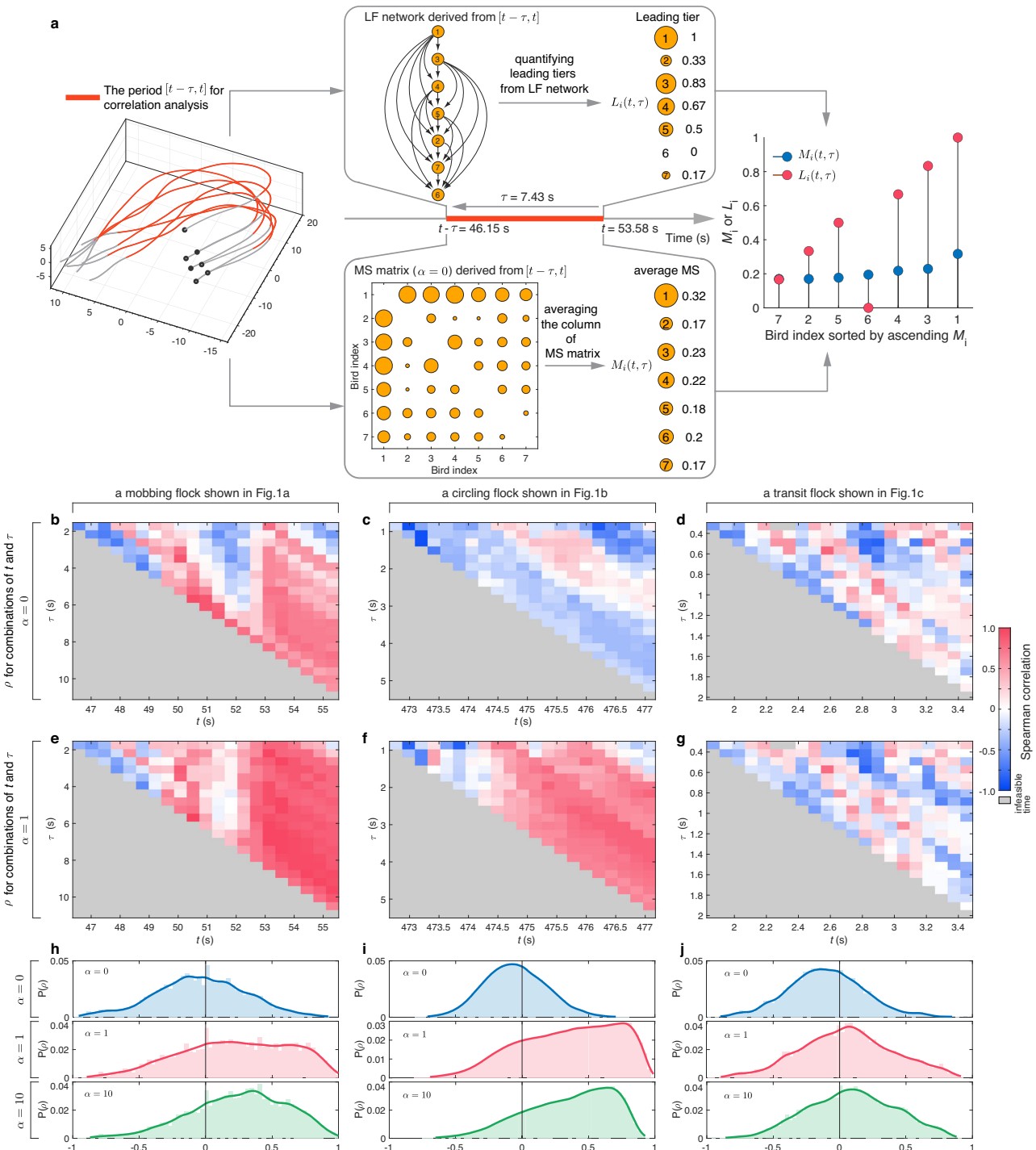

**Fig. 3 | Correlation analysis between LF and MS in collective motions. a** For a flock within the period $[t-\tau,t]$ (highlighted by red), first we could construct the LF network and MS matrix ($\alpha=0$), and consequently derive the leading tier $L_i(t,\tau)$ and average MS $M_i(t,\tau)$ of each individual. Interestingly, two vectors composed of $L_i(t,\tau)$ and $M_i(t,\tau)$ show a strong positive correlation. Finally, we perform the Spearman correlation ($\rho$) between two vectors composed of $L_i(t,\tau)$ or $M_i(t,\tau)$ over the different combinations of $t$ and $\tau$. The heatmap of $\rho$ under $\alpha=0$ (**b**–**d**) and $\alpha=1$

(**e**–**g**) for mobbing, circling and transit flocks shown in Fig. 1a–c. The gradient color from blue to white to red maps $\rho\in[-1,1]$, respectively. The gray color means the infeasible time for Eq. (1), i.e., $t-\tau<0$. The distribution of $\rho$ over different combinations of $t$ and $\tau$ from mobbing (**h**), circling (**i**), and transit (**j**) datasets when $\alpha=0$, $\alpha=1$, and $\alpha=10$. In (**b**–**d**), for each flock, we take 21 time stamps for $t$ and 22 time points for $\tau$.

Conversely, for the transit flocks, the correlation of LF-Speed is significantly greater than that of LF-MS (Supplementary Fig. 32). In summary, LF-MS may be more effective in describing the flocks with maneuver motions, whereas LF-Speed is better suited for characterizing flocks with smooth motions. These finding do not contradict the

well-known property that faster birds tend to occupy leadership positions[53,54]. One reason is that we also observe a tendency for leaders to fly at the front of the flock in the datasets we analyze (Supplementary Fig. 25). The difference between MS and Speed could mainly be attributed to the scope of their respective definitions. As per the

definition, MS directly or indirectly encompasses variations in position, heading, speed and acceleration over a period of flock, whereas three types of speed serve as scalar representations of collective motions. It becomes apparent that speed, in contrast to MS, does not offer a comprehensive attribute for discerning the modes of collective motion, particularly in the case of highly agile flocks.

## Individuals will accelerate convergence of velocity with neighbors who have higher MS

Nonetheless, the empirical observation pertaining to MS and LF is currently situated at the group level, underscoring the necessity for additional research at the individual level to elucidate how individuals align with neighbors upon perceiving their MS. Such investigations will serve to strengthen the justification for translating empirical insights derived from real flocks into mathematical models in swarm robotics. In this context, our focus lies in individual-$i$'s computation of its neighbor's MS within the past interval $[t - \tau_{\mathrm{pre}}, t]$, aimed at examining the future trend of motion changes between the focal individual and its neighbors during the subsequent time span of $[t, t + \tau_{\mathrm{next}}]$. For the mobbing flock shown in Fig. 4a, b shows the perception of neighbors' $M_{1j}(t, \tau_{\mathrm{pre}})$ (and the normalization $w_{1j}(t, \tau_{\mathrm{pre}}) = \frac{M_{1j}(t, \tau_{\mathrm{pre}})}{\sum_{j \in S_1} M_{1j}(t, \tau_{\mathrm{pre}})}$) from the focal bird-1 within the past period $[t - \tau_{\mathrm{pre}}, t]$ (colored by blue in Fig. 4a), and Fig. 4c displays the temporal velocity consensus $\langle \mathbf{v}_1(t) \cdot \mathbf{v}_2(t) \rangle$ between bird-1 and 2 within the subsequent period $[t, t + \tau_{\mathrm{next}}]$ (colored by red in Fig. 4a). We use two kinds of metrics to describe the trend of motion changes between a pair of birds.

A common approach to represent motion trend is by calculating the average of temporal velocity consensus over the interval $[t, t + \tau_{\mathrm{next}}]$, denoted as $\phi_{ij}(t, \tau_{\mathrm{next}}) = \langle \mathbf{v}_i(t) \cdot \mathbf{v}_j(t) \rangle$. Then for bird-1, we perform the correlation analysis between two vectors composed of $w_{1j}(t, \tau_{\mathrm{pre}})$ and $\phi_{1j}(t, \tau_{\mathrm{next}})$, respectively (Fig. 4d). It is intriguing that across all individuals present in the flock, negative correlations between $w_{ij}(t, \tau_{\mathrm{pre}})$ and $\phi_{ij}(t, \tau_{\mathrm{next}})$ are consistently prevalent for different combinations of $\tau_{\mathrm{pre}}$ and $\tau_{\mathrm{next}}$ (Fig. 4f). Moreover, we also notice the persistence of this negative correlations in three flocking datasets (Supplementary Fig. 34). This result agrees well with our intuition, as seen from the perspective of individual-$i$. When perceiving its neighbor-$j$ who moved with significant motion changes (reflected in the higher value of $w_{ij}(t, \tau_{\mathrm{pre}})$) in the past period $[t - \tau_{\mathrm{pre}}, t]$, it could lead to a less synchronized velocity consensus (indicated by a lower value of $\phi_{ij}(t, \tau_{\mathrm{next}})$) in the subsequent period $[t, t + \tau_{\mathrm{next}}]$.

Another method is calculating the average slope of temporal velocity consensus within the future period, which is denoted as $k_{ij}(t, \tau_{\mathrm{next}}) = \langle$ slope of the curve $\mathbf{v}_i(t) \cdot \mathbf{v}_j(t)$ at time $t \rangle$ (Fig. 4c) and the slope at time $t$ is calculated as $\mathbf{v}_i(t) \cdot \mathbf{v}_j(t) - \mathbf{v}_i(t-1) \cdot \mathbf{v}_j(t-1)$. From the definition, unlike $\phi_{ij}(t, \tau_{\mathrm{next}})$ measures the degree of velocity consensus in the future period, $k_{ij}(t, \tau_{\mathrm{next}})$ directly depicts the change rate of the convergence or divergence of velocity consensus. It is worth noting that the value of $\phi_{ij}(t, \tau_{\mathrm{next}})$ calculated from the subsequent period may be impacted by the past $M_{ij}(t, \tau_{\mathrm{pre}})$. For example, a higher $M_{ij}(t, \tau_{\mathrm{pre}})$ may lead to the reduced velocity consensus between a pair of birds by the end of $[t - \tau_{\mathrm{pre}}, t]$, subsequently resulting in a lower level of $\mathbf{v}_i(t) \cdot \mathbf{v}_j(t)$ at the beginning of $[t, t + \tau_{\mathrm{next}}]$. However, $k_{ij}(t, \tau_{\mathrm{next}})$ overcomes this issue by solely indicating the rate of changes of $\mathbf{v}_i(t) \cdot \mathbf{v}_j(t)$ within the interval $[t, t + \tau_{\mathrm{next}}]$, irrespective of whether $\mathbf{v}_i(t) \cdot \mathbf{v}_j(t)$ is high or low at the beginning of this interval.

Clearly, a larger value of $k_{ij}(t, \tau_{\mathrm{next}})$ implies that two individuals are aiming to align with each other more rapidly in the future. Figure 4e shows the correlation analysis between two vectors composed of $w_{1j}(t, \tau_{\mathrm{pre}})$ and $k_{1j}(t, \tau_{\mathrm{next}})$ for bird-1. Interestingly, Fig. 4g indicates that among all individuals, positive correlations between $w_{ij}(t, \tau_{\mathrm{pre}})$ and $k_{ij}(t, \tau_{\mathrm{next}})$ prevail consistently across different combinations of $\tau_{\mathrm{pre}}$ and $\tau_{\mathrm{next}}$. Furthermore, we also observe the enduring presence of this

positive correlations in three flocking datasets (MS with $\alpha = 0$ in Fig. 4h–j and MS with $\alpha = 1$ in Supplementary Fig. 35). These phenomena may assist us in comprehending why individuals with higher MS tend to occupy the higher leading tier, because the positive correlations could be interpreted that if neighbor-$j$ is perceived to have a higher MS by the focal individual-$i$ (equals to higher $w_{ij}(t, \tau_{\mathrm{pre}})$), the focal one will be quicker to align with neighbor-$j$'s velocity (equals to higher $k_{ij}(t, \tau_{\mathrm{next}})$). This perhaps leads to a time delay for individual-$i$ to converge with the heading of neighbor-$j$. Coincidentally, the time delay accords with definition of an LF relation from neighbor-$j$ to individual-$i$ in this study.

To ensure a thorough analysis of the results, we conduct the correlation analysis between $w_{ij}$ and $k_{ij}$ (or $\phi_{ij}$) within the same period of $[t - \tau_{\mathrm{pre}}, t]$ (Supplementary Fig. 36a), instead of using $w_{ij}$ from $[t - \tau_{\mathrm{pre}}, t]$ and $k_{ij}$ (or $\phi_{ij}$) from $[t, t + \tau_{\mathrm{next}}]$ respectively. The results show that $\phi_{ij}(t, \tau_{\mathrm{pre}})$ continues to exhibit a predominance of negative correlation with $w_{ij}(t, \tau_{\mathrm{pre}})$ (Supplementary Fig. 36b–d). It makes sense that substantial alterations manifest in the relative positions of a pair of birds, simultaneously resulting in a reduced level of synchronized velocity consensus. However, shifting our focus to $k_{ij}(t, \tau_{\mathrm{pre}})$, we could not observe a prevalence of positive correlations between $w_{ij}(t, \tau_{\mathrm{pre}})$ and $k_{ij}(t, \tau_{\mathrm{pre}})$ as opposed to the emergence of such correlations between $w_{ij}(t, \tau_{\mathrm{pre}})$ and $k_{ij}(t, \tau_{\mathrm{next}})$ in a consecutive timeframe (Supplementary Fig. 36e–g). The disparity in correlation outcomes obtained from $k_{ij}(t, \tau_{\mathrm{next}})$ and $k_{ij}(t, \tau_{\mathrm{pre}})$ respectively confirms the existence of a time delay for a bird to adapt its heading subsequent to perceiving its neighbors' motion.

Hence, our conclusion is that the prevalence of positive correlations between $w_{ij}(t, \tau_{\mathrm{pre}})$ and $k_{ij}(t, \tau_{\mathrm{next}})$ signifies that individuals will accelerate the convergence of velocity with neighbors who have higher MS. In other words, if individual-$i$ detects neighbor-$j$ with higher MS in the past, it will promptly modify its heading to align more swiftly with neighbor-$j$ in the future.

Finally, this empirical finding motivates us to introduce the concept of adaptive MS-based (AMS) interaction in swarm model, such as, $\mathbf{v}_i(t+1) = \hat{\mathbf{v}}_i(t) + \sum_{j \in S_i} w_{ij}(t) \hat{\mathbf{v}}_j(t)$ and $w_{ij}(t) = \frac{M_{ij}(t, \tau)}{\sum_{j \in S_i} M_{ij}(t, \tau)}$ (Fig. 5a and "Methods"). In AMS, the coefficient $w_{ij}$ signifies that the impact of neighbor-$j$'s velocity is directly proportional to $M_{ij}$. If individual-$i$ observes that neighbor-$j$ has a higher MS compared to other neighbors, individual-$i$ will weigh the influences of neighbor-$j$ more heavily to adapt its heading in the subsequent step. As a result, this leads to a better alignment between individual-$i$ and $j$.

## Adaptive MS-based interaction effectively captures the fundamental characteristics observed in real bird flocks

A natural concern arises regarding the capability of AMS to encapsulate the fundamental characteristics observed in actual bird flocks, particularly its influence on shaping the development of nested and hierarchical LF relations in collective motions. For the sake of comparison, we also consider two MS-free interactions: (i) adaptive interaction based on transient heading difference[55,56] (ATHD), which could adaptively make the neighbors with larger (or smaller) transient heading difference exert larger (or smaller) influences on the focal individual (Fig. 5b and "Methods"); (ii) average interaction[17] representing the standard Vicsek model where the focal agent equally interacts with its neighbors located in the sensing radius (Fig. 5c and "Methods").

To verify this, we use a self-propelled particle model in 3D, where agents follow the local interaction rules, and the additional potential well is imposed on one of individuals to lead the flock come back to the origin (see "Methods"). For example, starting from the same initial conditions, the flock trajectories in Fig. 5a–c show that AMS and ATHD interaction could make the informed individual (red trajectories)

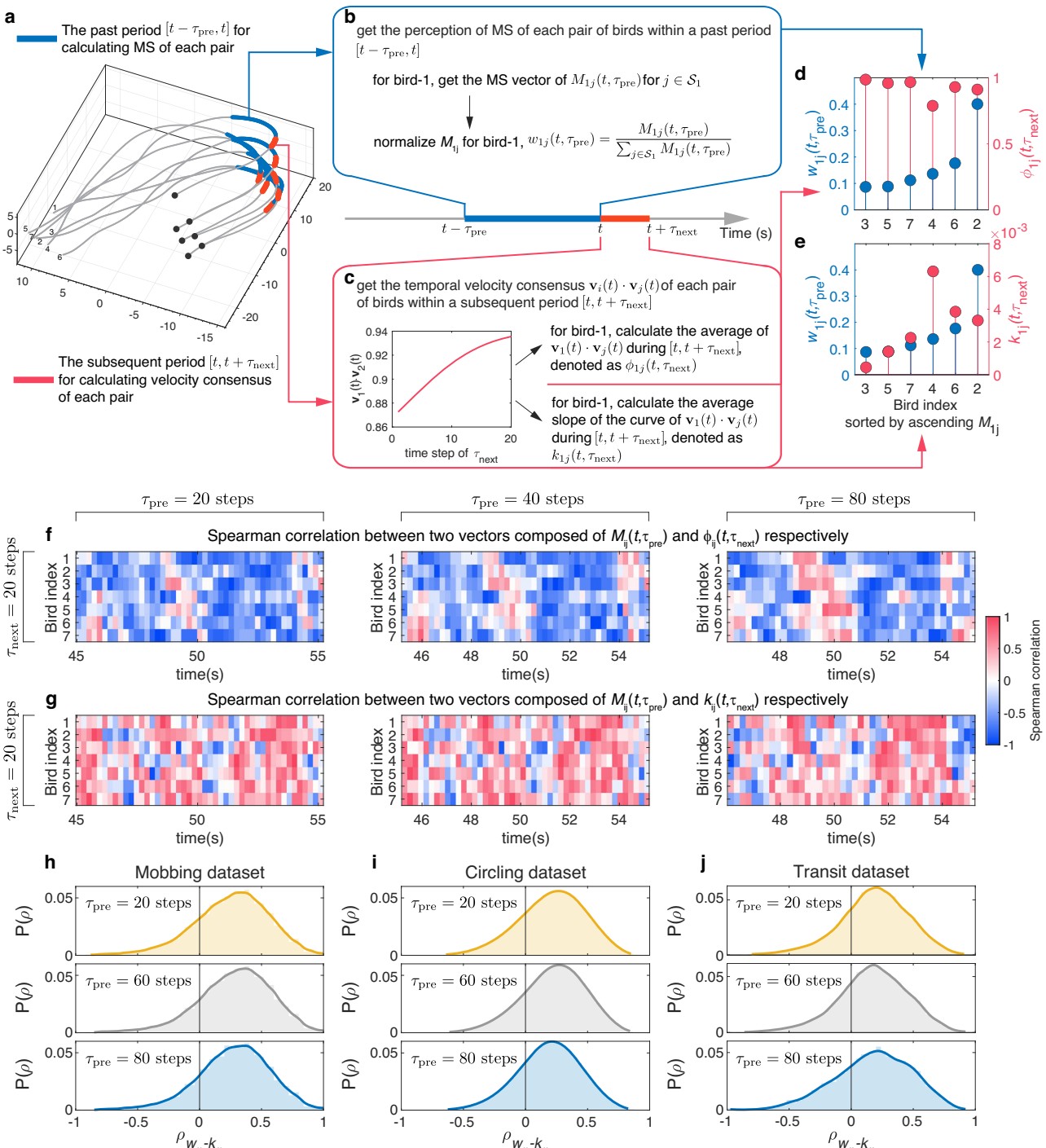

**Fig. 4 | Correlation analysis between MS and average of (or average slope of) temporal velocity consensus at the individual level.** We use the flock within the past time interval of $[t - \tau_{\text{pre}}, t]$ (highlighted by blue in panel **a**) and to generate $M_{1j}(t, \tau_{\text{pre}})$ (and the normalization $w_{1j}(t, \tau_{\text{pre}}) = \frac{M_{1j}(t, \tau_{\text{pre}})}{\sum_{j \in \mathscr{S}_1} M_{1j}(t, \tau_{\text{pre}})}$, see panel **b**). **c** Then we examine two kinds of metrics to describe the future trend of motion changes between the focal individual-1 and neighbor-$j$ in the subsequent time period of $[t, t + \tau_{\text{next}}]$ (highlighted by red in panel **a**). One is the average of temporal velocity consensus, denoted as $\phi_{1j}(t, \tau_{\text{next}}) = \langle \mathbf{v}_1(t) \cdot \mathbf{v}_j(t) \rangle$. Another is calculating the average slope of $\phi_{1j}(t, \tau_{\text{next}})$ according to the curve of temporal velocity consensus (see inset in panel **c**), denoted as $k_{1j}(t, \tau_{\text{next}}) = \langle$slope of the curve $\mathbf{v}_1(t) \cdot \mathbf{v}_j(t)$ at time $t \rangle$. The slope at time $t$ is calculated as $\mathbf{v}_1(t) \cdot \mathbf{v}_j(t) - \mathbf{v}_1(t-1) \cdot \mathbf{v}_j(t-1)$. For individual-1

in the flock shown in (**a**), we could perform the correlation analysis between two vectors composed of $w_{1j}(t, \tau_{\text{pre}})$ and $\phi_{1j}(t, \tau_{\text{next}})$, respectively (**d**), and between two vectors composed of $w_{1j}(t, \tau_{\text{pre}})$ and $k_{1j}(t, \tau_{\text{next}})$, respectively (**e**). Across all individuals appeared in the flock, we observe the prevalence of negative correlations between $w_{ij}(t, \tau_{\text{pre}})$ and $\phi_{ij}(t, \tau_{\text{next}})$ (**f**), and positive correlations between $w_{ij}(t, \tau_{\text{pre}})$ and $k_{ij}(t, \tau_{\text{next}})$ (**g**) for different combinations of $\tau_{\text{pre}}$ and $\tau_{\text{next}}$. Note that one step corresponds to 1/60 s in mobbing and transit datasets, and 1/30 s in circling dataset. In (**h**–**j**), we also observe the enduring presence of positive correlations between $w_{ij}(t, \tau_{\text{pre}})$ and $k_{ij}(t, \tau_{\text{next}})$ among all birds in three flocking datasets. In this figure we take $\alpha = 0$ to calculate MS.

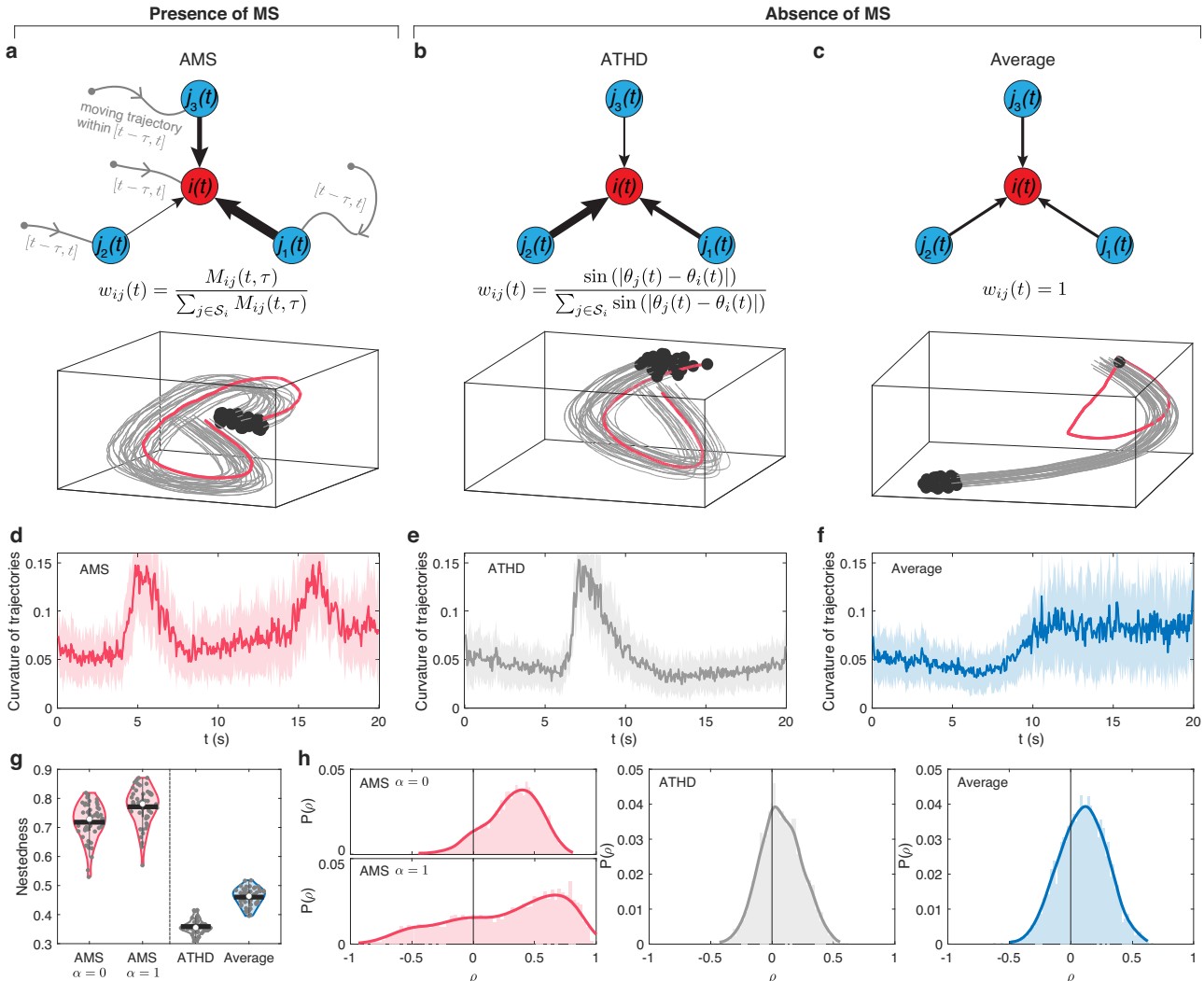

**Fig. 5 | Comparison of self-propelled particle model in presence or absence of MS.** We use a self-propelled particle model that particles follow the local interaction rules, i.e., AMS (**a**), ATHD (**b**) and average interaction (**c**). The additional potential well, imposed on one of individuals (red trajectories), aims to lead the flock come back to origin. The swarm size is 30 particles. The black points represent the end of trajectories. In (**a**), the flocking trajectories are generated by AMS with $\alpha = 0$. **d**–**f** The temporal curvature of flock trajectories respectively shown in (**a**–**c**). In (**d**–**f**), the solid curves represent the average of trajectory curvatures from 30 particles, and the shadow area represents the standard deviation (SD). **g** The distribution of nestedness of LF networks derived from flocking trajectories using AMS, ATHD and average interaction. The white points (or black lines) represent the median (or mean) value. **h** The distribution of Spearman correlation ($\rho$) between LF and MS over different combinations of $t$ and $\tau$ from flocking trajectories using AMS, ATHD and average interaction. For each flock we take 10 time stamps for $t$ and 11 time points for $\tau$ to perform the correlation analysis between LF and MS. In (**g**) and (**h**), we run 50 independent simulations for each interaction type.

successfully lead the whole flock, but the average interaction fails. Interestingly, we find that within the same period of time, the curvature of trajectories generated by AMS (two peaks in Fig. 5d) is larger than that of ATHD (one peak in Fig. 5e). It indicates that AMS could generate more maneuver motions than that of ATHD. Furthermore, in comparison with ATHD and average interaction, AMS interaction results in highly nested LF relations (Fig. 5g) and a prevalence of positive correlation between LF and MS (Fig. 5h).

To deeply understand the effect of AMS, we conduct the hybrid simulations by integrating AMS with ATHD (or average interaction). For instance, Supplementary Fig. 37 illustrates our approach where all individuals initially follow ATHD (or average interaction), followed by a gradual adjustment of the percentage of individuals utilizing AMS from 0% to 100% (x-axis of Supplementary Fig. 37a, b). Subsequently, we compare the nestedness of LF networks and distribution of LF-MS correlation across varying percentages of individuals employing AMS. Our findings indicate an increase in the nestedness of LF networks with a higher percentage of individuals utilizing AMS (Supplementary

Fig. 37a, b). Furthermore, as the number of individuals employing AMS increases, the distribution of LF-MS correlation across different combinations of $t$ and $\tau$ progressively shifts towards a prevalence of positive correlation (Supplementary Fig. 37c, d). The results validate the significant role of AMS interaction in shaping the emergence of nested and hierarchical LF relationships in collective motions, as well as replicating the trend of predominantly positive correlation between MS and LF.

Note that MS is capable of capturing variations in both azimuth and elevation angles in a 3D context, while in a 2D setting, it simplifies assessment to changes solely in the azimuth angle within a plane. The results obtained from simulations in 2D are consistent with those in 3D (Supplementary Fig. 38). In addition, we introduce a configurable parameter for the number of individuals leading the group to evaluate the effectiveness of leadership in this simulation (see Supplementary Note 8). Our findings indicate that AMS shows advantages over the other two MS-free interactions in terms of promoting effective leadership using fewer informed individuals.

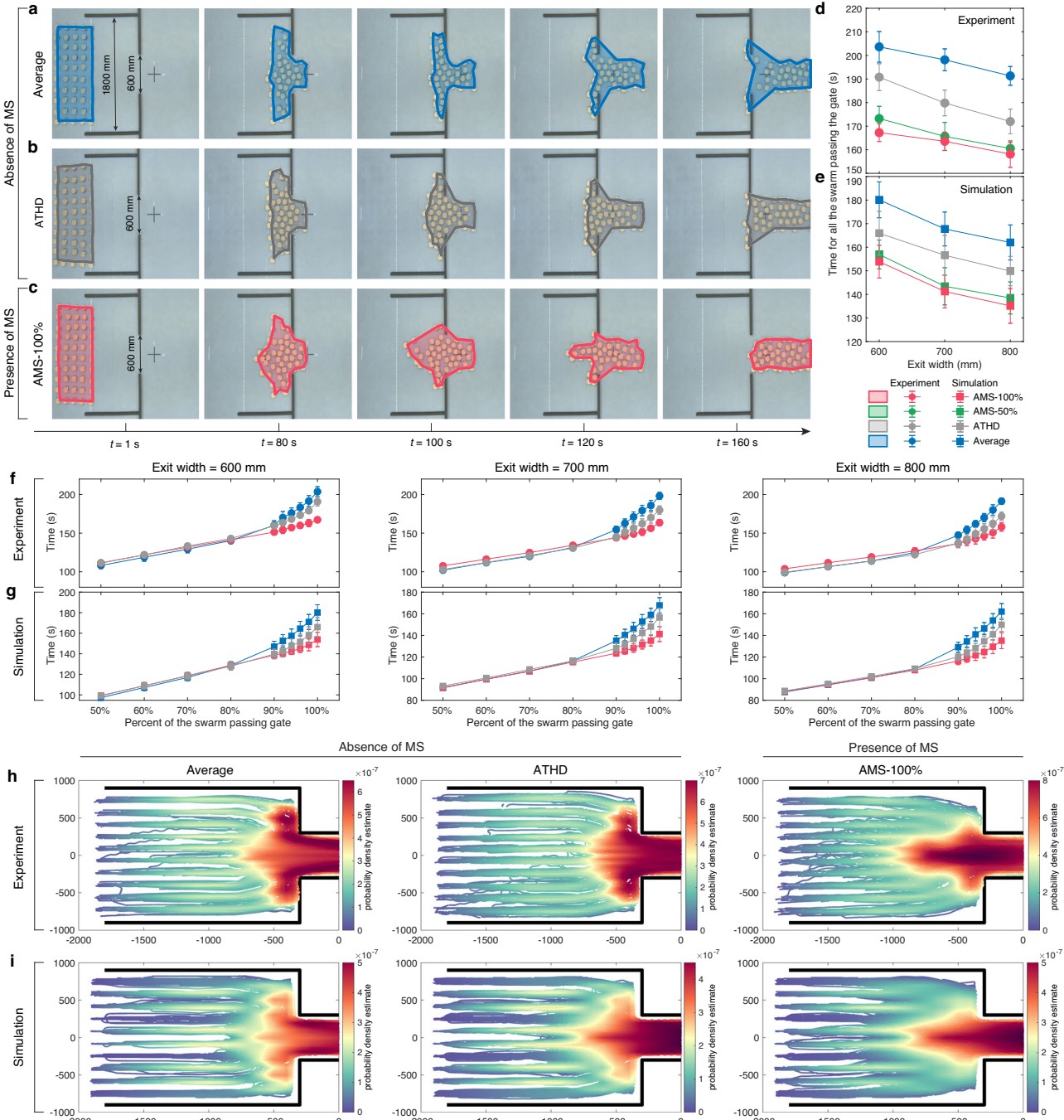

**Fig. 6 | Collective evacuation experiments.** The snapshots of collective evacuation experiments for the swarm consisted of 50 robots using average interaction (**a**), ATHD (**b**) and AMS with 100% MS (**c**). The exit width is 600 mm. The colored areas highlight the spatial distributions of the swarm. The spending time of all the swarm evacuating the narrow exit as a function of different exit widths from experiments (**d**) and simulations (**e**). The spending time as a function of different percent of the swarm successfully passing the gate from experiments (**f**) and simulations (**g**). In (**d**–**g**), the error bar represents the standard deviation (SD) calculated from 10 independent experiments and 50 independent simulations. The statistics of historical trajectories of all 50 robots in 10 independent experiments (**h**) and 50 independent simulations (**i**). The color scales in (**h**), and (**i**) correspond to the probability density estimate of historical trajectories. The black line indicates the wall. The swarm size is 50 in the experiments and simulations.

## Collective evacuation experiments in swarm robotics

Considering the impact of AMS in collective motions, we further adopt it to swarm robotics (see "Methods" and Supplementary Fig. 40) to demonstrate the advantages of self-organization in swarm experiments. For simplicity in the experiments, we ignore the blind area of motion perception and thus set $\alpha = 0$ in AMS.

Collective evacuation experiments are designed to explore the impact of AMS on enhancing the self-organization of the swarm for evacuating a narrow exit (Fig. 6a). Similar to the application of a drone swarm successfully navigating in confined and cluttered environments, we integrate a state-of-the-art framework[57] into the collective evacuation experiments. This framework encompasses diverse forms of interactions, including but not limited to aligned movement, collision avoidance between agent or agent-wall, and guidance for agents to navigate towards the exit. In this study, AMS is only introduced into the alignment component, replacing the commonly used average

interaction in prior works[8,57], while retaining the other relevant terms (see "Methods" and Supplementary Fig. 41 for detailed information about the swarm model of collective evacuation). Note that this experiment has no informed robot to explicitly lead the swarm to evacuate the narrow exit.

The spatiotemporal distributions of the swarm with 50 robots to evacuate a narrow exit (600 mm) are shown in Fig. 6a–c. When the swarm utilizes average or ATHD interaction, the observed behavior aligns with the intuitive anticipation that in cases where the exit is not adequately spacious, the swarm would disperse and occupy the entire space within the narrow exit[58], ultimately leading to congestion (Fig. 6a, b and Supplementary Movies 5–7).

AMS interaction surpasses this limitation by spontaneously inducing three distinct collective behaviors in sequence: first, the swarm undergoes a contraction process away from the narrow exit; then, it aligns directly with the center of the narrow exit; and finally, it evacuates smoothly as a cohesive unit (Fig. 6c and Supplementary Movies 5–7). In particular, we observe that in order to spare space for swarm contraction, some individuals temporarily sacrifice their evacuation time to move towards the opposite direction of the exit (highlighted by red boxes in Supplementary Fig. 42). To gain a deeper understanding of self-organization of AMS, we focus on comparing the dot product between desired alignment heading $\hat{\mathbf{v}}_{\text{al},i}$ and real moving heading $\mathbf{v}_i$. Note that the other four parts of velocity components except $\hat{\mathbf{v}}_{\text{al},i}$ remain consistent across AMS, ATHD, and average interaction in the evacuation experiments (see "Methods" for details). After analyzing the results shown in Fig. 6a–c, we find that the dot product between $\mathbf{v}_i(t)$ and $\hat{\mathbf{v}}_{\text{al},i}(t)$ in descending order is AMS, ATHD and Average, and AMS has the least variance of dot product (Supplementary Fig. 43). It demonstrates that throughout the evacuation process, AMS, compared to other interactions, enables each individual to achieve maximum consistency between aligning with neighbors ($\hat{\mathbf{v}}_{\text{al},i}$) and following the actual direction ($\mathbf{v}_i$, resulted from various types of interactions with surrounding environments and neighbors) of movement. Given that all individuals are potentially guided towards the exit, if each individual maintains maximum alignment consensus with neighbors and the actual direction of movement, the swarm should be well-organized to efficiently evacuate through the narrow exit. Moreover, we also test the correlation between $w_{ij}(t,\tau_{\text{pre}})$ and $k_{ij}(t,\tau_{\text{next}})$ for the swarm trajectories generated by three interactions (Supplementary Fig. 43). The results show that only AMS indicates the prevalence of positive correlation between $w_{ij}(t,\tau_{\text{pre}})$ and $k_{ij}(t,\tau_{\text{next}})$, whereas we could not observe this in the other two interactions.

Therefore, for spending time of swarm evacuation, AMS interaction shows tremendous advantages over average and ATHD interaction. For instance, AMS interaction with 100% MS shortens the evacuation time by about 18% compared to the average interaction under three different gate widths (Fig. 6d, e). To fully understand the advantage, we record the spending time as a function of the percent of the swarm successfully evacuating the exit under different interaction types (Fig. 6f, g) and different swarm sizes (Supplementary Fig. 44). For AMS, the evacuation time increases almost linearly with the percent of the swarm that succeeds in exiting. Conversely, the average and ATHD interaction divide the evacuation process into two stages: the first 80% of the swarm spends the time as the same of AMS interaction, but the last 20% evacuate very slowly. The findings align with the observed phenomena in the experiments: when employing average or ATHD interaction, robots positioned directly in front of the exit could evacuate smoothly, whereas robots situated on either side of the swarm tend to rush straightly towards the sides of the exit, thereby leading to congestion. Interestingly, AMS interaction could induce the swarm to spontaneously emerge from the contraction process to directly face the exit and then evacuate as a whole.

The spontaneous emergence of swarm contraction is also validated by the statistics of historical trajectories of all individuals from 10 independent experiments (Fig. 6h) and 50 simulations (Fig. 6i). (See Supplementary Fig. 45 for statistics of historical trajectories of other exit widths). Furthermore, different parameters, i.e., swarm size and the top $x\%$ MS used in AMS interaction, are systematically investigated to verify the generalization of AMS interaction in collective evacuation (Supplementary Fig. 46).

## Discussion

The research on collective behaviors of animal flocks and swarm robotics is increasingly dependent and mutually supportive[59–61]. The extraction of self-organization principles from animal groups is essential for the development of bio-inspired distributed control strategies in swarm robots. Conversely, these swarm robots serve as crucial test-beds for validating the effectiveness and expanding the potential applications of self-organization for collective tasks. Despite extensive exploration of these three aspects, either independently or as integral components of the research chain, a comprehensive research continuum spanning from biological observation to bio-inspired mechanisms and onward to swarm robotics is still lacking.

Our research endeavors to establish a comprehensive and interdisciplinary framework that involves the exploration of interactions governing self-organization in collective motions, and the subsequent application of these principles to swarm robotics for collective tasks. In particular, this study utilizes three large bird-flocking datasets encompassing diverse ecological context and motion patterns, namely mobbing, circling, and transit. Given that the emergence of a leading relation is intrinsically connected to the adjustment of movement upon perceiving neighbors' motions, we propose a heuristic measure called perception of neighbors' motion salience from the individual's first-person view, and systematically investigate the relationships between MS and LF. At flock level, the correlation between MS and LF indicates that individuals with higher MS tend to lead the group in the maneuver motions. Delving deeper into the individual level, the correlation between the past perception of MS and future change rate of velocity consensus vividly suggests that individuals will accelerate the convergence of velocity with neighbors who have higher MS. Hence, the empirical evidence strongly supports the justification for incorporating the adaptive MS-based interaction observed in natural flocks into swarm models.

We acknowledge collective behaviors (e.g., mobbing, roosting, or transit) in relation to the ecological environment and physiological characteristics of the study species. However, our primary focus of this study is to elucidate the fundamental mechanisms that shape the emergence of collective motions and subsequently translate them into a bio-inspired swarm model, supported by explicit empirical evidence. Regrettably, due to our limited expertise in ecology and animal behavior, we are unable to thoroughly consider the intricate ecological context in our analysis. For example, in the mobbing and transit datasets[15], Jackdaw (*Corvus monedula*) consistently forms long-term monogamous relationships, with both parents contributing to rearing the young. Consequently, the large number of individuals recorded in these datasets encompasses mated pairs, unpaired individuals, and juveniles. We openly admit that our analysis overlooks the influence of pair-bonded relationships on quantifying MS, however, these relationships can notably impact MS measurements. We anticipate that future research endeavors will address this gap, offering a more nuanced understanding of MS within the ecological and physiological attributes of the study species.

Besides, we believe that one of the advantages of MS eliminates the necessity for an exhaustive depiction of the entirety of collective motion processes. This is because MS only necessitates the relative position of the initiation and termination time stamps during a given interval to measure the relative alterations in motion between a pair of birds. As a result, MS offers a versatile approach that can be applied to different types of collective tasks, making it highly adaptable and

generic. Meanwhile, MS-based adaptive interaction could be readily integrated into diverse modeling frameworks for collective tasks, primarily requiring incorporation of the alignment term. For instance, this approach is implemented in a swarm of miniature robots to conduct collective evacuation experiments, demonstrating notable improvements in the self-organization of the swarm facilitated by AMS. The swarm exhibits a spontaneous emergence of three distinct collective behaviors in succession: initially undergo a contraction process far from the narrow exit, then directly face the center of narrow exit, and finally evacuate as a whole. The entire evacuation process effectively prevents the congestion on both sides of the narrow gate. Moreover, we also test the performance of AMS in another type of swarm experiment known as collective following, which simulates the collective behaviors observed during foraging activities[62,63]. As implied by its name, collective following presupposes the presence of an informed individual guiding the group towards multiple destinations that frequently vary during foraging activities. The results demonstrate that AMS enables the swarm to promptly respond to the transient perturbation, thereby sustaining a high level of collective responsiveness within the group (see details in Supplementary Note 9 and Supplementary Movies 9–11). We believe that MS-based adaptive interaction will have a positive translational impact on the deployment of more advanced autonomous swarm robots[64] and more sophisticated collective tasks[65–67].

While the MS-based adaptive interaction presents a promising avenue for integrating bionic mechanisms into bio-inspired swarm robotics, we acknowledge that significant efforts are still needed. These efforts include advancing beyond the current position-based measurement of MS to accurately reconstructing the retina-based individual perception[34,68,69], as well as bridging the disparity between mathematical modeling of motion salience and perceptual devices on robot[70,71], etc.

Overall, this study underscores the significance of interdisciplinary integration in advancing the exploration of collective behavior and illustrates a promising pathway for successfully translating bio-inspired mechanisms observed in natural flocks into the realm of swarm robotics. Particularly noteworthy are the impressive outcomes of self-organization observed in the swarm evacuation experiments, emphasizing the pivotal role of swarm robotics as a testing ground. These findings not only affirm the efficacy of bio-inspired mechanisms but also broaden their scope for applications in collective tasks, facilitating interdisciplinary collaboration across the domains of biology, physics, engineering, and allied disciplines.

## Methods
### Constructing the leader-follower relation matrix of a flock
For a pair of individuals in a flock with the recording time from $t_s$ to $t_e$ (Fig. 1a), we could compute the temporal degree of motion alignment between the focal one's flying direction ($\hat{\mathbf{v}}_i$) and those of neighbors ($\hat{\mathbf{v}}_j$) in the form of,

$$C_{ij}(t,\tau) = \hat{\mathbf{v}}_i(t) \cdot \hat{\mathbf{v}}_j(t+\tau) \, (i \neq j, j \in \mathcal{S}_i). \tag{2}$$

Here $t$ is the time stamp, $\tau$ is the time interval belonging to $(t_s - t_e, t_e - t_s)$, and $\mathcal{S}_i$ indicates the collection of neighbors of the focal $i$. Hereinafter, the vector symbols with hat (i.e., $\hat{\mathbf{v}}_i$) means the normalized vector. Note that due to the limitation of recording area in the original observation studies, we roughly assume that, for a bird flock from three dataset, $\mathcal{S}_i$ represents all individuals appeared in the flock when calculating $C_{ij}(t,\tau)$.

For a given $\tau$, the average of $C_{ij}(t,\tau)$ over different time stamps within the period $[t_s, t_e]$ is denoted as $\langle C_{ij} \rangle(\tau)$. Thus, we collect the curve of $\langle C_{ij} \rangle(\tau)$ as a function of $\tau$. $\tau_{ij}^{\text{LF}}$ is the value to make the curve of $\langle C_{ij} \rangle(\tau)$ reach the maximal value. We disregard $\tau_{ij}^{\text{LF}}$ if it locates at first 25% or last 25% of $\tau$-axis, as we consider this $\tau$ to be too short for directional

copying[72]. Meanwhile, $\tau_{ij}^{\text{LF}}$ is not considered if the maximal $\langle C_{ij} \rangle(\tau)$ is not larger than 0.8. See Supplementary Fig. 16 for details. Note that $\langle C_{ij} \rangle(\tau) \neq \langle C_{ji} \rangle(\tau)$ and $\tau_{ij}^{\text{LF}} \neq \tau_{ji}^{\text{LF}}$, because Eq. (2) does not satisfy the exchange law for a pair of individuals.

Besides, within the circling dataset, we observe the assemblage of numerous swifts hovering over the roost actually comprises multiple sub-flocks. We find that LF relation matrix could successfully classify the circling flocks into sub-communities with more similar motion patterns (see Supplementary Note 2 and Supplementary Fig. 10 for an example).

### Quantifying the leading tier of each individual from LF networks
The leading tier of each individual from an LF network of a flock within $[t - \tau, t]$ could be calculated by the local reaching centrality of an unweighted directed graph[42] as following,

$$L_i = \frac{1}{N-1} \sum_{j \in \mathcal{S}_i} \frac{1}{d_{ij}^{\text{out}}}, \tag{3}$$

where $N$ is the number of nodes, $d_{ij}^{\text{out}}$ means the out-distance from node $i$ to node $j$, and $\mathcal{S}_i = \{j \in \{1, \cdots, N\} | 0 \leq d_{ij}^{\text{out}} < \infty\}$ indicates the set of nodes with finite out-distance from node $i$. For example, in an unweighted directed graph, if node $i$ without out-degree edges must locate at the bottom layer, then $d_{ij}^{\text{out}} = \infty$ and $L_i = 0$; if node $i$ with 1-step out-degree edges to the rest nodes must locate at the top layer, then $d_{ij}^{\text{out}} = 1$ and $L_i = 1$. Therefore, $L_i$ implies the hierarchical layer of each node in a directed network. For a flock within the time period $[t - \tau, t]$, we first calculate the LF network, and consequently yield the leading tier of each individual $L_i(t,\tau)$ by Eq. (3). Note that we set the non-zero elements in LF relation matrix as 1 when calculating $L_i(t,\tau)$.

### The adaptive MS-based interactions in swarm model
The prevalence of positive correlations between $w_{ij}(t,\tau_{\text{pre}})$ and $k_{ij}(t,\tau_{\text{next}})$ inspires us to introduce the adaptive MS-based interaction (AMS) to the classical self-propelled swarm model[17]. In the model, each agent moves towards a heading with a constant speed $v_0$. The position of agent $i$ is updated as

$$\mathbf{x}_i(t+1) = \mathbf{x}_i(t) + v_0 \hat{\mathbf{v}}_i(t) \Delta t. \tag{4}$$

Hereinafter, $\hat{\mathbf{v}}_i(t) = \mathbf{v}_i(t)/||\mathbf{v}_i(t)||$ is the normalized velocity of agent $i$, and $\mathbf{v}_i(t)$ is updated as

$$\mathbf{v}_i(t+1) = \hat{\mathbf{v}}_i(t) + \sum_{j \in \mathcal{S}_i} w_{ij}(t) \hat{\mathbf{v}}_j(t). \tag{5}$$

Here, $\mathcal{S}_i$ represents the neighboring agents (except $i$ itself) within a circle of sensing radius $r_{\text{al}}$ that is centered at agent $i$. $w_{ij}$ is the weighted coefficient to reflect the heterogeneity of local interactions between a pair of individuals, which indicates the influences from neighbors exerted on the focal individual. According to the findings in bird flocks, AMS assumes that the neighbor with higher MS could impose much more influences on the focal individual. Thus, $w_{ij}$, as the coefficient of adaptive interaction, could be quantified by MS in the form of,

$$w_{ij}(t) = \frac{M_{ij}(t,\tau)}{\sum M_{ij}(t,\tau)} \text{ for } j \in \mathcal{S}_i, \tag{6}$$

where $M_{ij}(t,\tau)$ calculated by Eq. (1) equals the MS of neighbor $j$ perceived by agent $i$ within the period $[t - \tau, t]$.

Furthermore, to further investigate the effects of paying more attention to neighbors with larger MS in AMS, we introduce a tunable percent parameter, denoted as $x\%$, to select the neighbors from $\mathcal{S}_i$

according to their MS values. For the focal agent $i$, (i) getting the metric-based neighbor set $\mathcal{S}_i$ by the given distance criterion $r_{al}$ and calculating $w_{ij}$ for $j \in \mathcal{S}_i$ based on Eq. (6); (ii) picking up a part of neighbors as a new set $\mathcal{S}_i^{x\%}$ if the cumulative sum of $w_{ij}$ in a descending order is just larger than $x\%$; (iii) renormalizing $w_{ij} = \frac{w_{ij}}{\sum w_{ij}}$ for $j \in \mathcal{S}_i^{x\%}$ and setting $w_{ij} = 0$ for $j \notin \mathcal{S}_i^{x\%}$. Here AMS-$x\%$ means at least the top $x\%$ of MS are involved in AMS. By this definition, the tunable $x\%$ could even further amplify the attention effect of neighbors with higher MS. For instance, if $x\% = 50\%$, AMS-50% makes the focal agent exclusively align with the least neighbors cumulatively possessing the top 50% MS, disregarding any influences from neighbors with the latter 50% MS.

## The simulation of swarm model

For comparison purpose, we also consider two MS-free interactions. One is a kind of adaptive interaction, called adaptive interaction based on transient heading difference[55,56] (ATHD). Unlike perceiving MS within a time period, ATHD could adaptively make the neighbors with larger (or smaller) transient heading difference exert larger (or smaller) influences on the focal individual, such as, $w_{ij}(t) = \frac{\sin(\Delta_{ij}(t))}{\sum_{j \in \mathcal{S}_i} \sin(\Delta_{ij}(t))}$ with $\Delta_{ij}(t) = \min\{|\theta_i(t) - \theta_j(t)|, 2\pi - |\theta_i(t) - \theta_j(t)|\} \in [0, \pi]$. $\theta_i(t) \in [-\pi, \pi]$ denotes the heading of agent-$i$ at time $t$. This interaction strength adaptively increases with the increment of heading differences between neighbors and the focal individual. Once the difference is larger enough, i.e., larger than $\pi/2$ in ATHD, the contribution will decrease and eventually will become zero. Another MS-free interaction is the commonly used average interaction, where $w_{ij} = 1$ for $j \in \mathcal{S}_i$, leading the adaptive interaction to degenerate to the standard Vicsek model[17]. This implies that the focal agent interacts equally with neighbors within its sensing radius.

In Fig. 5, we use a classical self-propelled particle model to compare the above different interaction rules as following[15,73], $\mathbf{v}_i(t+1) = v_0 \boldsymbol{\Omega}_\eta (\boldsymbol{\Theta}[\sum_{j \in \mathcal{S}_i} w_{ij}(t)\mathbf{v}_j(t) - \beta_i\mathbf{x}_i(t)])$. In this model, all particles move with a constant speed $v_0$ and align their directions of motion based on the local interaction rules, with some noise added. The operator $\boldsymbol{\Omega}_\eta$ imposes the noise through rotating the vector by a random angle chosen from a uniform distribution with maximum amplitude of $\eta$, and operator $\boldsymbol{\Theta}$ normalizes the argument to be a unit vector. Note that the term $-\beta_i\mathbf{x}_i(t)$ is introduced as additional potential well that pushes particle $i$ back towards the origin[73]. Here to mimic the leading effect, the potential well is only imposed on individual-1 ($\beta_1 \neq 0$, red trajectories in Fig. 5a–c) to lead the flock come back to origin, and $\beta_i = 0$ for other particles. We run this model in 3D without boundary conditions. Initially, particles are randomly distributed in a 3D sphere with a radius of $R = 50$ m and move in random directions with constant speed $v_0 = 10$ m/s. The parameters used in Fig. 5 are $N = 30, r_{al} = 30$ m, $\Delta t = 0.05$ s, $\eta = 0.05$, and $\beta_1 = 0.08$. We set the perceiving time $\tau = 20\Delta t$ in AMS.

## Swarm robotics system

To demonstrate the implication of AMS to enhance the self-organization of real swarming robots, we build a swarm robotics system that consists of $\sim 10^2$ miniature two-wheel differential mobile robots in a motion capture arena (Supplementary Fig. 40). Each robot is equipped with two stepper motors with reduction gears, a PCB board (with STM32F103RC8T6 microcontroller) for motion control and power management, a PCB board (with STM32F103RCT6 microcontroller and NRF24L01 wireless communication module) for communication and decision-making, and a marker deck at the top for hosting passive infrared reflective balls (4–5 balls with diameter of 10 mm) for localization (Supplementary Fig. 40a). The diameter of robot's main body is 60 mm and the diameter of marker deck at the top is 84 mm. The robot, powered by two 3.7 V rechargeable lithium batteries (2*800 mAh), can move according to specified linear and angular speed control commands. A NOKOV motion capture system is used to track the robot position $\mathbf{x}(t)$ (center of body) and heading $\theta(t)$, from which real-time linear speed $v(t)$ and angular rate $\omega(t)$ of the robot are obtained by differential calculation. For simplicity, all motion control commands for swarming robots in our experiments are computed on a service computer and then simultaneously broadcasted to the swarm with a fixed time interval $\Delta t$ through a customized wireless communication protocol. After receiving the motion command of the desired linear speed $v_d(t+1)$ and heading $\theta_d(t+1)$ calculated from the swarm model, each robot performs the angular rate command $\omega = \min\{\Delta\theta/\Delta t, \omega_{max}\}$ within a control loop, where $\Delta\theta = \theta_d(t+1) - \theta(t)$ and $\omega_{max}$ is the maximum angular rate. To guarantee the transferability of the swarm model to hardware experiments, the system is capable of sending motion control commands as fast as 20 Hz (i.e., support $\Delta t \geq 0.05$ s) and capturing each robot's motion states up to 300 Hz. We take the maximum angular rate $\omega_{max} = 19.1$ deg/s in the swarm experiments. See Supplementary Fig. 40c for the architecture of swarm robotics system.

Due to the limitation of arena size, we perform the swarm experiments with up to 50 robots (Supplementary Fig. 40b). However, to perform the swarm experiments with hundreds of robots, we transfer the real robots to semi-physical simulation with the same motion characteristic in Pybullet (Supplementary Fig. 41c and Supplementary Movie 8).

## Swarm model of collective evacuation

The swarm model of collective evacuation in this work is inspired by a state-of-the-art framework that enables drone swarm to navigate successfully in confined and cluttered environments[8,57]. Adopting from the original framework, it includes 5 parts for agent-$i$:
(i)  alignment with neighbors in the sensing radius, $\hat{\mathbf{v}}_{al,i}$;
(ii)  repulsion among the near neighbors, $\hat{\mathbf{v}}_{rep,i}^a$;
(iii)  inter-agent collision avoidance, $\hat{\mathbf{v}}_{col,i}^a$;
(iv)  agent-wall collision avoidance, $\hat{\mathbf{v}}_{col,i}^w$;
(v)  guidance velocity for agent to pass through the exit, $\hat{\mathbf{v}}_{g,i}$.

Thus, in any instant, the velocity for agent-$i$ resulting from the contributions above is

$$\mathbf{v}_i(t + \Delta t) = (1 - \delta_t)(\hat{\mathbf{v}}_{al,i} + \hat{\mathbf{v}}_{rep,i}^a) + \hat{\mathbf{v}}_{g,i} + \delta_t(\hat{\mathbf{v}}_{col,i}^a + \hat{\mathbf{v}}_{col,i}^w), \tag{7}$$

where $\delta_t = 1$ if the collision avoidance of either inter-agent or agent-wall is active at time $t$. Otherwise, $\delta_t = 0$. Once $\delta_t = 1$, in order to prevent the inter-agent or agent-wall collision avoidance, we reduce the normal speed $v_0$ to a very low level $v_{col}$. All the parameters used in collective evacuation experiments and simulations are listed in Supplementary Table 1.

The alignment exerted on agent-$i$ is the weighted average of neighbor-$j$'s velocity,

$$\mathbf{v}_{al,i} = \hat{\mathbf{v}}_i + \sum_{j \in \mathcal{S}_{al,i}} w_{ij}\hat{\mathbf{v}}_j \quad \text{if } d_{ij} < r_{al}. \tag{8}$$

$w_{ij}$, calculated by Eq. (6), denotes the MS-based adaptive interaction. If $w_{ij} = 1$, it reduces to the average interaction.

$\hat{\mathbf{v}}_{rep,i}^a$ represents the repulsion among inter-agent to push them farther apart when the neighbors are closer than the pre-defined distance $d_{rep}^a$. The repulsion term pushes the near neighbor $j$ farther apart as below,

$$\mathbf{v}_{rep,ij}^a = \begin{cases} \left(d_{rep}^a - d_{ij}\right)\frac{\mathbf{x}_i - \mathbf{x}_j}{d_{ij}} & \text{if } d_{ij} < d_{rep}^a \\ \mathbf{0} & \text{otherwise} \end{cases}. \tag{9}$$

The total repulsion calculated for agent $i$ with respect to the repulsive set $\mathcal{S}_{\mathrm{rep},i} = \{j | d_{ij} < d^a_{\mathrm{rep}}\}$ is $\mathbf{v}^a_{\mathrm{rep},i} = \sum_{j \in \mathcal{S}_{\mathrm{rep},i}} \mathbf{v}^a_{\mathrm{rep},ij}$.

The inter-agent collision avoidance exerted on agent $i$ from neighbor $j$ is similar with repulsion of Eq. (9), while the threshold of inter-agent distance $d^a_{\mathrm{col}}$ to trigger the collision avoidance is smaller than that of $d^a_{\mathrm{rep}}$ and we assume it only considers the neighbors are standing in the front view within $[-\pi/2, \pi/2]$, i.e.,

$$\mathbf{v}^a_{\mathrm{col},ij} = \begin{cases} \left(d^a_{\mathrm{col}} - d_{ij}\right)\dfrac{\mathbf{x}_i - \mathbf{x}_j}{d_{ij}} & \text{if } d_{ij} < d^a_{\mathrm{col}} \text{ and } \angle(\hat{\mathbf{v}}_i, \mathbf{x}_j - \mathbf{x}_i) \in [-\pi/2, \pi/2] \\ \mathbf{0} & \text{otherwise} \end{cases}. \quad (10)$$

Here $\angle(\hat{\mathbf{v}}_i, \mathbf{x}_j - \mathbf{x}_i)$ represents the relative angle between neighbor-$j$'s position and agent-$i$'s moving direction. Thus, $\mathbf{v}^a_{\mathrm{col},i} = \sum \mathbf{v}^a_{\mathrm{col},ij}$ for $j \in \mathcal{S}^a_{\mathrm{col},i} = \{j | d_{ij} < d^a_{\mathrm{col}} \text{ and } \angle(\hat{\mathbf{v}}_i, \mathbf{x}_j - \mathbf{x}_i) \in [-\pi/2, \pi/2]\}$.

Besides, as the walls act the role of border in the experiments, we assume the agent-wall collision avoidance only responds to the nearest wall located in agent $i$'s front view:

$$\mathbf{v}^w_{\mathrm{col},i} = \begin{cases} \mathbf{x}_i - \mathbf{x}^w_i & \text{if } d^w_i < d^w_{\mathrm{col}} \text{ and } \angle(\hat{\mathbf{v}}_i, \mathbf{x}^w_i - \mathbf{x}_i) \in \left[-\frac{\pi}{2}, \frac{\pi}{2}\right] \\ \mathbf{0} & \text{otherwise} \end{cases}. \quad (11)$$

Here $d^w_{\mathrm{col}}$ is distance of agent-wall collision avoidance, $d^w_i$ is distance between agent $i$ and the nearest wall, and $\mathbf{x}^w_i$ indicates position of wall that are the nearest to agent $i$. Note that for simplification in the calculations we assume the walls are consisted of many virtual cylinders with diameter of 50 mm (Supplementary Fig. 41b).

Finally, to ensure the goal-oriented swarm evacuation in a confined environment, we assume that each virtual cylinder in the wall could indicate the exit direction according to their location (Supplementary Fig. 41b). Therefore, we consider that agent-$i$ could receive the moving guidance from the walls that are within a circle of radius $r_g$ centered at agent-$i$,

$$\mathbf{v}_{g,i} = \begin{cases} \sum_{k \mathcal{S}^w_i} \hat{\mathbf{v}}^w_k & \text{if } d^w_{ik} < r_g \\ \mathbf{0} & \text{otherwise} \end{cases}. \quad (12)$$

Here $d^w_{ik}$ means the distance between agent $i$ and wall $k$, $\mathcal{S}^w_i = \{k | d^w_{ik} < r_g\}$ is the collection of walls that located in a circle of radius $r_g$ centered at agent-$i$, and $\hat{\mathbf{v}}^w_k$ represents the guidance direction of wall $k$. The meaning of Eq. (12) is similar with the case that agent walks along the wall to the exit if it is close to wall.

## Reporting summary

Further information on research design is available in the Nature Portfolio Reporting Summary linked to this article.

## Data availability

The mobbing and transit datasets from ref. 15 have been deposited in https://figshare.com/s/472d354cc9e823a8f48f. The circling dataset from ref. 16 is available at https://doi.org/10.5061/dryad.p68f8.

## Code availability

All source codes and minimal dataset related to the work can be found at[74] https://github.com/xiaoyandong08/Perception_of_Motion_Salience.

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

## Acknowledgements

We thank Professors Nicholas T. Ouellette and Dennis J. Evangelista for kindly sharing the flocking datasets. Y.D.X. is supported by the National Natural Science Foundation of China (61902418). X.P. is supported by the National Natural Science Foundation of China (62076203).

## Author contributions

Y.D.X. conceived the project. Y.D.X. designed and managed the whole project. Y.D.X. performed all the real data analysis, analytical/numerical calculations, and simulations. Y.D.X. designed the collective evacuation experiment. X.L. designed the collective following experiment. X.P., X.L., Z.Z., and Y.L.X. built the swarming robotic system. X.L., Y.L.X., and Z.Z. performed swarm experiments. Y.D.X. wrote the manuscript. Y.-Y.L. edited the manuscript.

## Competing interests

The authors declare no competing interests.
