## [Peer Review File · Nature Communications]

REVIEWER COMMENTS

Reviewer #1 (Remarks to the Author):

The manuscript deals with the emergence of collective motion patterns in groups of interacting individuals (e.g., birds, robots). It sets the ambitious objective to study the implications at the collective level of a proposed local interaction mechanism from observational data analysis (in the “Empirical data analysis” section) to model-based theory (“A swarm model with adaptive MS-based interactions”) and robot experiments (“Application to swarm robotics”). In doing so, the authors aim to cover the full “research chain” from observation to experimental swarm robotics, and achieve “system-level understanding” of the proposed mechanism.

The diversity of methodologies adopted in the paper is one of its main strengths, in my view, but it also makes it difficult to summarize the key contributions of the paper. To my current understanding, the main contribution is the introduction of a new local variable, the perception of motion salience (MS), which, according to the authors, (1) correlates with individuals’ leadership status and degree of individual freedom within observed bird flocks, (2) can be used to explain the observed nested structure of leader-follower matrices, and (3) to design effective robot swarms that solve effectively specific collective motion tasks such as evacuation.

Overall, this manuscript is the result of an impressive effort to study the collective implications of a local interaction mechanism from observational data to models and experiments, which is rare in a single paper. The paper could have impact on the collective behavior field, by highlighting the importance of covering the full research cycle from observation to validation through actual implementation. At the same time, I have major concerns about some of the methodologies adopted by the paper, and the novelty and strength of the findings (especially in relation to the observational results). Below, I summarize my concerns and present some suggestions that may be helpful to determine which claims are actually supported by the data and models, stress their novelty, and extensively revise this work as a result.

Major comments

(A) Methodology

A1. Nestedness.

The authors build the whole nestedness analysis on the raw NODF values. They compare the nestedness of different empirical matrices by directly comparing the NODF values. They assume that “the NODF normalizes the matrix size, and thus allows matrices of different sizes to be compared” (Supplementary Note 4).

While it’s true that the NODF has a size-dependent normalisation factor, unfortunately in practice, it cannot be used to compare matrices of different sizes and densities.

The NODF metric is indeed very sensitive to the input matrix’s size and density, as proven by various works (e.g., Ulrich et al., 2009; Payrato-Borras et al., 2020). In light of this major issue, while some of the nestedness-related claims seem to hold (e.g., by visually looking into the matrices in fig. 1), the current quantitative procedure is not correct.

One related issue is the absence of comparisons against null models. The authors can refer to Gotelli et al. 2012; Mariani et al., 2019; Molina and Stone, 2020 for extensive discussions of the essential role of null models in inferring the significance of structural patterns in network analysis. Null model analysis enables a researcher to establish whether an empirically observed structural metric (e.g., nestedness) is significantly higher than that observed in suitably randomized networks that preserve some chosen properties of the empirical one (see also Staniczenko et al, 2013 for their considerations on the risks of a too conservative null model).

I have two suggestions for the authors, to attempt to validate their claim that certain matrices are more nested than others. One is to perform comparisons against null models of various degree of constraint (see Staniczenko et al, 2013), and establish that some (more nested) matrices exhibit higher discrepancies with respect to their randomized counterparts (e.g., higher z scores) than the less nested ones. This analysis could be done with different nestedness metrics, such as the NODF and spectral radius, to show that the results are independent of the choice of the nestedness metric.

Another suggestion is to prioritize nestedness metrics that are known to have weaker dependencies on matrix size and density. According to the most recent systematic comparison of nestedness metrics (Payrato-Borras et al., 2020), the ATC and, to a lesser degree, the spectral radius would be better candidates than the (more popular) NODF.

A2. MS and velocity.

According to its definition, an element of the MS quantifies the perceived movement of an individual perceived by another individual within a time window. I wonder how well the MS

correlates with a bird's radial and angular velocity. If the correlation is very high, the MS-LF correlation found by the author could be a manifestation of the well-known property that faster birds tend to occupy leadership positions in observed groups (Krause et al., 2000; Pettit et al., 2015).

See (B) below for related comments, on the lack of critical reflection of the manuscript with respect to existing literature on leadership.

A3. Mechanism behind observational results.

The manuscript falls short of providing possible reasons why the MS correlates with leadership. The authors proceed by incorporating MS-based mechanisms into the model-based and experimental analyses, but do not discuss why the MS is to be expected to correlate with leadership. Is it already implied by the definition?

If not, one possible explanation is that faster birds have both high MS and leadership. But this would be a confounding factor coming from an already-known result from the literature.

What are alternative explanations?

A related concern: isn't the explanation in the paragraph starting with "The above phenomena in different patterns..." circular? In flocks with maneuverable motions, individuals have higher radial and angular speeds, which can result in higher MST. What does the MST add to the intuition that individuals in flocks with maneuverable motions are just faster?

To address these concerns, it would be beneficial if the authors could present in detail the correlations between the MST, individual-level radial and angular velocities, and clarify whether the MST can generate **new** observational insight beyond those that can be generated by only looking at the birds' velocity.

A4. Individual freedom.

The authors seem to propose that MST quantifies an individual's freedom. However, I do not understand which result, if any, proves this very general claim. It seems more of a consequence of the definition of MST and its interpretation by the authors.

First, is "freedom" precisely defined? Besides, T is the time at which a pair's relative motion achieves its maximum, but the mere existence of the maximum does not automatically imply that there is less or no "freedom" for longer timescales. Perhaps, it could help to study carefully the actual shapes of M_{ij} as a function of time, especially around the maximum and in the long- τ limit. Finally, similar to my comment A3, it would be helpful to see correlations between the MST and individual velocities.

Because of these ambiguities, I do not think that the following claim is supported by the current evidence: "the above phenomena [...] demonstrate that the smaller the average MST, the smaller the individual freedom".

(B) Strength and significance of results

B1. Novelty of observational results.

As acknowledged by the authors, the manuscript is not the first one to study leader-follower relationship and, most importantly, correlates of leadership status in animal groups. However, the manuscript falls short at comparing its results from those already known from this literature. For example:

— Rosenthal et al. (2015) found how an individual position in the communication network correlates with its influence in the group.

— Previous studies related an individual's speed with its leadership status in the group (Krause et al., 2000; Pettit et al., 2015).

— Other properties correlated with leadership include nutritional requirements, physiological traits, spatial position within the group. The authors could refer to Jolles et al., 2020 for a recent review.

How do the observational results relate with these findings? Critical reflection with respect to this literature is needed to ensure the novelty of the results. For example, if one controlled for known properties in the literature (e.g., speed), would the association between MST and the leadership status remain significant?

B2. Strength of experimental results.

In the collective following experiments, the authors use R to measure the collective response performance of robot swarms under different local mechanisms. However looking at figures 5g-h, it seems unclear that the AMS brings an advantage compared to the ATHD.

The collective evacuation experiment seems to be the strongest result in the manuscript, to my current understanding, because here, the authors are able to show an actual performance improvement of the AMS compared to other methods. I wonder if this point could be strengthened in the manuscript, perhaps by providing more specific insights on the practical implications of this result.

Admittedly, while I could follow the logic and results of the experiments, I was unable to fully assess its technical details because of my lack of technical expertise on robot experiments.

(C) Clarity of exposition.

C1. Sentences implying intentionality and causality.

There are several sentences that imply that the system or its actors make intentional decisions to achieve some objective, and that the AMS has causal impact on some variables.

Examples include, “flocks with higher maneuverable motions (e.g., mobbing and circling) *prefer* to have a more nested structure of LF relations and a clear hierarchy to maintain group cohesion”; “individuals with higher MS *prefer* to lead the group”; “tactful *strategy to organize* the group”.

However, such intentional claims seem not supported by the purely correlational nature of the observational analysis performed by the authors. Because of this, I would recommend watering down “intentional statements” and replacing with correlational statements (e.g., “prefer to have” becomes “tend to exhibit”), unless stronger arguments can be provided by the authors.

C2. General claims.

Some of the claims in the introduction and discussion are very general, but the level of presented methods and evidence not strong enough to fully support them.

Examples include:

— “perception of MS may have major adaptive and evolutionary significance to balance the trade-off between individual freedom and group cohesion in collective motions”, which seems a consequence of the authors’ interpretation and not an actual result, as discussed above;

— “great advantages of AMS in shaping the emergence of collective motion”, which seems only supported in the specific evacuation experiments compared to the baselines adopted by the authors;

— “AMS not only empowers the swarm to promptly respond to the transient perturbation but also strengthens the self-organization of collective motions in terms of temporal cognition”, a claim for which I could not find a precise supporting result.

— “Our findings may also facilitate the design of fully autonomous swarm robotics with artificial neuronal circuits of see-sense-decision-direct, i.e., robots with insect brains”, but given the level of evidence, I’m unsure about which results will facilitate this design beyond what was previously possible.

C3. Readability.

I found the manuscript very difficult to read, overall. There are many complex sentences and paragraphs (e.g., in the introduction), unclear expressions (e. g. “intrinsic interactions”; “subtle mechanism of group organization”, why is the mechanism subtle?) and too general claims. A major revision to improve readability and water down hyperbolic claims is needed.

Minor issues:

— The fact that the leader-follower matrix is binarized for the nestedness analysis is quite important in light of Staniczenko et al, 2013, and it should be emphasized in the main text as well, not only in the SM.

— As the MS definition is central to the manuscript’s contribution, it would deserve to be placed in the main text, and thoroughly commented. The definition is provided in Eq. 2, but not extensively motivated against, for example, simpler alternatives.

References

Gotelli, N. J., & Ulrich, W. (2012). Statistical challenges in null model analysis. *Oikos*, 121(2), 171-180.

Jolles, J. W., King, A. J., & Killen, S. S. (2020). The role of individual heterogeneity in collective animal behaviour. *Trends in ecology & evolution*, 35(3), 278-291.

Krause, J., Hoare, D., Krause, S., Hemelrijk, C. K., & Rubenstein, D. I. (2000). Leadership in fish shoals. *Fish and Fisheries*, 1(1), 82-89.

Mariani, M. S., Ren, Z. M., Bascompte, J., & Tessone, C. J. (2019). Nestedness in complex networks: observation, emergence, and implications. *Physics Reports*, 813, 1-90.

Molina, C., & Stone, L. (2020). Difficulties in benchmarking ecological null models: an assessment of current methods. *Ecology*, 101(3), e02945.

Payrató-Borràs, C., Hernández, L., & Moreno, Y. (2020). Measuring nestedness: A comparative study of the performance of different metrics. *Ecology and Evolution*, 10(21), 11906-11921.

Pettit, B., Akos, Z., Vicsek, T., & Biro, D. (2015). Speed determines leadership and leadership determines learning during pigeon flocking. *Current Biology*, 25(23), 3132-3137.

Rosenthal, S. B., Twomey, C. R., Hartnett, A. T., Wu, H. S., & Couzin, I. D. (2015). Revealing the hidden networks of interaction in mobile animal groups allows prediction of complex behavioral contagion. *Proceedings of the National Academy of Sciences*, 112(15), 4690-4695.

Staniczenko, P., Kopp, J. C., & Allesina, S. (2013). The ghost of nestedness in ecological networks. *Nature communications*, 4(1), 1-6.

Ulrich, W., Almeida-Neto, M., & Gotelli, N. J. (2009). A consumer's guide to nestedness analysis. *Oikos*, 118(1), 3-17.

Reviewer #2 (Remarks to the Author):

The paper combines the analysis of empirical data of collective motion in bird flocks with a new measurement of interaction between group members, an agent-based model and a swarm robotics experiment; a very interesting combination that can be valuable for many fields of research. The authors present their framework, methods and results clearly, with particularly nice figures. However, given that such an interdisciplinary endeavour is challenging, I think that the authors need to improve in the biological relevance of their text, mostly through the terminology used in the manuscript. I further have some concerns about the statistical analysis of the empirical data and the simplicity of the agent-based model, after the introduction argumentation suggesting that the collective motion algorithms need to include a higher degree of realism. Overall I enjoyed the paper and I hope my comments will help to improve it and prepare it for a large interdisciplinary audience, please find them below.

Main comments:

- Lines 50-58: I agree with the authors that vision specifics and perceptions are not often taken into account, but I find this first part a bit unclear, perhaps too technical for the introduction (especially 56-58), and not very convincing for the need of a new approach. Given that a model should reduce the complexity of the system it aims to study, so some realism will be sacrificed in the process, perhaps the authors can revise this part focusing on what the problem is (e.g., why is something problematic for flocks with highly manoeuvrable motion?) instead of just describing the assumptions of existing models.

- Line 58-64: I am not sure I understand what the authors mean in their second part. There are models with realistic motion characteristics and their effect on the emergent properties of the group are of course very important. For example, in reference of the manuscript 27 individual variation in speed and accelerating attraction in pigeons gives the more wide flock formation and lead-follower relations, in [1] the banging behaviour of the birds while turning is found to underlie the agitation waves in starlings, and in [2] the shape and internal structure of fish and birds is explained by locomotion properties. So perhaps the authors emphasize more in physics-oriented Vicsek-like models that are used to inspire swarm robotics algorithms and are often quite simplistic in such characteristics?

- Following the previous comment, the authors do not include realistic motion characteristics in their model.

- Line 77-79: There is quite some variability in the field of collective animal behaviour about what leadership means. Given the interdisciplinary audience of the journal, I think the authors should first clarify that. Leadership in flocks usually refers to emergent leadership that is linked to the frontal positioning of individuals in the flock, not biased interactions that make some individuals more influential than others given for instance an underlying social structure (that exists in some species).

- Lines 80-81: This comment refers to the wording in many parts of the manuscript. The wording that the flocks 'prefer' is misleading (also in the abstract with 'prefer to evolve'), the flock as a collective is a self-organized system with no wishes. The authors should revise this throughout. Same applies for line 29: the flocks do not 'choose' a strategy or organize its members.

- Since the data are from stable camera frames, the author should mention in the main text the average duration of the tracks in each dataset (how much data on average per individual) and the sampling frequency. From figure 1a-c there seems to be duration of 10, 5, and 2 seconds respectively. Also, they should mention the average group size of the flocks and how much variation there is in group size across datasets.

- Since the duration of a 'flock' in transit seems from figure 1 to be much smaller than the others, can this be the reason behind the LF matrix not displaying any hierarchy?

- I find that the statistical analysis of the empirical data is not enough to provide support for the findings. Figure 1 shows quite some variation with flock id. The authors should thus test for the effect of group size and random effect of id (or track duration), for instance concerning the relation between average order and nestedness (lines 152-154)

- Lines 171- 172: The time-lag or perceiving time that the authors propose in the MS model, if I am not mistaken, is similar with what the models of [3] and [4] refer to as update or reaction time. The novelty of this work is that this varies per pair of interacting neighbours. Is this the case? I think this should anyway be clarified in the methods or discussion.

- Lines 181: I find the MST a very interesting measurement. I however disagree with the 'flying freedom' mentioning; it flags more the tolerance of the focal individual to not stay close to its flock mates or follow its own drive. I do think that replacing the word 'freedom' throughout the manuscript with individual drive or something equivalent will improve the clarity of the results.

- Lines 193: Isn't that an effect of the nature of the behaviour? If the threshold to react to difference in positioning is larger than what the birds do during the few seconds of transit in that dataset, wouldn't then MST be very small? Also for lines 210-212: the initial conditions can also play an important role, if the group is very ordered and with a common destination, individuals have the same drive. In mobbing and circling turns individuals need to stay together while individually deciding to turn, and since it is a fast phenomenon, they cannot instantaneously react to one another. Given that individuals need time to perceive their neighbours, if I understand correctly, if the neighbour doesn't move much, MST will be small, but if it does, for instance when turning, the MST will be biased having a minimum time that matches the cognitive and locomotion abilities of each individual. Perfect coordination and order is extremely difficult and expensive while flying.

For lines 213-221: I disagree with the interpretation of the authors here. The authors do not take into account the ecological context of each behaviour. Especially the sacrificing freedom statement is very anthropomorphic; as I mention above, foraging individuals know where they are going and they are already in an ordered flock. They don't have an individual drive to change their motion, and the density of the flock I would expect to be stable. During collective turns, the density and shape of the group changes, and thus individuals may need to turn to avoid collisions more often. This is a 'disadvantage', rather than individual freedom. The explanation of the flock maintaining manoeuvrable motion is thus confusing.

- Having said these the authors could measure group density and diffusion (such as global, mutual diffusion or stability of neighbours [5]) across datasets. They can also measure turning rate in the mobbing and circling flights and see whether that affects MST and order, given that it has been shown to affect collective motion dynamics [4]. If for instance the reshuffling rate is high or the group is denser, individuals have to avoid each other more.

- Lines 158-159: does the LF structure and hierarchy lead to high group order, or the hierarchy emerges because of the order?

- Lines 134-135: it will be helpful to repeat here which species these result cover, especially because of the comparison with pigeons.

- How does leadership links to relative positions in the group? This should be very clear in the results given previous research in the field. Especially in [6], they show the position of neighbours initiating each transit and mobbing turn. It will be very interesting to see whether the new approach matches or not the previous findings, in particular because it is the same dataset.

- Lines 242-243: what does the correlation analysis performed only in these two flocks first and then to the datasets?

- Figure 1, e and f: how are the axis of these plots ordered?

- Lines 278-279: I am not sure what this sentence means, the authors should clarify it.

- In the swarm application the authors exclude the blind area, while they showed earlier that any relation between MS and LF depends on a being > 0 . The authors should better support this decision and any effect of it.

- Is the model two dimensional? Mobbing flights are quite expanded in altitude as seen in the original paper that the data are from. What do the authors think the effect of this difference may be, since avoidance of neighbours is exaggerated in 2D models?

- The authors find that AMS can lead to the emergence of leadership, but they define a leader in their simulations and check its performance. Doesn't this show that AMS can facilitate leading with fewer number of informed individuals [7], rather than being responsible for the emergence of stable leader networks? Perhaps a discussion on this with previous findings on leadership in agent-based models would be useful.

- The failure of the average algorithm in the simulations may be due to the time that following robots are given to follow or not. Flocks in nature can continuously align with each other and slowly make the flock turn, but if the change is instantaneous or very fast, the group doesn't have time to respond. I think simulations with restricted angular velocity of the leader can clarify whether this is the case.

- Lines 351-353: it would be interesting for the self-organized mechanism behind this to be explained, since the rules do not change.

- The agent-based model has constant speed, something that is not true for many species (especially the ones with evidence of leadership, since differentiation in speed is always enough for emergent leadership to appear). Can the authors run simulations with variable speed or measure the speed variation in the bird trajectories to check that there isn't strong variation there? Especially

when it comes to turning and changing altitude, acceleration can have a large effect on individual interactions.

- Lines: 399-403: the flocks cannot really evolve a mechanism neither individual 'preference' can be measured. Higher MS will make an individual less influenced by others, meaning it will follow its own tendency more and perhaps end up leading the group, which is a very interesting self-organized mechanism that I don't think it is given enough importance from the wording of the paragraph. If the authors wish to discuss the adaptive reason behind different behaviour between mobbing, roosting, and transiting, they should link their discussion better with the ecological context and the specific characteristics of their study species.

- The discussion doesn't include the empirical results on bird flocks or mention any effect of across species differences, even though the authors mention in their introduction that species have been found to differ in their interactions with plasticity even found in one of the datasets (ref [15] in the manuscript).

- Lines 394-395: I am not sure if MS is an entry point for movement changes of neighbours but rather their timing. For instance, there are many established measurements on changes in neighbour relative positions [4,5]. Some argumentation of whether the new measurement can be complimentary or replace the previous ones will improve the impact of the paper.

- The font size of many figures should increase.

References:

1. Hemelrijk CK, van Zuidam L, Hildenbrandt H. 2015 What underlies waves of agitation in starling flocks. *Behavioral Ecology and Sociobiology* 69, 755–764. (doi:10.1007/s00265-015-1891-3)
2. Hemelrijk CK, Hildenbrandt H. 2012 Schools of fish and flocks of birds: their shape and internal structure by self-organization. *Interface Focus* 2, 726–737. (doi:10.1098/rsfs.2012.0025)
3. Bode NWF, Faria JJ, Franks DW, Krause J, Wood AJ. 2010 How perceived threat increases synchronization in collectively moving animal groups. *Proceedings of the Royal Society B: Biological Sciences* 277, 3065–3070. (doi:10.1098/rspb.2010.0855)
4. Papadopoulou M, Hildenbrandt H, Hemelrijk CK. 2023 Diffusion during collective turns in bird flocks under predation. *Frontiers in Ecology and Evolution* 11. (doi:10.3389/fevo.2023.1198248)

5. Cavagna A, Queiros SMD, Giardina I, Stefanini F, Viale M. 2013 Diffusion of individual birds in starling flocks. *Proceedings of the Royal Society B: Biological Sciences* 280, 20122484–20122484. (doi:10.1098/rspb.2012.2484)

6. Ling H, Mclvor GE, Westley J, Van Der Vaart K, Yin J, Vaughan RT, Thornton A, Ouellette NT. 2019 Collective turns in jackdaw flocks: Kinematics and information transfer. *Journal of the Royal Society Interface* 16. (doi:10.1098/rsif.2019.0450)

7. Couzin ID, Krause J, Franks NR, Levin SA. 2005 Effective leadership and decision-making in animal groups on the move. *Nature* 433, 513–516. (doi:10.1038/nature03236)

Reviewer #3 (Remarks to the Author):

In the paper “Perception of Motion Saliency Shapes the Emergence of Collective Motions”, the authors present a bio-inspired flocking algorithm. In the first part, the authors study leader-follower relations in different collective motion patterns of two species of birds. The authors compute the lag and advance time between pairs of birds and use these to identify leader-follower relationships in the flock. The authors show that collective motion patterns with higher motion saliency also exhibit a stronger leader-follower hierarchy. Based on these results, the authors also develop an algorithm (AMS) to control the flocking behavior of a swarm of ground robots. Experiments are conducted both in simulation and with real robots.

I, unfortunately, lack the expertise to judge about the quality and soundness of the biological part of the study but the paper appears to be sound overall, especially in the swarm robotics part. It is commendable that experiments with real robots have been performed.

In the discussion, the authors state that there exists “a lack of a comprehensive research chain from biological observation to bionic mechanism to bio-inspired swarm robotics”. While it is true that only few works consider the full pipeline, it is also to note that the field of swarm robotics has moved away from pure bio-inspiration (see for example Winfield et al., *SAB* 2004; Brambilla et al., *Swarm Intelligence* 2013; Francesca and Birattari, *Frontiers in Robotics and AI* 2016).

A general comment: The content of this paper is very dense, with many formulas to keep track of. Unfortunately, definitions and formulas are split between “Results”, “Methods”, and “Supplementary Material”. For example, the authors give the definition for τ_{LF} in line 122 (“Results”), the definition for $M_{i,j}$ in line 454 (“Methods”), and the definition of R in the supplementary material. All three of these metrics are reported in the results (Figure 1, Figure 2, and Figure 5, respectively). Due to the distributed locations, it could be challenging for the reader to

find the location of a definition. I would suggest consolidating all definitions in the “Methods” section. Furthermore, the paper is introducing many abbreviations, such as MS, ATHD, LF, MST. I believe that the readability of the manuscript could be improved by replacing most abbreviations with their proper text (especially MS/motion salience and LF/leader-follower).

Response to Reviewer #1

Point 1.0: The manuscript deals with the emergence of collective motion patterns in groups of interacting individuals (e.g., birds, robots). It sets the ambitious objective to study the implications at the collective level of a proposed local interaction mechanism from observational data analysis (in the “Empirical data analysis” section) to model-based theory (“A swarm model with adaptive MS-based interactions”) and robot experiments (“Application to swarm robotics”). In doing so, the authors aim to cover the full “research chain” from observation to experimental swarm robotics, and achieve “system-level understanding” of the proposed mechanism.

The diversity of methodologies adopted in the paper is one of its main strengths, in my view, but it also makes it difficult to summarize the key contributions of the paper. To my current understanding, the main contribution is the introduction of a new local variable, the perception of motion salience (MS), which, according to the authors, (1) correlates with individuals’ leadership status and degree of individual freedom within observed bird flocks, (2) can be used to explain the observed nested structure of leader-follower matrices, and (3) to design effective robot swarms that solve effectively specific collective motion tasks such as evacuation.

Overall, this manuscript is the result of an impressive effort to study the collective implications of a local interaction mechanism from observational data to models and experiments, which is rare in a single paper. The paper could have impact on the collective behavior field, by highlighting the importance of covering the full research cycle from observation to validation through actual implementation. At the same time, I have major concerns about some of the methodologies adopted by the paper, and the novelty and strength of the findings (especially in relation to the observational results). Below, I summarize my concerns and present some suggestions that may be helpful to determine which claims are actually supported by the data and models, stress their novelty, and extensively revise this work as a result.

Response: We sincerely appreciate the valuable feedback and assessment provided by Reviewer #1. We are grateful for her/his recognition of the comprehensive “research chain” from observation to experimental swarm robotics in this work. Next, we addressed each of her/his comments/concerns in order.

Major comments

(A) Methodology

A1. Nestedness.

Point 1.1: The authors build the whole nestedness analysis on the raw NODF values. They compare the nestedness of different empirical matrices by directly comparing the NODF values. They assume that “the NODF normalizes the matrix size, and thus allows matrices of different sizes to be compared” (Supplementary Note 4).

While it’s true that the NODF has a size-dependent normalisation factor, unfortunately in practice, it cannot be used to compare matrices of different sizes and densities. The NODF metric is indeed very sensitive to the input matrix’s size and density, as proven by various works (e.g., Ulrich et al., 2009; Payrato-Borras et al., 2020). In light of this major issue, while some of the nestedness-related claims seem to hold (e.g., by visually looking into the matrices in fig. 1), the current quantitative procedure is not correct.

One related issue is the absence of comparisons against null models. The authors can refer to Gotelli et al. 2012; Mariani et al., 2019; Molina and Stone, 2020 for extensive discussions of the essential role of null models in inferring the significance of structural patterns in network analysis. Null model analysis enables a researcher to establish whether an empirically observed structural metric (e.g., nestedness) is significantly higher than that observed in suitably randomized networks that preserve some chosen properties of the empirical one (see also Staniczenko et al, 2013 for their considerations on the risks of a too conservative null model).

I have two suggestions for the authors, to attempt to validate their claim that certain matrices are more nested than others. One is to perform comparisons against null models of various degree of constraint (see Staniczenko et al, 2013), and establish that some (more nested) matrices exhibit higher discrepancies with respect to their randomized counterparts (e.g., higher z scores) than the less nested ones. This analysis could be done with different nestedness metrics, such as the NODF and spectral radius, to show that the results are independent of the choice of the nestedness metric. Another suggestion is to prioritize nestedness metrics that are known to have weaker dependencies on matrix size and density. According to the most recent systematic comparison of nestedness metrics (Payrato-Borras et al., 2020), the ATC and, to a lesser degree, the spectral radius would be better candidates than the (more popular) NODF.

Response: We thank Reviewer #1 for these very constructive comments and excellent suggestions. Moreover, we would like to express our gratitude to Reviewer #1 for bringing the review (Payrato-Borras et al., 2020) to our attention, which provides us with a comprehensive understanding of the performance of various nestedness metrics. To strengthen our analysis of nestedness in LF networks, we have incorporated three key elements:

(i) introducing a nestedness metric known as Spectral Radius (along with a variant called normalized Spectral Radius) (Payrato-Borras et al., 2020) for comparison.

The Spectral Radius metric (Staniczenko et al., 2020) is a nestedness measure that relies on the spectral properties of double nested graphs. This metric utilizes a theorem that states that within the connected bipartite graphs with $n + m$ nodes and E edges, the graph (represented by the form of an adjacency matrix) that yields the highest spectral radius is indicative of a perfectly nested structure. Notably, Ref.[Staniczenko et al., 2020] demonstrated that graphs with higher levels of nestedness tend to have larger spectral radius, although this relationship is not strictly monotonous. Interestingly, this metric overcomes certain limitations present in previous metrics. Since it involves diagonalizing a symmetric matrix, it is not affected by the ordering of the matrix and avoids any ambiguities associated with determining the maximally packed form of the bipartite matrix encountered in other nestedness metrics. Of course, the important drawback of the Spectral Radius metric is that it is not normalized. Ref.[Payrato-Borras et al., 2020] reported a method to get the normalized Spectral Radius (denoted as n-Spectral radius). That is, if we denoted λ as the spectral radius of a real network and λ_{\max} as the spectral radius of a perfectly nested graph with the same size and fill, the normalized index n-Spectral radius can be calculated as λ/λ_{\max} [Payrato-Borras et al., 2020].

The LF networks of flocks exhibit wide variability in terms of group size and edge density, and these two variables could have distinct effects on the Spectral Radius and NODF metrics, respectively. **Fig.R1** (in the end of response letter) shows three nestedness metrics (NODF, Spectral radius, and n-Spectral radius) as a function of group size, density of edges in LF networks for the Mobbing dataset. We found that there is a strong linear correlation between NODF and edge density of LF networks (**Fig.R1b**), and between Spectral radius and group

size (**Fig.R1c**). To clearly demonstrate these relations, we showed four examples of LF networks labeled as I, II, III and IV in **Fig.R1**. In the cases of example-I, II and III, as the edge density is similar (x-axis of **Fig.R1b**), we observed that their corresponding NODF values are also comparable (y-axis of **Fig.R1b**). However, the Spectral Radius values (y-axis of **Fig.R1c**) differ significantly due to variations in their group sizes (x-axis of **Fig.R1c**). Indeed, because example-III and IV have similar group sizes, their Spectral Radius values also exhibit a close resemblance. Note that there is a substantial disparity in the nestedness between these two examples. In **Fig.R1g**, we compared the nestedness metrics of NODF and Spectral Radius to illustrate the impact of group size on these two metrics. To visually emphasize this impact, the point size in Fig.R1g linearly scales with the corresponding group size. Interestingly, By utilizing the normalized Spectral Radius, the influence of group size on the Spectral Radius metric is eliminated (**Fig.R1e,f**). This allows for a more focused assessment of the nestedness without being confounded by variations in group size. Obviously, the normalized Spectral Radius effectively eliminates the influence of group size on the metric. Therefore, we used normalized Spectral Radius as a comparison with NODF.

(ii) replenishing the comparisons of nestedness against four null models with various degree constraints of LF networks.

Following the reviewer’s constructive suggestion, we realized the crucial role of null models in accurately inferring and interpreting the structural patterns in nestedness analysis. Here, we introduced four null models with various degree constraint of LF networks.

- **Null-1 Equiprobable:** the connectance (proportion of existing interactions) of the newly generated network is preserved. However, the number of interactions in which each node participates is not specifically controlled in this context. It is the same with null mode-(i) in Ref.[Staniczenko et al., 2020].
- **Null-2 Average:** In the newly generated network, both the connectance and the expected number of interactions for each node are preserved. It is the same with null mode-(ii) in Ref.[Staniczenko et al., 2020].
- **Null-3 Rows Average:** In the newly generated network, both the connectance and the expected number of interactions of column nodes are preserved.
- **Null-4 Fixed:** In the newly generated network, both the connectance and the degree distribution of each node (not just their expected number of interactions) are preserved. It is the same with null mode-(iii) in Ref.[Staniczenko et al., 2020].

Fig.R2a and **Fig.R3a** show the original LF network and four null modes for example-I, II, III, and IV. Since Null-4 preserves the overall structure and degree distribution of the original network to a great extent, Null-4 is only able to generate networks that are identical to the original network in the case of perfectly nested LF networks (example-I in **Fig.R2a**). For near-perfectly nested networks (e.g., example-I, II, III in **Fig.R2a**), Null-4 still produces networks that exhibit minimal changes compared to the original network.

(iii) establishing that the more nested LF matrices exhibit higher discrepancies with respect to their randomized counterparts (e.g., higher z-scores) than the less nested ones.

In using NODF (**Fig.R2b-e**), we found that the average NODF of Null-4 (over 100 independent randomizations) is extremely close to real NODF in the entire range of $[0, 1]$ (**Fig.R2e**). For the normalized Spectral Radius, the average value of Null-4 is also extremely comparable to that of real LF network (**Fig.R3e**). Two results calculated by different nestedness metrics are consistent. For the other three null models, their average NODF values only match the real

NODF in the range [0,0.2]; and as the real NODF value increases, there is a growing difference between the average NODF and the real NODF (**Fig.R2b-d**). This indicates that the null-4 model with fixed degree distribution effectively preserves the nestedness patterns present in the original network. Moreover, four null models show the more nested LF networks exhibit higher z-scores with respect to their randomized counterparts than the less nested ones, no matter what the nestedness is measured by NODF (**Fig.R2f-i**) or normalized Spectral Radius (**Fig.R3f-i**).

Many thanks to Reviewer #1 for this constructive comment and suggestion. We have incorporated the above revisions in Supplementary Sec.4 (see **SI: pages 7-10** and **Supplementary Figs.17-19**).

A2. MS and velocity.

Point 1.2: According to its definition, an element of the MS quantifies the perceived movement of an individual perceived by another individual within a time window. I wonder how well the MS correlates with a bird's radial and angular velocity. If the correlation is very high, the MS-LF correlation found by the author could be a manifestation of the well-known property that faster birds tend to occupy leadership positions in observed groups (Krause et al., 2000; Pettit et al., 2015).

See (B) below for related comments, on the lack of critical reflection of the manuscript with respect to existing literature on leadership.

Response: We thank Reviewer #1 for this constructive comment. We overlooked a discussion about the relationship between MS and speed in the previous version of manuscript. Following reviewer's excellent suggestions, we systematically analyzed their correlation and compared with MS-LF relations.

(i) Statistics of bird's average speed, radial and angular speed of mobbing, transit and circling datasets

To investigate the relations between MS (or MST) and speed, we first counted each bird's average speed, average radial speed and average angular speed. Note that each bird's average speed, radial and angular speed is calculated over the entire duration of a flock. For mobbing, circling and transit datasets, **Fig.R4** shows three speeds of 3103, 75683, and 2597 birds, respectively. We found that the average speed and radial speed of transit dataset is significantly larger than that of mobbing and circling datasets, and they are quite similar to each other for mobbing and circling datasets. The average angular speed of flocks, in descending order, is observed to be highest in mobbing flocks, followed by circling flocks, and finally transit flocks. This order aligns with the observed motion patterns exhibited by these types of flocks.

(ii) Correlation between MS and three kinds of speed

In addition to the correlation analysis between MS and LF shown in Fig.3a (in the main text), we also conducted a correlation analysis of two vectors from MS and three types of speed for a given period $[t - \tau, t]$ within a flock (see workflow in **Fig.R5a-c**). Consider the entire flock, we calculated the Spearman correlation (ρ) for various combinations of (t, τ) to assess their level of correlation (**Fig.R5d**). Next we extended the correlation analysis to three datasets and yielded the distribution of $\rho_{MS-Speed}(t, \tau)$, $\rho_{MS-Radial}(t, \tau)$ and $\rho_{MS-Angular}(t, \tau)$ (**Fig.R5e,f**). In order to facilitate a clear comparison of the correlation levels between MS-LF (as shown in

Fig.3 of the main text) and three types of MS-Speed, **Fig.R6** presents boxplots of $\rho_{\text{MS-LF}}(t, \tau)$, $\rho_{\text{MS-Speed}}(t, \tau)$, $\rho_{\text{MS-Radial}}(t, \tau)$ and $\rho_{\text{MS-Angular}}(t, \tau)$. These boxplots provide a visual representation of the correlation strengths for the different combinations of (t, τ) values. Interestingly, regardless of three flocking datasets and α values in MS, the Spearman correlation for the three types of MS-Speed does not demonstrate a significant positive or negative dominance (**Fig.R6**). This suggests that the correlations between these variables are not consistently biased towards either positive or negative values across different scenarios. In particular, when $\alpha = 1$, $\rho_{\text{MS-LF}}$ exhibits predominantly positive values in mobbing and circling flocks. However, this phenomenon could not be observed in the correlation of the three types of MS-Speed, as depicted in **Fig.R6b1** and **b2**.

(iii) Comparison of correlation of LF-MS and LF-Speed

Given the well-known observation that faster birds tend to assume leadership positions in homing pigeon flocks [Pettit et al., 2015], a natural concern arises regarding the strength of the correlation between LF-MS and LF-Speed. Here we used a mobbing flock (shown in Fig.1a) to show the workflow of comparison:

- 1) *First*, performing the Spearman correlation to yield $\rho_{\text{LF-MS}}(t, \tau)$ and $\rho_{\text{LF-Speed}}(t, \tau)$, $\rho_{\text{LF-Radial}}(t, \tau)$, $\rho_{\text{LF-Angular}}(t, \tau)$ for different combinations of (t, τ) . In **Fig.R7a-c**, each panel contains 231 points corresponding to 231 combinations of (t, τ) within the mobbing flock.
- 2) *Second*, scatter-plotting of $\rho_{\text{LF-MS}}(t, \tau)$ and $\rho_{\text{LF-Speed}}(t, \tau)$ (or $\rho_{\text{LF-Radial}}(t, \tau)$ or $\rho_{\text{LF-Angular}}(t, \tau)$), and calculating the difference of ρ as $\Delta\rho(t, \tau) = \rho_{\text{LF-MS}}(t, \tau) - \rho_{\text{LF-Speed}}(t, \tau)$, $\rho_{\text{LF-MS}}(t, \tau) - \rho_{\text{LF-Radial}}(t, \tau)$, $\rho_{\text{LF-MS}}(t, \tau) - \rho_{\text{LF-Angular}}(t, \tau)$. The scatter plot of ρ and heatmap of $\Delta\rho$ for the mobbing flock are shown in **Fig.R7a-c** and **Fig.R7d-f**. The gradient color ranging from red to white to blue in **Fig.R7d-f** is indicative of the value of $\Delta\rho \in [-2, 2]$. To clearly demonstrate $\Delta\rho$ in different combinations of (t, τ) , **Fig.R7g** shows the flocking trajectory from $[t - \tau, t]$ with the gradient background color scaling with $\Delta\rho = \rho_{\text{LF-MS}}(t, \tau) - \rho_{\text{LF-Speed}}(t, \tau)$. We observed that the $\Delta\rho(t, \tau)$ values indicated by the red background color consistently correspond to the process of collective turn.
- 3) *Finally*, classifying $\Delta\rho$ from all the combinations of (t, τ) . For example, in **Fig.R7h-j**, the 231 points are categorized into 3 parts: when $|\Delta\rho| < 0.2$, it signifies that the two correlations are very similar; $\Delta\rho$ exceeding (or falling below) a specific threshold indicates that the $\rho_{\text{LF-MS}}(t, \tau)$ is noticeably greater than (or less than) the LF-Speed correlation. Besides, y-axis of **Fig.R7h-j** records the corresponding average angular and nestedness. Interestingly, for $\Delta\rho(t, \tau) = \rho_{\text{LF-MS}}(t, \tau) - \rho_{\text{LF-Speed}}(t, \tau)$ and $\rho_{\text{LF-MS}}(t, \tau) - \rho_{\text{LF-Angular}}(t, \tau)$ shown in **Fig.R7h,j**, the nestedness of those sub-flocks with $\Delta\rho > 0.5$ (the red background color in **Fig.R7g**) is the highest, while the nestedness of sub-flocks with $\Delta\rho < -0.5$ (the blue background color in **Fig.R7g**) is the lowest. This trend is the same with average angular speed.

In the mobbing flock, we observed a clear pattern where the sub-flocks with $\rho_{\text{LF-MS}}(t, \tau) - \rho_{\text{LF-Speed}}(t, \tau) > 0.5$ (or $\rho_{\text{LF-MS}}(t, \tau) - \rho_{\text{LF-Angular}}(t, \tau) > 0.5$) consistently exhibit the highest nestedness, while those sub-flocks with $\Delta\rho < -0.5$ display the lowest nestedness. We hypothesized that this phenomenon may be related to the collective motion patterns. Consequently, we replicated the same analysis process to a transit flock as illustrated in Fig.1c. In **Fig.R8**, we found three kinds of $\Delta\rho$ is significantly lower than that of mobbing flock, because the color intensity in **Fig.R8d-f** is noticeably lighter compared to that in **Fig.R7d-f**. And the

nestedness value of $\Delta\rho > 0.5$ and $\Delta\rho < -0.5$ consistently remains at a relatively low level (**Fig.R8h-j**). Based on these two examples with distinct motion patterns, we hypothesized that this phenomenon of different $\Delta\rho$ corresponding to different flocking properties may be associated with the collective motion patterns.

Furthermore, we extended the workflow to mobbing, circling and transit datasets (**Fig.R9**). In mobbing and circling dataset, the sub-flocks with $\Delta\rho > 1$ exhibit the highest nestedness, while those sub-flocks with $\Delta\rho < -1$ display the lowest nestedness. The nestedness of sub-flocks with $|\Delta\rho| < 0.2$ are positioned in the middle. Note that in **Fig.R9** the trend is the same for three kinds of $\Delta\rho$. For the transit dataset, we could not observe the clear pattern between $\Delta\rho > 1$ and $\Delta\rho < -1$. The validation with three large datasets confirms our hypothesis. The flock (or sub-flock) with maneuverable motions (such as mobbing and circling flocks), characterized by higher angular speed and nestedness, exhibits a notably stronger positive correlation between LF and MS when compared to the correlation between LF and Speed. Conversely, the correlation between LF and Speed is significantly greater than that between LF and MS for the flock with lower angular speed and nestedness (e.g., the transit flocks in this work). In summary, LF-MS may be more effective in describing the flocks with maneuverable motions, whereas LF-Speed is better suited for characterizing flocks with smooth motions. The difference between MS and Speed in this analysis could mainly be attributed to the scope of their respective definitions. From the definition, MS directly or indirectly takes into account the variations in position, heading, speed and acceleration over a period of flock. Three kinds of speed serve as scalar representation of collective motions. It is evident that speed, when compared to MS, is not a suitable description of the collective motion modes, especially for extremely agile flocks.

A3. Mechanism behind observational results.

Point 1.3: The manuscript falls short of providing possible reasons why the MS correlates with leadership. The authors proceed by incorporating MS-based mechanisms into the model-based and experimental analyses, but do not discuss why the MS is to be expected to correlate with leadership. Is it already implied by the definition?

If not, one possible explanation is that faster birds have both high MS and leadership. But this would be a confounding factor coming from an already-known result from the literature.

What are alternative explanations?

Response: We thank Reviewer #1 for this critical comment. As mentioned in our response to **Point 1.2**, we performed the correlation analysis between LF and Speed (see **Point 1.2-ii**), and then compared the correlation between LF-MS and LF-Speed (see **Point 1.2-iii**).

In the revised manuscript, we have absorbed the correlation results of MS-Speed and comparison of correlation between LF-MS and LF-Speed (see **main text: page 10, lines 318-340**):

“Given the well-known finding that faster individuals tend to assume leadership positions in animal groups^{56,57}, two natural concerns arise regarding the correlation of MS-Speed and comparison of correlation between LF-MS and LF-Speed. Here the “Speed” of each individual are three types: average speed, average radial speed and average angular speed. See Supplementary Sec.6 for detailed introduction and results. Interestingly, regardless of three flocking datasets and α values in MS, correlation between MS and three types of Speed do not demonstrate a significant positive or negative dominance. Moreover, we performed the correlation analysis between LF and Speed, and

then compared the correlation between LF-MS and LF-Speed in mobbing, circling and transit datasets. We found that the flock (or sub-flock) with maneuverable motions, such as mobbing and circling flocks, which are characterized by higher angular speed and nestedness, shows a notably stronger positive correlation of LF-MS when compared to the correlation of LF-Speed. Conversely, for the flock with lower angular speed and nestedness (e.g., the transit flocks), the correlation of LF-Speed is significantly greater than that of LF-MS. In summary, LF-MS may be more effective in describing the flocks with maneuverable motions, whereas LF-Speed is better suited for characterizing flocks with smooth motions. These findings do not contradict the well-known property that faster birds tend to occupy leadership positions^{57,58}. One reason is that we also observed a tendency for leaders to fly at the front of the flock in the datasets we analyzed (Supplementary Fig.29). The difference between MS and Speed could mainly be attributed to the scope of their respective definitions. From the definition, MS directly or indirectly takes into account the variations in position, heading, speed and acceleration over a period of flock. Three kinds of speed serve as scalar representation of collective motions. It is evident that speed, when compared to MS, is not a comprehensive characteristic for distinguishing the modes of collective motion, especially for extremely agile flocks.”

Moreover, in the revised Supplementary Information, we have added a section of “**6. Correlation analysis of MS-Speed and LF-Speed**” for details (see **SI: pages 11-13** and **Supplementary Figs.32-37**).

Point 1.4: A related concern: isn't the explanation in the paragraph starting with “The above phenomena in different patterns...” circular? In flocks with maneuverable motions, individuals have higher radial and angular speeds, which can result in higher MST. What does the MST add to the intuition that individuals in flocks with maneuverable motions are just faster?

To address these concerns, it would be beneficial if the authors could present in detail the correlations between the MST, individual-level radial and angular velocities, and clarify whether the MST can generate *new* observational insight beyond those that can be generated by only looking at the birds' velocity.

Response: We thank Reviewer #1 for this very constructive comment. For the concern about correlations between the MST, individual-level radial and angular velocities, we performed the correlation analysis between MST and three kinds of speed. **Fig.R10** indicates that neither positive nor negative correlations between MST and Speed are predominant in the three flocking datasets.

Given the relevance of next **Point 1.5** to the concern about MST, we systematically presented new observational insights regarding why MST is linked to the extent to which a focal individual aligns with its neighbors or follows its own drive. Please see our response **Point 1.5** for the details.

A4. Individual freedom.

Point 1.5: The authors seem to propose that MST quantifies an individual's freedom. However, I do not understand which result, if any, proves this very general claim. It seems more of a consequence of the definition of MST and its interpretation by the authors.

First, is "freedom" precisely defined? Besides, T is the time at which a pair's relative motion achieves its maximum, but the mere existence of the maximum does not automatically imply that there is less or no "freedom" for longer timescales. Perhaps, it could help to study carefully the

actual shapes of M_{ij} as a function of time, especially around the maximum and in the long- τ limit. Finally, similar to my comment A3, it would be helpful to see correlations between the MST and individual velocities.

Response: We thank Reviewer #1 for this very constructive comment. Using ‘flying freedom’ or ‘individual freedom’ to interpret the meaning of MST is anthropomorphic and lacks the computational evidence. We have revised the notion of MST in the revised manuscript (see **main text: pages 7-8, lines 229-230**):

“Therefore, we assumed that MST can quantify the extent to which the focal individual closely aligns with its neighbors or follows its own drive.”

Thanks to reviewer’s suggestion, we reexamined the computation and interpretation of MST. The computation of MST involves two steps. First, we averaged $M_{ij}(t, \tau)$ over all the time stamps t from the period $[t_s, t_e]$, resulting in $\langle M_{ij} \rangle(\tau)$ as a function of τ . Second, we chose the maximal motion salience time (MST), denoted as τ_{ij}^{MS} , to be the τ value where $\langle M_{ij} \rangle(\tau)$ reaches its maximum. Naturally, τ_{ij}^{MS} characterizes how long it will take individual- j to reach the maximum of relative motion changes from individual- i ’s perception. If τ is shorter than MST, $\langle M_{ij} \rangle(\tau)$ could initially approach the maximum as τ increases. The increase in $\langle M_{ij} \rangle(\tau)$ indicates that the motion difference between two individuals could become more pronounced. When τ approaches MST, the motion difference reaches its maximum, after which the movements of the two individuals start to converge. Please see examples of MST curves in **Fig.R11b,e**. Therefore, we assumed that MST can quantify the extent to which the focal individual closely aligns with its neighbors or follows its own drive. This is what we meant by “MST can quantify the freedom of the focal individual”.

To further demonstrate this point, we calculated the average velocity difference $\Delta v_{ij} = 1 - \langle \mathbf{v}_i(t) \cdot \mathbf{v}_j(t) \rangle$ throughout the entire duration of a flock for a bird pair- (i, j) . A higher value of $\langle \mathbf{v}_i(t) \cdot \mathbf{v}_j(t) \rangle$ indicates a closer alignment between the two individuals, resulting in a smaller average velocity difference Δv_{ij} . For example, **Fig.R11a-f** show the trajectories, $\langle M_{ij} \rangle(\tau)$ and temporal $\mathbf{v}_i(t) \cdot \mathbf{v}_j(t)$ of pair-(1,6) and (4,7) from the mobbing flock shown in Fig.1a. We found that MST of pair-(1,6) is larger than that of pair-(4,7), and correspondingly, Δv_{ij} of pair-(1,6) is also larger than that for pair-(4,7). This suggests that pair-(1,6) with higher MST experiences the larger velocity difference, while pair-(4,7) displays the opposite pattern compared to pair-(1,6). Thus, for the entire flock of Fig.1a, **Fig.R11j** demonstrates that the MST and average velocity difference for all bird pairs display an obvious positive relation, indicating that larger MST corresponds to a bigger velocity difference. Besides, we also considered the area under the curve of $\langle M_{ij} \rangle(\tau)$ to validate our hypothesis (highlighted by grey in **Fig.R11b,e**). **Fig.R11k** illustrates that there is also a clear positive relationship between MST area and the average velocity difference for all bird pairs.

Furthermore, we broadened the analysis to mobbing, circling and transit datasets (**Fig.R12**). The findings indicate that the predominant correlations in all three flocking datasets are positive for $\rho_{MST-\Delta v_{ij}}$ and $\rho_{MST-MST_{area}}$. This implies that the perception of MS could quantify the trade-off between individual drive and group cohesion in collective motions, because MST provides a measure of the degree to which the focal individual aligns closely with its neighbors or prioritizes its own internal drive.

Given the relevance of next **Point 1.6** to the concern about wording of MST in the manuscript, please see our response **Point 1.6** for the interpretation of MST.

Point 1.6: Because of these ambiguities, I do not think that the following claim is supported by the current evidence: “the above phenomena [...] demonstrate that the smaller the average MST, the smaller the individual freedom”.

Response: We thank Reviewer #1 for this critical comment. After receiving comprehensive and meticulous feedbacks from Reviewer #1, we have come to realize that there were shortcomings in the precision of our descriptions and instances of over-interpretation in the previous version.

In the revised manuscript, we have absorbed the new observations between MST and velocity difference, and rewritten the interpretation of MST (see **main text: pages 7-8, lines 221-247**):

“For a given $\tau > 0$, averaging $M_{ij}(t, \tau)$ over all the time stamps t from the period $[t_s, t_e]$ yields $\langle M_{ij} \rangle(\tau)$. We defined the maximal motion salience time (MST), denoted as τ_{ij}^{MS} , to be the τ value where $\langle M_{ij} \rangle(\tau)$ reaches its maximum. τ_{ij}^{MS} characterizes how long it will take individual- j to reach the maximum of relative motion changes from individual- i 's perception. If τ is shorter than MST, $\langle M_{ij} \rangle(\tau)$ could initially approach the maximum as τ increases. The increment of $\langle M_{ij} \rangle(\tau)$ means the motion difference between two individuals could become more and more pronounced. When τ approaches MST, the motion difference reaches its maximum, after which the movements of the two individuals start to converge (see the curves of $\langle M_{ij} \rangle(\tau)$ as a function of τ in **Fig.2c-e**). Therefore, we assumed that MST can quantify the extent to which the focal individual closely aligns with its neighbors or follows its own drive. To verify this assumption, we calculated the average velocity difference $\Delta v_{ij} = 1 - \langle \mathbf{v}_i(t) \cdot \mathbf{v}_j(t) \rangle$ throughout the entire duration of a flock for a bird pair- (i, j) . A higher value of $\langle \mathbf{v}_i(t) \cdot \mathbf{v}_j(t) \rangle$ indicates a closer alignment between the two individuals, resulting in a smaller average velocity difference Δv_{ij} . For example, Supplementary Fig.23a-f show the trajectories, $\langle M_{ij} \rangle(\tau)$ and temporal velocity difference $\mathbf{v}_i(t) \cdot \mathbf{v}_j(t)$ of pair-(1,6) and (4,7) from the mobbing flock shown in Fig.1a. Our findings reveal that MST of pair-(1,6) is greater than that of pair-(4,7), and correspondingly, Δv_{ij} of pair-(1,6) is also larger than that for pair-(4,7). This suggests that pair-(1,6), with a higher MST, experiences larger velocity difference, while pair-(4,7) displays the opposite pattern compared to pair-(1,6). Thus, for the entire mobbing flock, Supplementary Fig.23j demonstrates that MST and average velocity difference for all bird pairs display an obvious positive relation, indicating that larger MST corresponds to a bigger velocity difference. Furthermore, we broadened the above correlation analysis to mobbing, circling and transit datasets (Supplementary Fig.24). The findings indicate that the predominant correlations in all three flocking datasets are positive between MST and average velocity difference. This implies that the perception of MS could quantify the trade-off between individual drive and group cohesion in collective motions, because MST provides a measure of the degree to which the focal individual aligns closely with its neighbors or prioritizes its own internal drive.”

(B) Strength and significance of results

B1. Novelty of observational results.

Point 1.7: As acknowledged by the authors, the manuscript is not the first one to study leader-follower relationship and, most importantly, correlates of leadership status in animal groups. However, the manuscript falls short at comparing its results from those already known from this literature. For example:

— Rosenthal et al. (2015) found how an individual position in the communication network correlates with its influence in the group.

- Previous studies related an individual's speed with its leadership status in the group (Krause et al., 2000; Pettit et al., 2015).
- Other properties correlated with leadership include nutritional requirements, physiological traits, spatial position within the group. The authors could refer to Jolles et al., 2020 for a recent review.

Response: We appreciate Reviewer #1 for bringing these papers to our attention. We apologize for overlooking the discussion and comparison with existing studies on leadership in the previous version of our manuscript.

In the revised version, we have incorporated the analysis results about (i) correlation of MS-Speed, MST-Speed, (ii) comparison of correlation between LF-MS and LF-Speed, and (iii) new observations between MST and velocity difference. Based on the new results, we have rewritten the interpretations of MST/MS, cited the related references in the main text (see **main text: pages 6-8, lines 184-271**), and replenished the detailed analysis information in Supplementary Sec.6 (see **SI: pages 11-13 and Supplementary Figs.32-37**).

Point 1.8: How do the observational results relate with these findings? Critical reflection with respect to this literature is needed to ensure the novelty of the results. For example, if one controlled for known properties in the literature (e.g., speed), would the association between MST and the leadership status remain significant?

Response: We thank Reviewer #1 for this constructive comment. Yes, if we controlled the speed of different flocks, we still observed the predominantly positive correlations of LF-MS across the majority of speed ranges in both mobbing and circling datasets, while the transit dataset does not show any preference of positive or negative correlation.

For example, **Fig.R13** and **Fig.R14** illustrates the distribution of correlation between LF and MS if three flocking datasets are categorized by average speed in 1m/s interval and average angular speed in 0.1rad/s interval, respectively. In both mobbing and circling datasets, we observed predominantly positive correlations of LF-MS across the majority of speed ranges, regardless of whether they are categorized by average speed (**Fig.R13a,b**) or average angular speed (**Fig.R14a,b**). However, the transit dataset does not show any preference of positive or negative correlation across the majority of speed ranges of average speed (**Fig.R13c**) or average angular speed (**Fig.R14c**). To strength our correlation results of LF-MS, we have absorbed this result in the revised manuscript (see **main text: page 10, lines 311-317**):

“Furthermore, even when controlling for the average speed of a flock (e.g., categorizing three flocking datasets by average speed in 1m/s intervals or average angular speed in 0.1rad/s intervals), we consistently observed predominantly positive correlations between LF and MS across the majority of speed ranges in both mobbing and circling datasets. It is observed regardless of whether the categorization is based on average speed or average angular speed. However, this pattern is not evident in the transit dataset (Supplementary Fig.30,31).”

B2. Strength of experimental results.

Point 1.9: In the collective following experiments, the authors use R to measure the collective response performance of robot swarms under different local mechanisms. However looking at figures 5g-h, it seems unclear that the AMS brings an advantage compared to the ATHD.

Response: We thank Reviewer #1 for this critical comment. We apologize for not clearly addressing this in the previous version of our manuscript.

In the collective following experiments, optimal performance necessitates timely adapting the heading difference compared to the informed agent without delay. Indeed, ATHD could be considered as a kind of theoretically ideal response to neighbors' perturbations, because it could adaptively alter the heading difference with the leader and it does not involve any delay. For AMS, it involves the adaptive influences on neighbors' next decision, but the adaptive coefficient is calculated from a period of τ . Clearly, while a smaller τ leads to a smaller R in the collective following experiments, it also leads to an extended evacuation time for the swarm in the collective evacuation experiments. We discussed the effect of perceiving time τ on AMS in Supplementary Sec.7.

Interestingly, despite AMS taking into account a delay of τ , the collective following experiments show that the collective response of AMS interactions closely approximates that of ATHD interactions. This demonstrates the advantage of AMS in responding to transient perturbations. In the revised manuscript, we have rewritten this point (see **main text: page 12, lines 409-419**):

“On the other hand, we also compared the performance of AMS with ATHD in the collective following experiments. ATHD could be considered a theoretically ideal response to neighbors' perturbations, because it could adaptively alter the heading difference with the leader and does not involve any delay. Regarding AMS, it encompasses adaptive influences on a neighbor's subsequent decision, with the adaptive coefficient being derived from a period of τ . Clearly, while a smaller τ leads to a smaller R in the collective following experiments, it also leads to an extended evacuation time for the swarm in the next experiments about collective evacuation. We discussed the effect of perceiving time τ on AMS in Supplementary Sec.8.3. Interestingly, despite AMS taking into account a delay of τ , the collective following experiments show that the collective response of AMS interactions closely approximates that of ATHD interactions (**Fig.5g,h**). This demonstrates the advantage of AMS in responding to transient perturbations.”

Point 1.10: The collective evacuation experiment seems to be the strongest result in the manuscript, to my current understanding, because here, the authors are able to show an actual performance improvement of the AMS compared to other methods. I wonder if this point could be strengthened in the manuscript, perhaps by providing more specific insights on the practical implications of this result.

Response: We thank Reviewer #1 for this excellent suggestion. In Supplementary Sec.8.2, we presented the swarm model utilized for collective evacuation in this study as following:

$$\mathbf{v}_i(t + dt) = (1 - \delta_t)(\hat{\mathbf{v}}_{al,i} + \hat{\mathbf{v}}_{rep,i}^a) + \hat{\mathbf{v}}_{g,i} + \delta_t(\hat{\mathbf{v}}_{col,i}^a + \hat{\mathbf{v}}_{col,i}^w).$$

This model includes 5 parts for agent- i :

- (i) alignment with neighbors in the sensing radius, $\hat{\mathbf{v}}_{al,i}$;
- (ii) repulsion among the near neighbors, $\hat{\mathbf{v}}_{rep,i}^a$;
- (iii) inter-agent collision avoidance, $\hat{\mathbf{v}}_{col,i}^a$;
- (iv) agent-wall collision avoidance, $\hat{\mathbf{v}}_{col,i}^w$;
- (v) guidance velocity for agent to pass through the exit, $\hat{\mathbf{v}}_{g,i}$.

For AMS, ATHD and Average, the only difference is the adaptive factor ($w_{ij}(t)$) of $\hat{\mathbf{v}}_{al,i}(t)$ in the above framework of collective evacuation.

Interestingly, in the collective evacuation experiments shown in Fig.6c (in the main text) and Supplementary Fig.45, we observed that utilizing AMS facilitates the evacuation of the entire group through a narrow gate with the shortest time, even when there were individuals moving

in the opposite direction of the exit. As we know that the other four parts of velocity components except $\hat{\mathbf{v}}_{al,i}$ are same for AMS, ATHD and Average, we focused on comparing $\hat{\mathbf{v}}_{al,i}$ and \mathbf{v}_i . To gain a better understanding of the self-organized mechanism of AMS and why certain individuals (highlighted by red boxes in Supplementary Fig.45) are moving in the opposite direction of the exit, **Fig.R15a** presents three types of information about these individuals: trajectory, $\mathbf{v}_i(t)$ (black arrow), and $\hat{\mathbf{v}}_{al,i}$ (red arrow). Note that $\mathbf{v}_i(t)$ is the true direction of motion in the experiment shown in Fig.6c, and $\hat{\mathbf{v}}_{al,i}$ is the unit vector of alignment calculated by

$$\mathbf{v}_{al,i} = \hat{\mathbf{v}}_i + \sum_{j \in \mathcal{S}_{al,i}} w_{ij} \hat{\mathbf{v}}_j \quad \text{if } d_{ij} < r_{al}$$

and $w_{ij}(t) = \frac{M_{ij}(t,\tau)}{\sum M_{ij}(t,\tau)}$ for $j \in \mathcal{S}_i$. We found that the trends exhibited by the black and red arrows along the trajectory align remarkably well in **Fig.R15a**. This inspires us to compare the dot product between $\mathbf{v}_i(t)$ and $\hat{\mathbf{v}}_{al,i}(t)$ for the results of AMS, ATHD and Average shown in Fig.6a-c. **Fig.R15b** indicates that the dot product between $\mathbf{v}_i(t)$ and $\hat{\mathbf{v}}_{al,i}(t)$ in descending order is AMS, ATHD and Average, and AMS has the least variance of dot product. It demonstrates that the adaptive rule induced by MS contribute the most significance during the process of collective evacuation compared to ATHD and Average interaction. In the revised version, we added this result in Supplementary Sec.8.2 (see **SI: page 18** and **Supplementary Fig.46**).

Point 1.11: Admittedly, while I could follow the logic and results of the experiments, I was unable to fully assess its technical details because of my lack of technical expertise on robot experiments.

Response: The swarm robotics system could support $\sim 10^2$ magnitudes of miniature two-wheel differential mobile robots in a motion capture arena. Each robot is equipped with two stepper motors with reduction gears, a PCB board for motion control and power management, a PCB board for communication and decision making, and a marker deck at the top for hosting passive infrared reflective balls. For the architecture of swarm system, please see Methods in the main text and Supplementary Fig.40.

Reviewer #3 kindly provided the following comment on our work regarding swarm experiments: *“The paper appears to be sound overall, especially in the swarm robotics part. It is commendable that experiments with real robots have been performed”*.

In addition, the swarm robotics system successfully supported the experiments performed in the following published papers:

- Shuai Zhang, Xiaokang Lei, Mengyuan Duan, Xingguang Peng, and Jia Pan. A Distributed Outmost Push Approach for Multi-robot Herding. IEEE Transactions on Robotics, conditionally accepted (2023.11).
- Xiaokang Lei, Yalun Xiang, Mengyuan Duan, Xingguang Peng. Exploring the criticality hypothesis using programmable swarm robots with Viseck-like interactions. Journal of Royal Society Interface, 2023, 20:20230176.
- Xiaokang Lei, Shuai Zhang, Yalun Xiang. Self-organized adaptive multi-target trapping of swarm robots based-on density interaction. Complex & Intelligence Systems. 2023.
- Shuai Zhang, Xiaokang Lei, Zhicheng Zheng, Xingguang Peng. Collective fission behavior in swarming systems with density-based interaction, Physica A: Statistical Mechanics and its Applications, 2022, 603:127723.

Hence, the swarm robotics system and our setup of two swarm experiments are dependable for validating the swarm models from theory to practical applications.

(C) Clarity of exposition.

C1. Sentences implying intentionality and causality.

Point 1.12: There are several sentences that imply that the system or its actors make intentional decisions to achieve some objective, and that the AMS has causal impact on some variables. Examples include, “flocks with higher maneuverable motions (e.g., mobbing and circling) *prefer* to have a more nested structure of LF relations and a clear hierarchy to maintain group cohesion”; “individuals with higher MS *prefer* to lead the group”; “tactful *strategy to organize* the group”. However, such intentional claims seem not supported by the purely correlational nature of the observational analysis performed by the authors. Because of this, I would recommend watering down “intentional statements” and replacing with correlational statements (e.g., “prefer to have” becomes “tend to exhibit”), unless stronger arguments can be provided by the authors.

Response: We thank Reviewer #1 for this critical comment. Based on her/his suggestions, we have avoided intentional statements throughout the manuscript. For examples,

- see **main text: page 2, lines 26-28:**

“We find that flocks with higher maneuverable motions (i.e., *mobbing* and *circling*) **tend to exhibit** a more nested structure of leader-follower (LF) relations and a clear hierarchy to mitigate the damage of individual **drive** to group cohesion.”

- see **main text: page 3, lines 94-96:**

“*First*, we found that flocks with higher maneuverable motions (e.g., *mobbing* and *circling*) **tend to exhibit** a more nested structure of LF relations and a clear hierarchy to maintain group cohesion.”

C2. General claims.

Point 1.13: Some of the claims in the introduction and discussion are very general, but the level of presented methods and evidence not strong enough to fully support them.

Response: We thank Reviewer #2 for the very critical comments and suggestion. We have revised those sentences of over-interpretation in the previous version. Please review the detailed responses below.

Examples include:

— “perception of MS may have major adaptive and evolutionary significance to balance the trade-off between individual freedom and group cohesion in collective motions”, which seems a consequence of the authors’ interpretation and not an actual result, as discussed above;

Thanks to the constructive feedback from the reviewer, it is evident that this sentence goes beyond the actual results and represents an over-interpretation. Here, we have summarized our primary findings from the empirical analysis of three flocking datasets.

Moreover, we acknowledged the significance of collective behaviors such as mobbing, roosting, or transiting in relation to their ecological environment and physiological characteristics of the study species. Regrettably, due to our limited expertise in ecology and animal behavior, we were unable to thoroughly consider the intricate ecological context in our analysis. For example, for the mobbing and transit datasets, the Jackdaw (*Corvus monedula*)

always forms long-term monogamous relationships and both parents contribute to rearing the young, therefore, large numbers of individuals recorded in these datasets include mated pairs, unpaired individuals and juveniles. Indeed, we acknowledged that the impact of pair-bonded relationships on quantifying MS is overlooked in our analysis. It is evident that these relationships can significantly influence the measurement of MS. We hoped that future research will bridge this gap and provide a more nuanced perception of MS within the context of the study species' ecological and physiological attributes. In the new version, we openly admit this point in the Discussion (see **main text: pages 14-15, lines 491-519**):

“Our work represents a new and substantive departure from the status quo by shifting the focus from one of the specific aspects to a systems-level understanding of collective motions. In particular, we utilized three large bird-flocking datasets encompassing diverse ecological context and motion patterns, namely mobbing, circling, and transit. This enabled us to conduct a systematic investigation into the emergence of LF relations and concurrently identify the factors that facilitate to the development of such relationships. The empirical findings indicate that the flocks characterized by higher maneuverability, such as those observed in mobbing and circling behaviors, tend to exhibit a more nested structure of LF relations and a distinct hierarchy. This organizational pattern could help to potentially alleviate the negative impact of individual drive on group cohesion. In contrast, flocks with smooth motion (i.e., transit) do not display this strategic tendency to organize the group in a similar manner. To elucidate the empirical findings, we introduced the metric MS and performed the correlation analysis between LF and MS. Interestingly, we found that individuals with higher MS tend to occupy the higher leading tier in LF networks. It means that higher MS will make an individual less influenced by others, thereby implying a stronger inclination to follow its own tendencies and perhaps end up leading the group.

We acknowledged the significance of collective behaviors such as mobbing, roosting, or transiting in relation to their ecological environment and physiological characteristics of the study species. However, our primary focus in this study is to elucidate the underlying factors that shape the emergence of collective motions and to establish a comprehensive research chain from biological observation to bionic mechanism to bio-inspired swarm robotics. Regrettably, due to our limited expertise in ecology and animal behavior, we were unable to thoroughly consider the intricate ecological context in our analysis. For example, for the mobbing and transit datasets¹⁵, Jackdaw (*Corvus monedula*) always forms long-term monogamous relationships and both parents contribute to rearing the young, therefore, large numbers of individuals recorded in these datasets include mated pairs, unpaired individuals and juveniles. Indeed, we openly admitted that the impact of pair-bonded relationships on quantifying MS is overlooked in our analysis, but these relationships can significantly influence the measurement of MS. We hope that future research will bridge this gap and provide a more nuanced perception of MS within the context of the study species' ecological and physiological attributes.”

— “great advantages of AMS in shaping the emergence of collective motion”, which seems only supported in the specific evacuation experiments compared to the baselines adopted by the authors;

To be more precise, it could be rephrased as following: the presented swarm experiments of collective following and collective evacuation demonstrated the advantages of AMS in responding to perturbations and enhancing the self-organization of the swarm to achieve a smooth evacuation from confined environments. However, upon reviewing the new version of the Discussion, we have found that these sentences are redundant in this context. Therefore, we have decided to remove this sentence.

— “AMS not only empowers the swarm to promptly respond to the transient perturbation but also strengthens the self-organization of collective motions in terms of temporal cognition”, a claim for which I could not find a precise supporting result.

The term "temporal cognition" in this context refers to the rapid contraction of the swarm that emerges spontaneously during the evacuation. In order to be more precise and to avoid over-interpretation, we revised this point in the revised manuscript (see **main text: page 2, lines 38-40** and **page 15, lines 525-527**):

“AMS not only empowers the swarm to promptly respond to the transient perturbation but also strengthens self-organization of the swarm to evacuate from the confined environment smoothly.”

— “Our findings may also facilitate the design of fully autonomous swarm robotics with artificial neuronal circuits of see-sense-decision-direct, i.e., robots with insect brains”, but given the level of evidence, I’m unsure about which results will facilitate this design beyond what was previously possible.

To more accurately reflect the potential implications of our work, we deleted the discussion of our work related to the artificial neuronal circuits of see-sense-decision-direct. We revised this point in the new manuscript (see **main text: page 15, lines 539-541**):

“We believed that our findings on MS may also ease the burden of perception and motion planning in the design of fully autonomous swarm robotics^{66,73}, fostering the interdisciplinary exchange between biology, physics, and engineering.”

C3. Readability.

Point 1.14: I found the manuscript very difficult to read, overall. There are many complex sentences and paragraphs (e.g., in the introduction), unclear expressions (e. g. “intrinsic interactions”; “subtle mechanism of group organization”, why is the mechanism subtle?) and too general claims. A major revision to improve readability and water down hyperbolic claims is needed.

Response: We thank Reviewer #1 for this critical comment. In the revised manuscript, we have tried to avoid complex sentences, unclear expressions, and too general claims.

Minor issues:

Point 1.15: — The fact that the leader-follower matrix is binarized for the nestedness analysis is quite important in light of Staniczenko et al, 2013, and it should be emphasized in the main text as well, not only in the SM.

Response: We thank Reviewer #1 for this comment. We have added the following sentences in the revised manuscript to address the binarization of LF matrix for computing nestedness (see **main text: page 6, lines 169-170**):

“Note that we calculated the nestedness by binarizing the LF matrix, meaning that we set non-zero elements as 1 and all other elements as 0.”

Point 1.16: — As the MS definition is central to the manuscript’s contribution, it would deserve to be placed in the main text, and thoroughly commented. The definition is provided in Eq. 2, but not extensively motivated against, for example, simpler alternatives.

Response: We thank Reviewer #1 for this insightful comment. Following Reviewer #1's this suggestion and the above constructive comments, we have relocated the definition of MS from the Methods section to the main text, and have rewritten a description and interpretation of MS and MST in the revised manuscript (see **main text: pages 7-8, lines 198-247**).

“Therefore, we proposed the individual perception of motion salience (MS) to measure the relative movement changes of neighbor- j from the focal individual- i 's perception within the period $[t - \tau, t]$ in the form of

$$M_{ij}(t, \tau) = \frac{\angle(\hat{\mathbf{x}}_{ij}(t), \hat{\mathbf{x}}_{ij}(t - \tau))}{\tau} \left(\frac{1 + \hat{\mathbf{v}}_i(t) \cdot \hat{\mathbf{x}}_{ij}(t)}{2} \right)^\alpha \left(\frac{1 + \hat{\mathbf{v}}_i(t - \tau) \cdot \hat{\mathbf{x}}_{ij}(t - \tau)}{2} \right)^\alpha \quad (i \neq j, j \in \mathcal{S}_i, 0 < \tau < t). \quad [1]$$

$\hat{\mathbf{x}}_{ij}(t) = (\mathbf{x}_j(t) - \mathbf{x}_i(t)) / \|\mathbf{x}_j(t) - \mathbf{x}_i(t)\|$, $\mathbf{x}_j(t)$ is individual- j 's position vector at time t , \angle means the angle between two vectors in 3D Cartesian coordination, and \mathcal{S}_i indicates the collection of neighbors of focal- i . \mathcal{S}_i represents all individuals recorded in the flock when calculating $M_{ij}(t, \tau)$. Here τ , being $0 < \tau < t$, means the lag time for individuals i and j at time t , and is also referred to as the perceiving time. In line with a few flock models^{51,52}, τ shares the similar view that the sensory information during an interval is calculated and then updated.

The diagram of definition of motion salience is shown in **Fig.2a**. $M_{ij}(t, \tau)$ comprehensively quantifies the perception of neighbor- j 's motion changes relative to the focal individual- i involving the position and velocity simultaneously during the period $[t - \tau, t]$. Note that MS could capture variations in both azimuth and elevation angles in a 3D context, whereas in a 2D setting, it reduces to solely assessing changes in the azimuth angle. The larger $M_{ij}(t, \tau)$, the more pronounced movement of individual- j felt by individual- i during $[t - \tau, t]$. We incorporated an anisotropic factor $\alpha \geq 0$ in the definition of MS to model the fact of forward-oriented preference in biological perception^{15,53,54} (**Fig.2b**). With increasing α , the ability of individual perceiving movements of its neighbors gradually narrows to the front vision. We emphasized that the definition of MS in this work aims to measure and compare the relative motion changes of multiple neighbors from the context of collective motion cues, which is different from visual selection/search^{45,46} or salient motion detection⁵⁵ in the fields of cognitive science or computer vision.

For a given $\tau > 0$, averaging $M_{ij}(t, \tau)$ over all the time stamps t from the period $[t_s, t_e]$ yields $\langle M_{ij} \rangle(\tau)$. We defined the maximal motion salience time (MST), denoted as τ_{ij}^{MS} , to be the τ value where $\langle M_{ij} \rangle(\tau)$ reaches its maximum. τ_{ij}^{MS} characterizes how long it will take individual- j to reach the maximum of relative motion changes from individual- i 's perception. If τ is shorter than MST, $\langle M_{ij} \rangle(\tau)$ could initially approach the maximum as τ increases. The increment of $\langle M_{ij} \rangle(\tau)$ means the motion difference between two individuals could become more and more pronounced. When τ approaches MST, the motion difference reaches its maximum, after which the movements of the two individuals start to converge (see the curves of $\langle M_{ij} \rangle(\tau)$ as a function of τ in **Fig.2c-e**). Therefore, we assumed that MST can quantify the extent to which the focal individual closely aligns with its neighbors or follows its own drive. To verify this assumption, we calculated the average velocity difference $\Delta v_{ij} = 1 - \langle \mathbf{v}_i(t) \cdot \mathbf{v}_j(t) \rangle$ throughout the entire duration of a flock for a bird pair- (i, j) . A higher value of $\langle \mathbf{v}_i(t) \cdot \mathbf{v}_j(t) \rangle$ indicates a closer alignment between the two individuals, resulting in a smaller average velocity difference Δv_{ij} . For example, Supplementary Fig.23a-f show the trajectories, $\langle M_{ij} \rangle(\tau)$ and temporal velocity difference $\mathbf{v}_i(t) \cdot \mathbf{v}_j(t)$ of pair-(1,6) and (4,7) from the mobbing flock shown in Fig.1a. Our findings reveal that MST of pair-(1,6) is greater than that of pair-(4,7), and correspondingly, Δv_{ij} of pair-(1,6) is also larger than that for pair-(4,7). This suggests that pair-(1,6), with a higher MST, experiences larger velocity difference, while pair-(4,7) displays the opposite pattern compared to pair-(1,6). Thus, for the entire mobbing flock, Supplementary Fig.23j demonstrates that MST and average velocity difference for all bird

pairs display an obvious positive relation, indicating that larger MST corresponds to a bigger velocity difference. Furthermore, we broadened the above correlation analysis to mobbing, circling and transit datasets (Supplementary Fig.24). The findings indicate that the predominant correlations in all three flocking datasets are positive between MST and average velocity difference. This implies that the perception of MS could quantify the trade-off between individual drive and group cohesion in collective motions, because MST provides a measure of the degree to which the focal individual aligns closely with its neighbors or prioritizes its own internal drive.”

Finally, we thank Reviewer #1 again for her/his very insightful and constructive comments/suggestions. We hope our responses above have addressed those very legitimate issues/concerns in a satisfactory manner.

Response to Reviewer #2

Point 2.0: The paper combines the analysis of empirical data of collective motion in bird flocks with a new measurement of interaction between group members, an agent-based model and a swarm robotics experiment; a very interesting combination that can be valuable for many fields of research. The authors present their framework, methods and results clearly, with particularly nice figures. However, given that such an interdisciplinary endeavour is challenging, I think that the authors need to improve in the biological relevance of their text, mostly through the terminology used in the manuscript. I further have some concerns about the statistical analysis of the empirical data and the simplicity of the agent-based model, after the introduction argumentation suggesting that the collective motion algorithms need to include a higher degree of realism. Overall I enjoyed the paper and I hope my comments will help to improve it and prepare it for a large interdisciplinary audience, please find them below.

Response: We thank Reviewer #2 very much for reviewing our work and her/his positive assessment of interdisciplinary impact of our framework. We next address each of the reviewer's very constructive comments/suggestions in order.

Main comments:

Point 2.1: - Lines 50-58: I agree with the authors that vision specifics and perceptions are not often taken into account, but I find this first part a bit unclear, perhaps too technical for the introduction (especially 56-58), and not very convincing for the need of a new approach. Given that a model should reduce the complexity of the system it aims to study, so some realism will be sacrificed in the process, perhaps the authors can revise this part focusing on what the problem is (e.g., why is something problematic for flocks with highly maneuverable motion?) instead of just describing the assumptions of existing models.

Response: We thank Reviewer #2 for this constructive comment and suggestion. Here, we have simplified the technical description of the challenges related to vision perception in flocking models. Instead, we have emphasized the problem of accurately reproducing collective maneuvers in flocking models and subsequently adapting them for use in swarm robotics. In the revised manuscript, we have rewritten the second paragraph of Introduction section (see **main text: pages 3-4, lines 49-85**):

“Despite the explanatory success at the macro-behavioral level, the proposed topological, metric, or plastic interactions are *phenomenological* in essence. *First*, visual perception as the primary medium for most animals to response to conspecifics and environments, is highly specialized for gaze and saccade to process the sensory information (including size, color, orientation, motion, etc)¹⁷. There is a widely accepted consensus that the perception plays a crucial role in governing the local behavioral interactions among individuals, and further fundamentally shapes group self-organization and collective dynamics¹⁷. However, the majority of flocking models circumvent the specificity of visual perception at the initial stages of modeling. Instead, these models tend to favor idealized perceptions and phenomenological interactions. As an example, they do not explicitly incorporate the perception stream to process the sensory signals perceived from neighbors' motions but rather feed neighbors' coordinates and velocities directly to synthesize reactive actions. It is not biologically plausible and suffers from redundant inputs, effective information dilution¹⁸, and sensory overload¹⁹. *Second*, the animal flocks always emerge some sophisticated and highly maneuverable motions in the fields. For example, starling flocks display the so-called wave of agitation or shimmering waves to reduce risk of predation²⁰, and fish slow down to avoid collisions whereas birds fly at low variability of speed and lose altitude during turning²¹. However, the

physics-oriented Vicsek-like models are not capable of accurately describing these motion characteristics, as they rely on a simplified rule of averaging-alignment that is based on computational theories rather than biological evidence. Moreover, despite the inclusion of vision-mediated coordination in the flocking models^{22,23,24,25}, they still disregard the context and temporal cues of neighbors' motion. This is because these models rely on linear or Boolean-like visual projections, which result in transient perception outcomes that only capture the geometric contours from visual imaging. As a consequence, these models still face challenges when it comes to accurately reproducing collective maneuvers. *Third*, with an applied science standpoint, existing mechanical or visual projection models haven't facilitated the adoption of bio-inspired mechanisms for the applications of swarm robotics^{26,27}. Indeed, most of the theoretical models typically prescribe the position-based and averaging-type interactions to yield the phenomenological alignment, attraction, and repulsion^{28,29,30,31}, while practical applications require more sophisticated and diversified group adaptability across contexts, e.g., collective anti-predators¹⁵, collective turn³², cooperative transportation³³ or excavation³⁴, collective chase or escape^{35,36,37}, etc. Obviously, for accommodating different kinds of collective tasks, designing bio-inspired swarm robotics is inextricably linked to revealing the fundamental mechanisms underlying different collective patterns, dynamics, and functions from real animal flocks. Furthermore, beyond model organisms and numerical simulations, physical swarm robotics offers an arbitrarily customizable test-bed that combines physical realism with convenient analyzability³⁸. This allows for the exploration of the inherent connections between group-level organization and individual-level attributes such as perception, decision-making, and kinematics.”

Point 2.2: - Line 58-64: I am not sure I understand what the authors mean in their second part. There are models with realistic motion characteristics and their effect on the emergent properties of the group are of course very important. For example, in reference of the manuscript 27 individual variation in speed and accelerating attraction in pigeons gives the more wide flock formation and lead-follower relations, in [1] the banging behaviour of the birds while turning is found to underlie the agitation waves in starlings, and in [2] the shape and internal structure of fish and birds is explained by locomotion properties. So perhaps the authors emphasize more in physics-oriented Vicsek-like models that are used to inspire swarm robotics algorithms and are often quite simplistic in such characteristics?

Response: We thank Reviewer #2 for this constructive comment, and agree with Reviewer #2 that this part is misleading for audiences. In the revised manuscript, we have made revisions to ensure greater clarity (see **main text: page 3, lines 61-67**):

“*Second*, the animal flocks always emerge some sophisticated and highly maneuverable motions in the fields. For example, starling flocks display the so-called wave of agitation or shimmering waves to reduce risk of predation²⁰, and fish slow down to avoid collisions whereas birds fly at low variability of speed and lose altitude during turning²¹. However, the physics-oriented Vicsek-like models are not capable of accurately describing these motion characteristics, as they rely on a simplified rule of averaging-alignment that is based on computational theories rather than biological evidence.”

Point 2.3: - Following the previous comment, the authors do not include realistic motion characteristics in their model.

Response: We thank Reviewer #2 for this critical comment. While we acknowledged that our modeling framework does not incorporate the realistic motion characteristics of birds, MS provides a comprehensive quantification of the perception of neighbors' motion changes. For example, it encompasses the changes in position, heading, speed and acceleration over a

given period. We believed that one of the advantages of MS is its ability to bridge perception and motion, eliminating the necessity for a detailed portrayal of the specific collective motions observed in certain species, such as the intricate flight dynamics of birds. As a result, MS offers a versatile approach that can be applied to different types of collective tasks, making it highly adaptable and generic. Thus, we have addressed this point in the revised Discussion (see **main text: page 15, lines 520-524**):

“Besides, we believed that one of the advantages of MS is its ability to bridge perception and motion, eliminating the necessity for a detailed portrayal of the specific collective motions observed in certain species, such as the intricate flight dynamics of birds. As a result, MS offers a versatile approach that can be applied to different types of collective tasks, making it highly adaptable and generic.”

Point 2.4: - Line 77-79: There is quite some variability in the field of collective animal behaviour about what leadership means. Given the interdisciplinary audience of the journal, I think the authors should first clarify that. Leadership in flocks usually refers to emergent leadership that is linked to the frontal positioning of individuals in the flock, not biased interactions that make some individuals more influential than others given for instance an underlying social structure (that exists in some species).

Response: We apologize for not making this point clear in the previous manuscript. Based on Reviewer #2's suggestions, we have clarified the definition of leadership where we first mentioned the concept (see **main text: page 4, lines 89-94**):

“Leveraging recent advances in the leader-follower (LF) relationship analysis¹² and the structure analysis of complex systems³⁹⁻⁴³, we aimed to unveil the motion characteristic needed by an individual to lead the flock. (Note that LF relationship in this study refers to emergent leadership that arises from the consensus of velocity among pairs of individuals within the flock, not the underlying social structure that make some individuals more influential than others.)”

Point 2.5: - Lines 80-81: This comment refers to the wording in many parts of the manuscript. The wording that the flocks ‘prefer’ is misleading (also in the abstract with ‘prefer to evolve’), the flock as a collective is a self-organized system with no wishes. The authors should revise this throughout. Same applies for line 29: the flocks do not ‘choose’ a strategy or organize its members.

Response: We thank Reviewer #2 for this comment. Based on her/his suggestions, we have revised this term throughout the main text and Supplementary Information:

- see **main text: page 2, lines 26-28**:

“We find that flocks with higher maneuverable motions (i.e., *mobbing* and *circling*) **tend to exhibit** a more nested structure of leader-follower (LF) relations and a clear hierarchy to mitigate the damage of individual **drive** to group cohesion.”

- see **main text: page 3, lines 94-96**:

“*First*, we found that flocks with higher maneuverable motions (e.g., *mobbing* and *circling*) **tend to exhibit** a more nested structure of LF relations and a clear hierarchy to maintain group cohesion.”

Point 2.6: - Since the data are from stable camera frames, the author should mention in the main text the average duration of the tracks in each dataset (how much data on average per individual) and the sampling frequency. From figure 1a-c there seems to be duration of 10, 5, and 2 seconds respectively. Also, they should mention the average group size of the flocks and how much variation there is in group size across datasets.

Response: We apologize for not making this point clear in the previous manuscript. The statistics of some basic flocking information, e.g., group size, duration of flocks, and other metric of collective motions could be found in Supplementary Sec.3 and Supplementary Figs.11-13. Also, Supplementary Sec.1 introduces the data collection and processing of three datasets, and Supplementary Figs.2-10 display the overview of flocking trajectories. To highlight this point, we revised the manuscript accordingly (see **main text: page 5, lines 121-127**):

“In this work, we totally got 140, 94, and 1483 tracks of mobbing, circling, and transit flocks, respectively (see Supplementary Figs.2-10 for the overview of flocking trajectories). Supplementary Fig.11 counts the flock size and flocking time of all flocks analyzed in this work. Besides, we used four metrics, e.g., group order, trajectory curvature, group density and instability of neighbor to systematically measure the properties of collective motions (see Supplementary Sec.3 for the detailed definitions and results).”

Point 2.7: - Since the duration of a ‘flock’ in transit seems from figure 1 to be much smaller than the others, can this be the reason behind the LF matrix not displaying any hierarchy?

Response: We thank Reviewer #2 for this very legitimate concern. Here **Fig.R16** (in the end of response letter) displays the scatter plot between nestedness of LF networks and flocking time for all Transit flocks. We could see that the nestedness of LF networks from 94 transit flocks occupy the whole range of [0,1] (y-axis of **Fig.R16**). Some transit flocks with the similar duration (e.g., ~2 seconds) exhibit a mix of both high and low levels of nestedness. Therefore, the result does not support the notion that transit flocks lack a hierarchy of LF networks solely due to their shorter flocking duration.

Point 2.8: - I find that the statistical analysis of the empirical data is not enough to provide support for the findings. Figure 1 shows quite some variation with flock id. The authors should thus test for the effect of group size and random effect of id (or track duration), for instance concerning the relation between average order and nestedness (lines 152-154).

Response: We thank Reviewer #2 for this very critical comment. In Fig.1g-i, we collected flocks with different id together to calculate the relations between LF structure and group order. To support our findings, followed reviewer’s suggestions,

- (1) we calculated the relations from each flock id (see **Fig.R17**);
- (2) according to group size, we divided the flocks into three parts and then to detect the relations between their nestedness of LF networks and average order (see **Fig.R18**);
- (3) according to flocking time (or duration), we divided the flocks into three parts and then detected their relations (see **Fig.R19**).

Three new results support our findings that the positive relation between nestedness of LF networks and group order could be obviously observed in mobbing and circling datasets, while the relation is flat in transit datasets.

To strength this finding, we add the above results to the revised manuscript (see **main text: page 6, lines 179-181**):

“Besides, these findings are consistently validated across every separate flock record (a collection of some tracks, e.g., mobbing-01), and different group size and flocking time (see Supplementary Figs.20-22).”

Point 2.9: - Lines 171- 172: The time-lag or perceiving time that the authors propose in the MS model, if I am not mistaken, is similar with what the models of [3] and [4] refer to as update or

reaction time. The novelty of this work is that this varies per pair of interacting neighbours. Is this the case? I think this should anyway be clarified in the methods or discussion.

Response: We thank Reviewer #2 for very insightful comment and provide these references. Yes, the perceiving time τ in our work refers to the time window between two birds. Our definition and Ref.[Bode et al., 2010, Papadopoulou et al., 2023] share the similar view that the sensory information during an interval is calculated and then updated. In order to help readers better understanding the perception time, we added these two important references in the revised Method part (see **main text: page 7, lines 207-209**):

“Here τ , being $0 < \tau < t$, means the lag time for individuals i and j at time t , and is also referred to as the perceiving time. **In line with a few flock models^{51,52}, τ shares the similar view that the sensory information during an interval is calculated and then updated.**”

Point 2.10: - Lines 181: I find the MST a very interesting measurement. I however disagree with the ‘flying freedom’ mentioning; it flags more the tolerance of the focal individual to not stay close to its flock mates or follow its own drive. I do think that replacing the word ‘freedom’ throughout the manuscript with individual drive or something equivalent will improve the clarity of the results.

Response: We thank Reviewer #2 for the positive comment and very constructive suggestion. We fully agreed with reviewer that the statement, e.g., ‘flying freedom’ or ‘individual freedom’, is very anthropomorphic, and is inappropriate in the context of our manuscript. We have revised the notion of MST in the revised manuscript (see **main text: pages 7-8, lines 229-230**):

“**Therefore, we assumed that MST can quantify the extent to which the focal individual closely aligns with its neighbors or follows its own drive.**”

And we replaced previous “freedom/ individual freedom/flying freedom” with “drive/individual drive” throughout the main text and Supplementary Information. We appreciate Reviewer #2’ valuable input to express our key view of MST with greater precision and clarity.

Point 2.11: - Lines 193: Isn’t that an effect of the nature of the behaviour? If the threshold to react to difference in positioning is larger than what the birds do during the few seconds of transit in that dataset, wouldn’t then MST be very small?

Response: We thank Reviewer #2 for the critical comments. We fully agreed that transit flocks exhibit the very small MST perhaps due to two facts: (i) the group is very ordered and individuals possibly have the same drive to a common destination, and (ii) the duration of transit flocks (range is 0.5~4.5 seconds) is too short, which is less than the threshold of responding to the relative motion changes from its neighbors. Therefore, we replenished this point in the revised manuscript (see **main text: page 8, lines 248-255**):

“**For any flock, within a time period $[t_s, t_e]$ we could construct the MST matrix. Next we took the average of all the elements in MST matrix to evaluate the degree of individual drive for a flock. We found that the mobbing and circling flocks demonstrate significantly higher average MST than that of transit flocks (p -value=1.46e-17 and 2.39e-16, respectively, Mann–Whitney U-test, see y-axis of **Fig.2f-h**). Note that the transit flocks exhibit very small MST perhaps due to two facts: (i) the group is very ordered and individuals possibly have the same drive to a common destination, and (ii) the duration of transit flocks is too short compared with another two datasets, perhaps it is less than the threshold of responding to the relative motion changes from its neighbors.**”

Point 2.12: Also for lines 210-212: the initial conditions can also play an important role, if the group is very ordered and with a common destination, individuals have the same drive. In mobbing

and circling turns individuals need to stay together while individually deciding to turn, and since it is a fast phenomenon, they cannot instantaneously react to one another. Given that individuals need time to perceive their neighbours, if I understand correctly, if the neighbour doesn't move much, MST will be small, but if it does, for instance when turning, the MST will be biased having a minimum time that matches the cognitive and locomotion abilities of each individual. Perfect coordination and order is extremely difficult and expensive while flying.

Response: We thank Reviewer #2 for the very constructive comments. Yes, we realized this point and mentioned in the Methods section of previous version. Since Reviewer #1 also raised the similar concerns about definition and interpretation of MST (**Point 1.5** and **Point 1.6**), we have refined our analysis and rewritten the interpretation of MST in the revised manuscript. For detailed information, please see our responses **Point 1.5** and **Point 1.6**.

Point 2.13: For lines 213-221: I disagree with the interpretation of the authors here. The authors do not take into account the ecological context of each behaviour. Especially the sacrificing freedom statement is very anthropomorphic; as I mention above, foraging individuals know where they are going and they are already in an ordered flock. They don't have an individual drive to change their motion, and the density of the flock I would expect to be stable. During collective turns, the density and shape of the group changes, and thus individuals may need to turn to avoid collisions more often. This is a 'disadvantage', rather than individual freedom. The explanation of the flock maintaining manoeuvrable motion is thus confusing.

Having said these the authors could measure group density and diffusion (such as global, mutual diffusion or stability of neighbours [5]) across datasets. They can also measure turning rate in the mobbing and circling flights and see whether that affects MST and order, given that it has been shown to affect collective motion dynamics [4]. If for instance the reshuffling rate is high or the group is denser, individuals have to avoid each other more.

Response: We thank Reviewer #2 for the very critical comments. According to reviewer's suggestions, three points consummated the empirical analysis of motion characteristic and MST:

First, we used four flocking metrics, e.g., group order, trajectory curvature, group density and instability of neighbor [Papadopoulou et al., 2023] to systematically measure the properties of collective motions. **Fig.R20** shows four temporal metrics for the mobbing and transit flock shown in Fig.1 as examples. Overall, these metrics successfully quantify the highly maneuverable or smooth motions. Interestingly, we found that the group density does not show any difference across three datasets, regardless of whether the flocks exhibit collective turns or flying smoothly (**Fig.R21c**). However, for the instability of neighbors [Papadopoulou et al., 2023], the transit is significantly lower than that of mobbing and circling flocks (**Fig.R21d**). The result is consistent with the reviewer that the neighbor structure in transit flocks is much more stable than the behaviors of collective turns. Since the shape of flock changes frequently during collective turns, individuals may need to turn to avoid collisions more often. Oppositely, those foraging individuals know where they are going and they are already in an ordered flock.

Second, we tested the relations between different flocking measurement across three datasets. The relation between trajectory curvature and group order accords with our common sense and the observed collective behaviors across three datasets: the higher group order, the less trajectory curvature (**Fig.R22**).

As instability of neighbors measures the changes of neighbor structure over time, we focus on analysis of the relations between instability of neighbors and other flocking metrics that describes motion characteristic (**Fig.R23**). We found that in the circling and transit dataset, the instability of neighbors decreases with the increment of group order (negative trend in **Fig.R23a**), but the relation is not clear in mobbing flocks. Moreover, instability of neighbors grows with trajectory curvature (positive trend in **Fig.R23b**) in mobbing and circling datasets, but it is not clear in transit flocks. Three datasets all demonstrate that instability of neighbors increases with the increment of group density (positive trend in **Fig.R23c**), that is, the flock is denser, the neighbors of a focal individual change more frequently.

Based on the above empirical analysis, we concluded that: (1) irrespective of the collective motion's maneuverability or smoothness, an increase in group density leads to frequent changes in neighbors. This phenomenon could be attributed to the necessity for individuals to actively avoid collisions in dense environments; (2) flocks characterized by highly maneuverable motion exhibit a distinct tendency that the bigger trajectory curvature, the more instability of neighbors for a focal individual. Interestingly, this trend does not manifest in transit flocks.

We in detail introduced these metrics in the revised Supplementary Sec.3 (see **SI: pages 5-7** and **Supplementary Figs.11-15**). In the main txt, we briefly introduced these flocking metrics at the beginning of 'Empirical data analysis' part (see **main text: page 5, lines 124-128**):

“Besides, we used four metrics, e.g., group order, trajectory curvature, group density and instability of neighbor to systematically measure the properties of collective motions (see Supplementary Sec.3 for the detailed definitions and results). Here the instability of neighbors⁴⁴ is estimated by measuring how much the set of nearest neighbors of a focal individual changes over time.”

Thanks to reviewer's suggestion, we comprehensively detected the effect of four flocking metrics (e.g., group order, trajectory curvature, group density and instability of neighbor) on MST. **Fig.R24b** shows that three flocking datasets all emerge the positive relations between average MST and average instability of neighbors. Even in the transit flock, average MST increases with the increment of average instability of neighbors. It indicates that if the neighbors change frequently, it could increase MST for the focal individual. However, across three datasets, we could not detect the distinct relation between average MST and trajectory curvature (**Fig.R24c**) / group density (**Fig.R24d**).

Overall, we appreciate Reviewer #2 again for the critical comment about understanding and interpretation of MST. This helps us realize we do not take into account the ecological context of collective behaviors in the interpretation of MST and make the statement in the previous manuscript very anthropomorphic. After the above comprehensive analysis of motion metrics and their effects on MST, we have rephrased the explanation of MST in the revised manuscript (see **main text: pages 8-9, lines 256-271**):

“To gain a deeper understanding of the impact of motion properties on MST, we conducted a correlation analysis to ascertain the tendency between four flocking metrics (e.g., group order, trajectory curvature, group density and instability of neighbor) on average MST. In both mobbing and circling flocks, there is a notable downward trend observed in average MST as average order increases (**Fig.2f,g**). In contrast, the majority of transit flocks (average order > 0.95 in x-axis of **Fig.2h**) do not exhibit a distinct negative relation (**Fig.2h**). It demonstrates that for the flocks with highly maneuverable motions (e.g., mobbing and circling), the smaller the average MST, the smaller the individual drive, and the higher the group order.

Besides, all three flocking datasets exhibit positive relations between average MST and average instability of neighbors (Supplementary Fig.25a). Interestingly, there is an increasing trend even in

the transit dataset (Supplementary Fig.25a). This observation suggests that for the focal individual, frequent changes in neighbors could potentially lead to an increase of MST. However, across three datasets, we could not detect the distinct relation between average MST and trajectory curvature/group density (Supplementary Fig.25b,c). If we consider the anisotropic effect of motion perception ($\alpha > 0$), the relations between average MST and four flocking metrics are consistent with the case of $\alpha = 0$ (Supplementary Fig.26,27).”

Point 2.14: - Lines 158-159: does the LF structure and hierarchy lead to high group order, or the hierarchy emerges because of the order?

Response: We apologize for not making this point clear in the previous version of our manuscript. The relations between nestedness of LF networks and group order on three flocking datasets demonstrated that mobbing and circling flocks display the positive correlation, while the relation is flat in transit flocks. Technically, we admitted that the relations between nestedness and group order are correlation rather than causal inference. Besides, this concern depends on whether the collective motions are highly maneuverable or smooth. For example, for flocks with maneuverable motions, such as mobbing or circling, we could draw the conclusion that the clear LF structure and hierarchy (equals to higher nestedness of LF networks) lead to high group order, equally, the hierarchy in LF networks emerges when the flocks with highly maneuverable motion have high group order. Unfortunately, the relation between LF structure and group order in transit flocks (smooth motion pattern) is still not clear.

To express this point clearly, we have revised the manuscript as following (see **main text: page 6, lines 184-189**):

“Overall, the comparison in three datasets demonstrates that the mobbing and circling flocks tend to have a clearer hierarchy of LF relations than that of transit flocks. For mobbing or circling flocks, we could draw the conclusion that the clear LF structure and hierarchy lead to high group order, equally, the hierarchy in LF networks emerges when the flocks with highly maneuverable motion have high group order. Unfortunately, the relation between LF structure and group order in transit flocks is still not clear.”

Point 2.15: - Lines 134-135: it will be helpful to repeat here which species these result cover, especially because of the comparison with pigeons.

Response: We thank Reviewer #2 for this insightful suggestion. We have revised the sentence as following (see **main text: page 6, lines 157-159**):

“This finding observed in the jackdaws (*Corvus monedul*) and swifts (*Chaetura pelagica*) flocks is consistent with previous findings in pigeon flocks¹².”

Point 2.16: - How does leadership links to relative positions in the group? This should be very clear in the results given previous research in the field. Especially in [6], they show the position of neighbours initiating each transit and mobbing turn. It will be very interesting to see whether the new approach matches or not the previous findings, in particularly because it is the same dataset.

Response: We thank Reviewer #2 for the insightful comment. In Ref.[Ling et al., Journal of the Royal Society Interface, 2019], they calculated the time lag τ_{ij} between pairs of birds to assign the scores $W_i = \sum w_{ij}$ of each bird in the group, where $w_{ij} = -1$ if $\tau_{ij} > 0$ or $w_{ij} = 1$ if $\tau_{ij} < 0$. They defined the rank of each bird by the score W_i , i.e., the higher the score, the higher the rank of the bird, which indicated the leadership of turning initiation within the group (rank in **Fig.R25c**). For the approach proposed in our work, we calculated the local reaching

centrality to quantify the leading tier (L_i in **Fig.R25b**) of each bird in an LF network. To evaluate whether the new approach matches the previous findings, we conducted the correlation analysis between the outcomes of these two approaches within the same trajectory shown in Fig.1a. As shown in **Fig.R25**, for a given period of trajectory (highlighted by red in panel a), we found a strong correlation in the outcomes produced by these two approaches (**Fig.R25d**). The result suggests the concordance of our approach with previous finding described in Ref.[Ling et al., Journal of the Royal Society Interface, 2019] for assessing each bird's leadership in the group.

Point 2.17: - Lines 242-243: what does the correlation analysis performed only in these two flocks first and then to the datasets?

Response: We thank Reviewer #2 for her/his careful review, and apologize for not making this point clear in the previous version of our manuscript. To reach the logical consistency of figure presentation among Fig.1,2,3 in the main text, we first showed the analysis results from mobbing, circling and transit flocks. Three flocks were shown in Fig.1a-c. In Fig.3, panel a shows the correlation analysis between LF and MS with a mobbing flock as an input; panels b and c show the results of correlation analysis when τ and t take different values for three flocks. The results of correlation analysis performed in three datasets could be found at Fig.3d1-3d3 in the main text.

To express clearly, we revised the sentence as following (see **main text: page 9, lines 292-299**):

“Compared with the mobbing flock (shown in Fig.1a), we performed the same correlation analysis for circling and transit flocks shown in Fig.1b,c (**Fig.3b,c**). An intriguing observation from our study is that when considering the forward-oriented preference of visual perception ($\alpha = 1$), both mobbing and circling flocks, known for their highly maneuverable motions, exhibit the majority of positive correlations between LF and MS across various combinations of t and τ (**Fig.3c1,c2**). This suggests that individuals with higher MS tend to play the higher-tier leading role for the majority of a flock's duration.”

Point 2.18: - Figure 1, e and f: how are the axis of these plots ordered?

Response: We apologize for not making this point clear in the previous version of our manuscript. The axis of Fig.1e,f are ordered by the nestedness of LF networks. The detailed information about how to calculate the nestedness could be found in Supplementary Sec.4. We added this point in the caption of Fig.1 (see **main text: page 25, lines 882-883**).

“The axis in panels d-f is ordered by the nestedness of their corresponding LF networks.”

Point 2.19: - Lines 278-279: I am not sure what this sentence means, the authors should clarify it.

Response: We apologize for not making this point clear in the previous version of our manuscript. Here we aim to demonstrate that AMS could generate more maneuverable motions than that of ATHD within the same period of time, since AMS generates two peaks of trajectory curvature (Fig.4d) while there only exists one peak in ATHD (Fig.4e). We revised this sentence to clarify the phenomenon (see **main text: page 11, lines 357-360**):

“Interestingly, we found that within the same period of time, the curvature of trajectories generated by AMS (two peaks in **Fig.4d**) is obviously larger than that of ATHD (one peak in **Fig.4e**). It indicates that AMS could generate more maneuverable motions than that of ATHD.”

Point 2.20: - Is the model two dimensional? Mobbing flights are quite expanded in altitude as seen in the original paper that the data are from. What do the authors think the effect of this difference may be, since avoidance of neighbours is exaggerated in 2D models?

Response: We thank Reviewer #2 for the insightful suggestion. In this work, the swarm models are 3D in simulations (Fig.4) and 2D in swarm robotics application (Figs.5-6). Here we replenished the results of swarm models in 2D (**Fig.R26**). Interestingly, we could not observe any differences between 3D and 2D from the results of trajectory curvature (**Fig.R26d-e**), nestedness distribution (**Fig.R26g**) and correlation between LF and MS (**Fig.R26h**).

We thought that the independence of movement dimension in AMS could be attributed to the integration of relative motion changes among neighbors over a defined time interval (τ) by the metric MS, as expressed below (same as Eq.(1) in the main text):

$$M_{ij}(t, \tau) = \frac{\angle(\hat{\mathbf{x}}_{ij}(t), \hat{\mathbf{x}}_{ij}(t - \tau))}{\tau} \left(\frac{1 + \hat{\mathbf{v}}_i(t) \cdot \hat{\mathbf{x}}_{ij}(t)}{2} \right)^\alpha \left(\frac{1 + \hat{\mathbf{v}}_i(t - \tau) \cdot \hat{\mathbf{x}}_{ij}(t - \tau)}{2} \right)^\alpha$$

$(i \neq j, j \in \mathcal{S}_i, 0 < \tau < t).$

This integration includes both changes in velocity and changes in the angle between the two vectors ($\hat{\mathbf{x}}_{ij}(t)$ and $\hat{\mathbf{x}}_{ij}(t - \tau)$). As an illustration, the metric MS captures variations in both azimuth and elevation angles in a 3D context, whereas in a 2D setting, it reduces to solely assessing changes in the azimuth angle in a plane. Therefore, for collective motions with the same setting, AMS could quantify the perception of motion salience in both 3D and 2D.

To clearly address this point, we added the below sentences to the revised manuscript (see **main text: page 11, lines 366-368**):

“Note that MS could capture variations in both azimuth and elevation angles in a 3D context, whereas in a 2D setting, it reduces to solely assessing changes in the azimuth angle in a plane. The results simulated in 2D are consistent with that in 3D (Supplementary Fig.38).”

Point 2.21: - In the swarm application the authors exclude the blind area, while they showed earlier that any relation between MS and LF depends on a being > 0 . The authors should better support this decision and any effect of it.

Response: We thank Reviewer #2 for the insightful comment. In **Fig.4g,h** and **Fig.R26**, the simulation results show that regardless of the absence ($\alpha = 0$) or presence ($\alpha = 1$) of blind area, there are consistently positive correlations dominating the relation between MS and LF. However, in the real bird flocks, the positive correlations depend on $\alpha > 0$. This difference in simulations and empirical data analysis could be attributed to that: the flight status, locomotor ability, and flight dynamics of birds impose constraints that prevent them from achieving a perception free of blind spots. As a result, birds exhibit a preference for forward-oriented perception [Vallortigara et al, Current Biology, 2001, [DOI: 10.1016/S0960-9822\(00\)00027-0](https://doi.org/10.1016/S0960-9822(00)00027-0)][Fernández-Juricic, Trends in Ecology & Evolution, 2004, [DOI: 10.1016/j.tree.2003.10.003](https://doi.org/10.1016/j.tree.2003.10.003)], and Ref.[Pettit et al, Current Biology, 2015, [DOI: 10.1016/j.cub.2015.10.044](https://doi.org/10.1016/j.cub.2015.10.044)] reported a tendency for leaders to fly at the front of the flock.

To support this in the bird flocking datasets, we conducted the **spatial structure of the leading position** to investigate the relations between the leading tier and fraction of neighbors present in the front view (reported in Supplementary Sec.5). The example of analyzing spatial structure of the leading position was showed in Supplementary Fig.28. Our findings reveal a negative relationship between the leading tier and the fraction of neighbors present in the front view within mobbing and circling flocks (Supplementary Fig.29). In other words, individuals

with lower values of the leading tier tend to have higher values of the fraction of neighbors present in the front view. This suggests that individuals occupying a higher leading tier are more prone to being positioned at the front of the flock to lead the group. On the contrary, the transit flocks do not show any preference between the leading tier and neighbors' relative position.

Swarm simulations and miniature two-wheel differential mobile robots utilized in swarm robotics applications often have minimal or no movement restrictions, providing increased flexibility and freedom of motion. Thus, we ignored to consider the blind area of perception. We apologize for not making this clear in the previous version of our manuscript. Based on Reviewer #2's suggestion, we have added the following sentences in the revised Supplementary Sec.8 (see **SI: page 14**):

“In Fig.4g,h and Supplementary Fig.38, the simulation results showed that regardless of the absence ($\alpha = 0$) or presence ($\alpha = 1$) of blind area, there are consistently positive correlations dominating the relation between MS and LF. However, in the real bird flocks, the positive correlations depend on $\alpha > 0$. This difference in simulations and empirical data analysis could be attributed to that: the flight status, locomotor ability, and flight dynamics of birds impose constraints that prevent them from achieving a perception free of blind spots. As a result, birds exhibit a preference for forward-oriented perception, and Ref.[10] reported a tendency for leaders to fly at the front of the flock. Swarm simulations and miniature two-wheel differential mobile robots utilized in swarm robotics applications often have minimal or no movement restrictions, providing increased flexibility and freedom of motion. Thus, we ignored to consider the blind area of perception and set $\alpha = 0$ in the collective following and collective evacuation experiments.”

Point 2.22: - The authors find that AMS can lead to the emergence of leadership, but they define a leader in their simulations and check its performance. Doesn't this show that AMS can facilitate leading with fewer number of informed individuals [7], rather than being responsible for the emergence of stable leader networks? Perhaps a discussion on this with previous findings on leadership in agent-based models would be useful.

Response: We thank Reviewer #2 for the insightful comment. To further investigate this point, we added a tunable parameter to the simulation results of Fig.4: the number of individuals leading the group. **Fig.R27** shows the curves of average curvature of flocking trajectory and nestedness with the increment of the number of individuals leading the group.

The results from **Fig.R27a** indicates that:

- three interactions used in swarm models show the increment of average curvature of flocking tracks as the leading individuals in the group increases.
- with an increasing number of individuals leading the group, the average curvature of flocking trajectories for ATHD and Average approaches that of AMS.

If we recognized the average curvature as a kind of performance for the simulation setting of Fig.4, the augmentation of leading individuals enhances group performance; however, despite this improvement, AMS remains the most effective, while the performance of Average and ATHD only exhibit a tendency towards convergence with AMS.

For the results of nestedness, **Fig.R27b** indicates that:

- when the number of leading individuals in a group becomes excessive, it results in a reduction in the nestedness of LF networks. This observation aligns with our intuitive understanding that an abundance of leading individuals within a group diminishes the unambiguity of hierarchical relationships in the LF networks.

- Interestingly, in Average and ATHD, the increment of leading individuals initially results in an increase in the nestedness of LF networks, followed by a subsequent decrease. This suggests that in the case of these two interactions, a smaller addition of leaders could potentially enhance the hierarchical structure of LF networks.
- Compared with another two interactions, even with a small number of leading individuals, AMS could maintain a high level of nestedness and a clearer hierarchical structure of LF network.

Thanks to reviewer's suggestion, the new results in **Fig.R27** demonstrate that AMS not only enables effective leadership with a smaller number of informed individuals but also plays a pivotal role in the emergence of hierarchical structures of LF networks. To clearly address this point, we added the below sentences to the revised manuscript (see **main text: page 11, lines 368-373**):

“Moreover, we also incorporated a tunable parameter of the number of individuals leading the group to investigate the effectiveness of leadership in the flocking models (see detailed information in Supplementary Sec.7). The findings indicate that AMS not only facilitates the effective leadership with a smaller number of informed individuals but also plays a crucial role in the emergence of hierarchical structures in LF networks.”

Point 2.23: - The failure of the average algorithm in the simulations may be due to the time that following robots are given to follow or not. Flocks in nature can continuously align with each other and slowly make the flock turn, but if the change is instantaneous or very fast, the group doesn't have time to respond. I think simulations with restricted angular velocity of the leader can clarify whether this is the case.

Response: We thank Reviewer #2 for the insightful comment. Consider the scenario of restricted angular velocity, we set two kinds of collective following simulations: (i) the informed agent and others share the same max angular velocity (**Fig.R28a**); (ii) max angular velocity of the others is fixed at 10×1.91 rad/s but we changed the informed agent with different max angular velocity (**Fig.R28b**). Note that as the miniature mobile robot is constrained by the performance of the motor and battery, the maximum angular rate $\omega_{\max} = k * 1.91$ rad/s and k is an integer number between 1 and 10. In the experiments and physical simulations of two swarm robotics applications, we set $\omega_{\max} = 10 * 1.91$ rad/s for all robots (reported in the Methods and Supplementary Tab.1).

From two kinds of collective following simulation results with restricted angular velocity, we found that:

- For the informed agent and others share the same max angular velocity (**Fig.R28a**), the performance of collective response (R) does not change no matter what the max angular velocity is high or low. The factor that exerts the greatest influence on performance is still the flock size.
- If max angular velocity of the others is fixed at 10×1.91 rad/s but we increase max angular velocity of the informed agent from 2×1.91 to 10×1.91 (see x -axis in **Fig.R28b**), the R value indicates very slight increment for AMS and ATHD, but the increment is obvious for Average especially when the flock size is large (e.g., $N = 70,90$). Overall, AMS and ATHD exhibit insensitivity to variations in the max angular velocity between leaders and the others in the group. This implies that regardless of whether the difference in maximum angular velocity between leaders and others is larger or smaller, the collective response performance from AMS and ATHD stabilizes at a good level. However, the performance from Average interaction deteriorates progressively as the

difference in maximum angular velocity between leaders and the rest of the flock decreases.

Of course, the above findings only work when the group size is larger than 30 (**Fig.R28**). When the flock size is less than 20, the performance from three kinds of swarm interactions displays no difference. Hence, the simulations of collective following with constrained angular velocity reaffirm the robustness of AMS in ensuring optimal performance in collective response.

To clearly address this point, we added the results of restricted angular velocity to the Supplementary Sec.8.1 (see **SI: page 16**).

Point 2.24: - Lines 351-353: it would be interesting for the self-organized mechanism behind this to be explained, since the rules do not change.

Response: We thank Reviewer #2 for the comment. In the collective evacuation experiments, we observed that AMS facilitates the evacuation of the entire group through a narrow gate with the shortest time, even when there were individuals moving in the opposite direction of the exit. We compared 5 parts

$$\mathbf{v}_i(t + dt) = (1 - \delta_t)(\hat{\mathbf{v}}_{al,i} + \hat{\mathbf{v}}_{rep,i}^a) + \hat{\mathbf{v}}_{g,i} + \delta_t(\hat{\mathbf{v}}_{col,i}^a + \hat{\mathbf{v}}_{col,i}^w)$$

in the swarm model utilized for collective evacuation. As we know that the other four parts of velocity components except $\hat{\mathbf{v}}_{al,i}$ are same for AMS, ATHD and Average, we focused on comparing the contribution of three kinds of $\hat{\mathbf{v}}_{al,i}$ to \mathbf{v}_i .

As this concern is closely related to **Point 1.10** raised by Reviewer #1, please see our response **Point 1.10** for details.

Point 2.25: - The agent-based model has constant speed, something that is not true for many species (especially the ones with evidence of leadership, since differentiation in speed is always enough for emergent leadership to appear). Can the authors run simulations with variable speed or measure the speed variation in the bird trajectories to check that there isn't strong variation there? Especially when it comes to turning and changing altitude, acceleration can have a large effect on individual interactions.

Response: We thank Reviewer #2 for the very constructive comment. Since Reviewer #1 also proposed a few concerns regarding the impact of speed on MS or MST, we have comprehensively discussed the relations between MS/MST and speed with a unified manner in **Point 1.2, 1.3, 1.4, 1.5, 1.6** and **1.8**. Please refer to these points for detailed information regarding the impact of speed on MS or MST.

In the revised version, we have incorporated the analysis results about (i) correlation of MS-Speed, MST-Speed, (ii) comparison of correlation between LF-MS and LF-Speed, and (iii) new observations between MST and velocity difference. Based on the new results, we have rewritten the interpretations of MST/MS, cited the related references in the main text (see **main text: pages 6-8, lines 184-271**), and replenished the detailed analysis information in Supplementary Sec.6 (see **SI: pages 11-13** and **Supplementary Figs.32-37**).

Point 2.26: - Lines: 399-403: the flocks cannot really evolve a mechanism neither individual 'preference' can be measured. Higher MS will make an individual less influenced by others, meaning it will follow its own tendency more and perhaps end up leading the group, which is a very interesting self-organized mechanism that I don't think it is given enough importance from

the wording of the paragraph. If the authors wish to discuss the adaptive reason behind different behavior between mobbing, roosting, and transiting, they should link their discussion better with the ecological context and the specific characteristics of their study species.

The discussion doesn't include the empirical results on bird flocks or mention any effect of across species differences, even though the authors mention in their introduction that species have been found to differ in their interactions with plasticity even found in one of the datasets (ref [15] in the manuscript).

Response: We thank Reviewer #2 for the very constructive suggestion.

We acknowledged the significance of collective behaviors such as mobbing, roosting, or transiting in relation to their ecological environment and physiological characteristics of the study species. However, our primary focus in this study is to elucidate the underlying factors that shape the emergence of collective motions and to establish a comprehensive research chain from biological observation to bionic mechanism to bio-inspired swarm robotics. Regrettably, due to our limited expertise in ecology and animal behavior, we were unable to thoroughly consider the intricate ecological context in our analysis. For example, for the mobbing and transit datasets [Ling et al., *Nature Ecology & Evolution*, 2019, DOI: [10.1038/s41559-019-0891-5](https://doi.org/10.1038/s41559-019-0891-5)], the Jackdaw (*Corvus monedula*) always forms the long-term monogamous relationships and both parents contribute to rearing the young, therefore, large numbers of individuals recorded in these datasets include mated pairs, unpaired individuals and juveniles. Indeed, we acknowledged that the impact of pair-bonded relationships on quantifying MS is overlooked in our analysis. It is evident that these relationships can significantly influence the measurement of MS. We hoped that future research will bridge this gap and provide a more nuanced perception of MS within the context of the study species' ecological and physiological attributes. Thanks to reviewer's constructive comment, we openly admitted this point in the Discussion of revised manuscript.

Combining some of the previous constructive comments, we have rewritten this paragraph in the Discussion part (see **main text: pages 14-15, lines 491-519**):

“Our work represents a new and substantive departure from the status quo by shifting the focus from one of the specific aspects to a systems-level understanding of collective motions. In particular, we utilized three large bird-flocking datasets encompassing diverse ecological context and motion patterns, namely mobbing, circling, and transit. This enabled us to conduct a systematic investigation into the emergence of LF relations and concurrently identify the factors that facilitate to the development of such relationships. The empirical findings indicate that the flocks characterized by higher maneuverability, such as those observed in mobbing and circling behaviors, tend to exhibit a more nested structure of LF relations and a distinct hierarchy. This organizational pattern could help to potentially alleviate the negative impact of individual drive on group cohesion. In contrast, flocks with smooth motion (i.e., transit) do not display this strategic tendency to organize the group in a similar manner. To elucidate the empirical findings, we introduced the metric MS and performed the correlation analysis between LF and MS. Interestingly, we found that individuals with higher MS tend to occupy the higher leading tier in LF networks. It means that higher MS will make an individual less influenced by others, thereby implying a stronger inclination to follow its own tendencies and perhaps end up leading the group.

We acknowledged the significance of collective behaviors such as mobbing, roosting, or transit in relation to their ecological environment and physiological characteristics of the study species. However, our primary focus in this study is to elucidate the underlying factors that shape the emergence of collective motions and to establish a comprehensive research chain from biological observation to bionic mechanism to bio-inspired swarm robotics. Regrettably, due to our limited

expertise in ecology and animal behavior, we were unable to thoroughly consider the intricate ecological context in our analysis. For example, for the mobbing and transit datasets^{14,15}, Jackdaw (*Corvus monedula*) always forms the long-term monogamous relationships and both parents contribute to rearing the young, therefore, large numbers of individuals recorded in these datasets include mated pairs, unpaired individuals and juveniles. Indeed, we openly admit that the impact of pair-bonded relationships on quantifying MS was overlooked in our analysis, but these relationships can significantly influence the measurement of MS. We hoped that future research will bridge this gap and provide a more nuanced perception of MS within the context of the study species' ecological and physiological attributes.”

Point 2.27: - Lines 394-395: I am not sure if MS is an entry point for movement changes of neighbors but rather their timing. For instance, there are many established measurements on changes in neighbor relative positions [4,5]. Some argumentation of whether the new measurement can be complimentary or replace the previous ones will improve the impact of the paper.

Response: We thank Reviewer #2 for the constructive suggestion. It reminds us to compare MS with other established metrics on changes in neighbor relative positions [Papadopoulou et al, 2023, Cavagna et al., 2013]. Here we chose the instability of neighbor as comparison for two reasons: (i) both instability of neighbor and MS could comprehensively quantify the relative changes without directly involving velocity or acceleration; (ii) two metrics are temporal or could quantify the relative changes for any period of a track.

Fig.R29 shows the comparison results for three flocks shown in Fig.1a-c. For example, for a period of track (highlighted by red in **Fig.R29a**), we derived the average MS ($M_i(t, \tau)$) and instability of neighbor ($Q_M^i(t, \tau)$) of each individual, then to perform the Spearman correlation between two vectors composed of ($M_i(t, \tau)$) and ($Q_M^i(t, \tau)$), respectively (**Fig.R29b**). Next, we extended the Spearman correlation to different combinations of t and τ , and found that for the mobbing flock shown in Fig.1a, the Spearman correlation of MS and instability of neighbors is almost positive for any period of the whole flock (**Fig.R29c**).

However, instability of neighbor could fail to quantify the changes of relative positions if $Q_M^i(t, \tau) = 0$. This situation happens when the relative positions of neighbors may be changing, but the ranking of the nearest neighbor remains unchanged. **Fig.R29d,e** show this case in the circling flock. For example, in the period of collective turn (highlighted by red in **Fig.R29d**), $Q_M^i(t, \tau) = 0$ for all individuals in the flock, but MS still works to quantify the relative changes (**Fig.R29e**). We found that this case happens many times in different combinations of t and τ (see blacks in **Fig.R29f**). Furthermore, we counted the percent of the case $Q_M^i(t, \tau) = 0$ in the whole combinations of t and τ , and **Fig.R29g-i** indicate that the case $Q_M^i(t, \tau) = 0$ is less in the mobbing flock, while the percent of $Q_M^i(t, \tau) = 0$ is larger than 50% in circling and transit flock. The comparisons in three flocks from three datasets demonstrate that MS has a wider range of applications to quantify changes in neighbors' relative positions.

Point 2.28: - The font size of many figures should increase.

Response: Thanks for pointing that out. We have revised those figures in the main text and Supplementary Information.

Finally, we thank Reviewer #2 again for her/his very insightful and constructive comments/suggestions. We hope our responses above have addressed those very legitimate issues/concerns in a satisfactory manner.

Response to Reviewer #3

Point 3.0: In the paper “Perception of Motion Saliency Shapes the Emergence of Collective Motions”, the authors present a bio-inspired flocking algorithm. In the first part, the authors study leader-follower relations in different collective motion patterns of two species of birds. The authors compute the lag and advance time between pairs of birds and use these to identify leader-follower relationships in the flock. The authors show that collective motion patterns with higher motion saliency also exhibit a stronger leader-follower hierarchy. Based on these results, the authors also develop an algorithm (AMS) to control the flocking behavior of a swarm of ground robots. Experiments are conducted both in simulation and with real robots.

I, unfortunately, lack the expertise to judge about the quality and soundness of the biological part of the study but the paper appears to be sound overall, especially in the swarm robotics part. It is commendable that experiments with real robots have been performed.

Response: We thank Reviewer #3 very much for reviewing our work and her/his positive assessment of our work. We next address each of the reviewer’s comments in order.

Point 3.1: In the discussion, the authors state that there exists “a lack of a comprehensive research chain from biological observation to bionic mechanism to bio-inspired swarm robotics”. While it is true that only few works consider the full pipeline, it is also to note that the field of swarm robotics has moved away from pure bio-inspiration (see for example Winfield et al., SAB 2004; Brambilla et al., Swarm Intelligence 2013; Francesca and Birattari, Frontiers in Robotics and AI 2016).

Response: We appreciate Reviewer #3 for bringing these papers to our attention. In the revised manuscript, we have incorporated these insightful reviews in the Discussion (see **main text: page 14, lines 479-483**):

“Despite the shift in swarm robotics away from pure mathematical modeling or bio-inspiration^{68,69,70}, the current state regarding fundamental mechanisms and potential applications of collective motions could still be encapsulated as follows: a deficiency in establishing a comprehensive research chain from biological observation to bionic mechanism to bio-inspired swarm robotics.”

Point 3.2: A general comment: The content of this paper is very dense, with many formulas to keep track of. Unfortunately, definitions and formulas are split between “Results”, “Methods”, and “Supplementary Material”. For example, the authors give the definition for τ_{LF} in line 122 (“Results”), the definition for $M_{i,j}$ in line 454 (“Methods”), and the definition of R in the Supplementary material. All three of these metrics are reported in the results (Figure 1, Figure 2, and Figure 5, respectively). Due to the distributed locations, it could be challenging for the reader to find the location of a definition. I would suggest consolidating all definitions in the “Methods” section. Furthermore, the paper is introducing many abbreviations, such as MS, ATHD, LF, MST. I believe that the readability of the manuscript could be improved by replacing most abbreviations with their proper text (especially MS/motion saliency and LF/leader-follower).

Response: We thank Reviewer #3 for the suggestions. To more clearly express our findings, we have revised and adjusted some definitions in the new manuscript as following:

- The definitions of key metrics, such as τ_{ij}^{LF} , $M_{ij}(t, \tau)$, τ_{ij}^{MS} and R , are all introduced in the Results to facilitate seamless reading without the need to switch between Results and Methods. The descriptions in Methods supplement to the corresponding definitions.

- Based on reviewer's suggestion, we have substituted some abbreviations with their full terms, particularly for MS or LF with motion salience or leader-follower. Besides, in the end of Introduction, we have added a sentence to emphasize the abbreviations of LF and MS (see **main text: page 4, lines 105-106**):

“Hereinafter, we used the abbreviations of LF and MS to refer to leader-follower and motion salience, respectively.”

Finally, we thank Reviewer #3 again for her/his very insightful and constructive comments. We hope our responses above have addressed those very legitimate issues/concerns in a satisfactory manner.

Response Figure

Figure R1 | The comparison of NODF and Spectral Radius of LF networks for Mobbing dataset. There are in total 140 mobbing flocks. Each point corresponds to a mobbing flock. The NODF as a function of group size (**a**) and density of edges in an LF network (**b**). The Spectral Radius as a function of group size (**c**) and density of edges in an LF network (**d**). The normalized Spectral Radius as a function of group size (**e**) and density of edges in an LF network (**f**). The NODF as a function of corresponding Spectral Radius (**g**) and normalized Spectral Radius (**h**). The point size of panels g,h linearly scales with the corresponding group size of each flock. In the rightmost column (labeled as I,II,III,IV), we showed four LF networks by nested-plot layout. We also labeled four LF networks in panels a-h.

Figure R2 | The null models of NODF. **a**, We presented 4 null models for networks, each applied to the 4 LF networks of example-I, II, III, and IV. Within each row, the violin plot of NODF represents 100 independent randomizations. **b-e**, For the mobbing dataset, the average NODF of 4 null models based on 100 independent randomizations, displays as a function of the NODF of real LF networks. Each blue circle represents a mobbing flock, and the light red color represents the standard deviation of 100 independent randomizations. **f-i**, For the mobbing dataset, the boxplot for z-scores of average NODF of 4 null models. Each box represents a mobbing flock, and is calculated from the corresponding null model of 100 independent randomizations. Note that the x-axis in panels f-i is sorted as the ascending order of real NODF.

Figure R3 | The null models of normalized Spectral Radius. **a**, We presented 4 null models for networks, each applied to the 4 LF networks of example-I, II, III, and IV. Within each row, the violin plot of normalized Spectral Radius (n-Spectral Radius) represents 100 independent randomizations. **b-e**, For the mobbing dataset, the average n-Spectral Radius of 4 null models based on 100 independent randomizations, displays as a function of the n-Spectral Radius of real LF networks. Each blue circle represents a mobbing flock, and the light red color represents the standard deviation of 100 independent randomizations. **f-i**, For the mobbing dataset, the boxplot for z-scores of average n-Spectral Radius of 4 null models. Each box represents a mobbing flock, and is calculated from the corresponding null model of 100 independent randomizations. Note that the x-axis in panels f-i is sorted as the ascending order of n-Spectral Radius of real LF networks.

Figure R4 | Statistics of bird's average speed, radial and angular speed for three flocking datasets. For mobbing, circling and transit datasets, we counted three speeds of 3103, 75683, and 2597 birds, respectively. Note that each bird's average speed, radial and angular speed is calculated over the entire duration of a flock.

Figure R5 | The workflow to perform the correlation analysis between MS and three kinds of speed. Similar with Fig.3a in the main text, a period of flock $[t - \tau, t]$ (a) could derive each bird's average speed, radial and angular speed (b), and average MS of each individual (c). For two vectors, such as average MS and average speed derived from $[t - \tau, t]$, we calculated their Spearman correlation (ρ). Finally, we could get the ρ between MS and three kinds of speed for different combinations of (t, τ) within a flock (d). We extended the correlation analysis to three datasets and yielded the distribution of $\rho_{\text{MS-Speed}}(t, \tau)$, $\rho_{\text{MS-Radial}}(t, \tau)$ and $\rho_{\text{MS-Angular}}(t, \tau)$. Here we took MS with $\alpha = 0$ (e) and $\alpha = 1$ (f). For each flock, we took 21 time stamps for t and 22 time points for τ no matter how long a flock lasts.

Figure R6 | Boxplot of the correlation of MS-LF and three types of MS-Speed for three flocking datasets. In order to facilitate a clear comparison of the correlation levels between MS-LF (as shown in Fig.3 of the main text) and three types of MS-Speed, we presented the boxplots of $\rho_{MS-LF}(t, \tau)$, $\rho_{MS-Speed}(t, \tau)$, $\rho_{MS-Radial}(t, \tau)$ and $\rho_{MS-Angular}(t, \tau)$ for mobbing, circling and transit datasets.

Figure R7 | Illustration of comparison of correlation of LF-MS and LF-Speed for a mobbing flock. Here we used a mobbing flock (shown in Fig.1a of the main text) to show the workflow of

comparison. **a-c**, we performed the Spearman correlation to yield $\rho_{\text{LF-MS}}(t, \tau)$ and $\rho_{\text{LF-Speed}}(t, \tau)$, $\rho_{\text{LF-Radial}}(t, \tau)$, $\rho_{\text{LF-Angular}}(t, \tau)$ for different combinations of (t, τ) . In panels a-c, each panel contains 231 points corresponding to 231 combinations of (t, τ) within the mobbing flock. **d-f**, The heatmap of difference of ρ as $\Delta\rho(t, \tau) = \rho_{\text{LF-MS}}(t, \tau) - \rho_{\text{LF-Speed}}(t, \tau)$, $\rho_{\text{LF-MS}}(t, \tau) - \rho_{\text{LF-Radial}}(t, \tau)$, $\rho_{\text{LF-MS}}(t, \tau) - \rho_{\text{LF-Angular}}(t, \tau)$. The gradient color ranging from blue to white to red in panels d-f is indicative of the value of $\Delta\rho \in [-2, 2]$. **g**, We showed the flocking trajectory from $[t - \tau, t]$ with the gradient background color scaling with $\Delta\rho = \rho_{\text{LF-MS}}(t, \tau) - \rho_{\text{LF-Speed}}(t, \tau)$. We observed that the $\Delta\rho(t, \tau)$ values indicated by the red background color consistently correspond to the process of collective turn. **h-j**, the 231 points are categorized into 3 parts: $|\Delta\rho| < 0.2$, $\Delta\rho > 0.5$ and $\Delta\rho < -0.5$. The y-axis of panels h-j records the corresponding average angular and nestedness calculated from the period $[t - \tau, t]$.

Figure R8 | Illustration of comparison of correlation of LF-MS and LF-Speed for a transit flock. Here we used a transit flock (shown in Fig.1c of the main text) to show the workflow of

comparison. **a-c**, we performed the Spearman correlation to yield $\rho_{\text{LF-MS}}(t, \tau)$ and $\rho_{\text{LF-Speed}}(t, \tau)$, $\rho_{\text{LF-Radial}}(t, \tau)$, $\rho_{\text{LF-Angular}}(t, \tau)$ for different combinations of (t, τ) . In panels a-c, each panel contains 231 points corresponding to 231 combinations of (t, τ) within the transit flock. **d-f**, The heatmap of difference of ρ as $\Delta\rho(t, \tau) = \rho_{\text{LF-MS}}(t, \tau) - \rho_{\text{LF-Speed}}(t, \tau)$, $\rho_{\text{LF-MS}}(t, \tau) - \rho_{\text{LF-Radial}}(t, \tau)$, $\rho_{\text{LF-MS}}(t, \tau) - \rho_{\text{LF-Angular}}(t, \tau)$. The gradient color ranging from blue to white to red in panels d-f is indicative of the value of $\Delta\rho \in [-2, 2]$. We found three kinds of $\Delta\rho$ is significantly lower than that of mobbing flock, because the color intensity in Fig.R7d-f is noticeably lighter compared to that in panels d-f. **g**, We showed the flocking trajectory from $[t - \tau, t]$ with the gradient background color scaling with $\Delta\rho = \rho_{\text{LF-MS}}(t, \tau) - \rho_{\text{LF-Speed}}(t, \tau)$. **h-j**, the 231 points are categorized into 3 parts: $|\Delta\rho| < 0.2$, $\Delta\rho > 0.5$ and $\Delta\rho < -0.5$. The y-axis of panels h-j records the corresponding average angular and nestedness calculated from the period $[t - \tau, t]$.

Figure R9 | Comparison of correlation of LF-MS and LF-Speed for three flocking datasets.

We extended the three kinds of $\Delta\rho$ to mobbing, circling and transit datasets. In mobbing and circling dataset, the sub-flocks with $\Delta\rho > 1$ exhibit the highest nestedness, while those sub-flocks with $\Delta\rho < -1$ display the lowest nestedness. The nestedness of sub-flocks with $|\Delta\rho| < 0.2$ are positioned in the middle. Note that the trend observed in mobbing and circling datasets is the same for three kinds of $\Delta\rho$. For the transit dataset, we could not observe the clear pattern between $\Delta\rho > 1$ and $\Delta\rho < -1$.

Figure R10 | The distribution of Spearman correlation between MST and three kinds of speed for three flocking datasets. We found neither positive nor negative correlations between MST and Speed are predominant in the three flocking datasets. Here we took MST with $\alpha = 0$.

Figure R11 | MST and velocity difference of individual pairs. The trajectories, $\langle M_{ij} \rangle(\tau)$ and temporal $\mathbf{v}_i(t) \cdot \mathbf{v}_j(t)$ of pair (1,6) (a-c) and (4,7) (d-f) from a mobbing flock of Fig.1a in main text. The MST area, highlighted by grey in panels b,e, is under the curve of $\langle M_{ij} \rangle(\tau)$. The average velocity difference is $\Delta v_{ij} = 1 - \langle \mathbf{v}_i(t) \cdot \mathbf{v}_j(t) \rangle$ throughout the entire duration of a flock for a bird pair-(i, j). In this example, we found that MST of pair-(1,6) is larger than that of pair-(4,7), and correspondingly, Δv_{ij} of pair-(1,6) is also larger than that for pair-(4,7). The MST ($\alpha = 0$) matrix (g), average velocity difference (Δv_{ij}) matrix (h) and MST area matrix (i) of a mobbing flock of Fig.1a in main text. In panels g-i, the gradient colors from black to white map the descending order of MST and $\langle \mathbf{v}_i(t) \cdot \mathbf{v}_j(t) \rangle$. j, The scatter plot of corresponding elements of MST matrix and average velocity difference matrix from panels g and h. k, The scatter plot of corresponding elements of MST area matrix and average velocity difference matrix from panels g and i. We consistently observed a clear positive relationship for all bird pairs, whether considering MST or MST area, indicating that larger MST corresponds to a greater velocity difference.

Figure R12 | The distribution of Spearman correlation between MST (or MST area) and average velocity difference for three flocking datasets. We broadened the analysis to mobbing, circling and transit datasets, and found that the predominant correlations in all three flocking datasets are positive for $\rho_{\text{MST}-\Delta v_{ij}}$ (a-c) and $\rho_{\text{MSTarea}-\Delta v_{ij}}$ (d-f). In this figure, we took MST with $\alpha = 0$ since the dot product of $\mathbf{v}_i(t)$ and $\mathbf{v}_j(t)$ also does not consider the fact of front-oriented preference in the flocks.

Figure R13 | The distribution of correlation between LF and MS if three flocking datasets are categorized by average speed in 1m/s interval. Here the speed range is categorized by the flock's average speed. Then the flock is performed by the correlation analysis between LF and MS as shown in Fig.3a.

Figure R14 | The distribution of correlation between LF and MS if three flocking datasets are categorized by average angular speed in 0.1rad/s interval. Here the range of average angular speed is categorized by the flock's average angular speed. Then the flock is performed by the correlation analysis between LF and MS as shown in Fig.3a.

Figure R15 | The individuals moving in the opposite direction of the exit in experiment of Fig.6c. **a**, To gain a better understanding of the self-organized mechanism of AMS and why certain individuals (highlighted by red boxes in Supplementary Fig.45) are moving in the opposite direction of the exit, we presented three types of information about these individuals: trajectory, $v_i(t)$ (black arrow), and $\hat{v}_{al,i}$ (red arrow). **b**, We compared the dot product between $v_i(t)$ and $\hat{v}_{al,i}(t)$ for the results of AMS, ATHD and Average shown in Fig.6a-c (in the main text). We found the dot product between $v_i(t)$ and $\hat{v}_{al,i}(t)$ in descending order is AMS, ATHD and Average, and AMS has the least variance of dot product.

Figure R16 | The scatter plot between nestedness of LF networks and flocking time for all Transit flocks. There are in total 94 transit flocks. Each point corresponds to a transit flock.

Figure R17 | The scatter plot between nestedness of LF networks and average order for different flock id. Different with Fig. 1g-i that we collected flocks with different id together to show the relations between nestedness of LF networks and group order, we separately calculated the relations of each flock id for three flocking datasets. Each point represents a flock, and the nonparametric regression and bootstrap sampling are performed to calculate the trend (red curve) and its 94% confidence interval (red shadow) between nestedness and average order.

Figure R18 | The scatter plot between nestedness of LF networks and average order for different proportions of group size. **a**, The distribution of group size for three datasets. According to group size, we divided the flocks into three parts and then to calculate their nestedness of LF networks and average order: **(a)** the first third of all group size, **(b)** the middle third of all group size and **(d)** the last third of all group size. Each point represents a flock from three datasets, and the nonparametric regression and bootstrap sampling are performed to calculate the trend (red curve) and its 94% confidence interval (red shadow) between nestedness and average order.

Figure R19 | The scatter plot between nestedness of LF networks and average order for different proportions of flocking time. **a**, The distribution of flocking time (or called duration) for three datasets. According to flocking time, we divided the flocks into three parts and then to calculate their nestedness of LF networks and average order: **(a)** the first third of all flocking time, **(b)** the middle third of all flocking time and **(d)** the last third of all flocking time. Each point represents a flock from three datasets, and the nonparametric regression and bootstrap sampling are performed to calculate the trend (red curve) and its 94% confidence interval (red shadow) between nestedness and average order.

Figure R20 | Four temporal metrics for the mobbing and transit flock shown in Fig.1. We used four flocking metrics, e.g., group order, trajectory curvature, group density and instability of neighbor to systematically measure the properties of collective motions.

Figure R21 | The average of four temporal metrics for three flocking datasets. Here we averaged the temporal metrics over the entire period of a flock. Each point corresponds to a transit flock.

Figure R22 | The relation between trajectory curvature and group order accords with our common sense and the observed collective motions across three datasets: the higher group order, the less trajectory curvature. Each point represents a flock from three datasets, and the nonparametric regression and bootstrap sampling are performed to calculate the trend (red curve) and its 94% confidence interval (red shadow) between two measurements.

Figure R23 | The relation between average instability of neighbors and other flocking metrics across three datasets. Average instability of neighbors vs average order (a), average curvature (b), and average group density (c), respectively. Each point represents a flock from three datasets, and the nonparametric regression and bootstrap sampling are performed to calculate the trend (red curve) and its 94% confidence interval (red shadow) between two measurements.

Figure R24 | The relation between average MST and other flocking metrics across three datasets. Average MST ($\alpha = 0$) vs average order (a), average instability of neighbor (b), average curvature (c), and average group density (d), respectively. Each point represents a flock from three datasets, and the nonparametric regression and bootstrap sampling are performed to calculate the trend (red curve) and its 94% confidence interval (red shadow) between two measurements.

Figure R25 | Correlation analysis between two approaches of quantifying individual leadership in a mobbing flock. This flock is same with Fig.1a. For a flock within the period $[t - \tau, t]$ (highlighted by red in panel a), first we could construct the LF network and lag time matrix, and consequently derive the leading tier $L_i(t, \tau)$ and rank $R_i(t, \tau)$ of each individual. Interestingly, two vectors composed of $L_i(t, \tau)$ and $R_i(t, \tau)$ show the highly positive correlation.

Figure R26 | Comparison of self-propelled particle model in presence or absence of MS in 2D. The panel layout is identical to Fig. 4 in the main text, with the only distinction being that the results in this figure are derived from 2D data. We used a self-propelled particle model that particles follow the local interaction rules, i.e., AMS (a), ATHD (b) and average interaction (c). The additional potential well, imposed on one of individuals (red trajectories), aims to lead the flock come back to origin. The swarm size is 30 particles. The black points represent the end of trajectories. In panel a, the flocking trajectories are generated by AMS with $\alpha = 0$. **d-f**, The temporal curvature of flock trajectories respectively shown in panels a-c. In panels d-f, the solid curves represent the average of trajectory curvatures from 30 particles, and the shadow area represents the standard deviation (SD). **g**, The distribution of nestedness of LF networks derived from flocking trajectories using AMS, ATHD and average interaction. The white points (or black lines) represent the median (or mean) value. **h**, The distribution of Spearman correlation (ρ) between LF and MS over different combinations of t and τ from flocking trajectories using AMS, ATHD and average interaction. For each flock we took 10 time stamps for t and 11 time points for τ to perform the correlation analysis between LF and MS. In panel g and h, we ran 50 independent simulations for each interaction type.

Figure R27 | A tunable parameter of the number of individuals leading the group in the simulation results of Fig.4. The simulation results are generated by the same setting and parameters with Fig.4 except the number of individuals leading the group. The curves of average curvature of flocking trajectory (**a**) and nestedness (**b**) as a function of the number of individuals leading the group. The percent in the parenthesis indicates the proportion of informed agent to lead the group.

Figure R28 | Consider the restricted angular velocity in the simulation of collective following. We set two kinds of restricted angular velocity in collective following simulations: **(a)** the informed agent and others share the same max angular velocity, **(b)** max angular velocity of the others is fixed at 10×1.91 rad/s but we changed the informed agent with different max angular velocity.

Figure R29 | Correlation analysis between MS and instability of neighbor. For a period of track (highlighted by red in **a,d**), we derived the average MS ($M_i(t, \tau)$) and instability of neighbor ($Q_M^i(t, \tau)$) of each individual, then to perform the Spearman correlation between two vectors composed of ($M_i(t, \tau)$) and ($Q_M^i(t, \tau)$), respectively (**b,e**). Next, we extended the Spearman correlation to different combinations of t and τ , and found that for the mobbing flock shown in Fig.1a, the Spearman correlation of MS and instability of neighbors is almost positive for any period of the whole flock (**c**). However, instability of neighbor could fail to quantify the changes of relative positions if $Q_M^i(t, \tau) = 0$ (**e,f**). The percent of $Q_M^i(t, \tau) = 0$ in the mobbing (**g**), circling (**h**) and transit (**i**) flock.

REVIEWER COMMENTS

Reviewer #1 (Remarks to the Author):

The authors have performed a substantial revision to the manuscript, and they have done extensive work to respond to the reviewers' concerns, which has improved the manuscript. I believe that the manuscript, in its current stage, remains interesting exploratory work which takes on the ambitious challenge to connect observational evidence, model-based simulations, and implementation in the laboratory with actual swarm experiments. I still believe that the manuscript is an impressive effort whose results contain some interesting insights.

However, given the substantial nature of the major issues, I cannot unfortunately recommend publication in Nature Communications.

Specifically, I remain very concerned about the lack of clear mechanisms behind the observed results, whether the correlational results are non-trivial, the significance of the results and their contribution to the literature. Other shortcomings concern the nestedness-related results and the readability of the manuscript. I consider it unlikely that these major issues could be solved without performing substantially more research.

I report below my major concerns, in decreasing order of importance. I hope the authors will find my comments helpful to further develop their paper.

(A) Lack of clear mechanisms

A.1. Where does the MS come from?

The authors define a quantity, the perception of motion salience (MS, eq. 1), which is found to correlate with individual leadership. In my previous report, I mentioned that no hint at potential mechanisms was provided, and worried that the result could be confounded by the birds' velocities. The authors did a comprehensive analysis to show that the correlations between MS and leadership are not explained by velocity, but they did not clarify or validate alternative mechanisms behind their results.

Some sentences that potentially point to a mechanism are:

“The difference between MS and Speed could mainly be attributed to the scope of their respective definitions. From the definition, MS directly or indirectly takes into account the variations in position, heading, speed and acceleration over a period of flock “

“It is evident that speed, when compared to MS, is not a comprehensive characteristic for distinguishing the modes of collective motion, especially for extremely agile flocks.”

However, these statements remain vague. It is left unclear *how* the MS directly or indirectly takes into account all those properties. Perhaps more importantly, without detailed empirical validation at the individual level, it is unclear if the MS is the actual property that is perceived by the birds and drives their movement. The interpretation of the three factors in Eq. 1 is clear, but why to take their product? Is it the result of a principled way to combine these properties or simply a heuristic attempt that, somehow, “works”?

The swarm model section proposes the hypothesis that “AMS interactions could be responsible for shaping the emergence of nested and hierarchical LF relations in collective motions“, and then tests it through simulations. But it remains unexplained, also in the model-based results presented, how the AMS causes the emergence of LF relations.

What are possible mechanisms? If there are, do they apply equally to birds, particles in the self-propelled model, and robots in the lab? How to validate them in the data?

A.2. Are the correlations non-trivial?

One of the main correlational claims of the manuscript is the correlation between LF and MS. In my initial report, I already expressed the concern that the high correlation might be already implied by the definition. I remain not persuaded that the correlation is non-trivial. The MS depends on the velocities and positions of the birds at times t and $t-\tau$ (averaged over columns as explained in line 275). The results show that it correlates with another quantity that depends on the relative velocities of the birds at times t and $t-\tau$ (i.e., the individual leadership whose definition can be inferred from lines 140, 153, and 276 + Methods).

The average MS rewards birds toward which many other birds tend to get closer especially when there is a large angle between the relative positions of the birds at times t and $t-\tau$, which seems conceptually related to the concept of leadership encapsulated in the LF definitions.

A.3. Contribution and relevance of the results.

As in the previous version, by reading the introduction, it remains very hard for the reader to infer the actual challenge being solved by the manuscript. Many broad problems are mentioned, but the introduction fails to give a clear sense of what is being solved and why it is significant. A broad spectrum of limitations of existing studies are mentioned — e.g., the use of simplified interaction rules in existing models, the use of phenomenological interactions, lack of integration of the perception stream.

But the only solution offered by the manuscript to this broad array of limitations seems to be the use of the MS, which does not seem to fully solve the mentioned problems. For example, as the MS seems to be a heuristic quantity, and the MS mechanism is not investigated or validated in detail, it seems again that we have a “phenomenological” model (not derived from first principles) similarly to previous work. The model by the authors makes parsimonious assumptions on individual behavior, without the interaction mechanism being detailedly validated in the data. Many other issues mentioned in the introduction are, in the end, not fully addressed by the present submission.

Similarly, in the Discussion, the authors define the status quo in the study of collective motions (line 479), but it remains unclear whether the manuscript constitutes a substantial shift in the status quo, given its limitations. For example, in the absence of a clear mechanism behind the observed correlations and experimental results, it is unclear if the obtained results would generalize beyond the analyzed flock datasets and the specific evacuation experiments performed.

In sum, it is hard to conclude that the authors have indeed achieved a “comprehensive research chain from biological observation to bionic mechanisms to bio-inspired robotics”. The manuscript presents interesting ideas and initial insights in three areas, but seems far from achieving a clear link between the three aspects without doing substantially more research on the individual-level mechanisms behind their findings.

(B) Causal claims not supported by the results.

Causal sentences have been reduced in the manuscript, but some key claims of the manuscript are still causal, e.g., the MS has the “ability to bridge perception and motion”, the AMS “empowers the swarm to promptly respond to the transient perturbation”; in the abstract, “flocks with higher maneuverable motions (i.e., mobbing and circling) tend to exhibit a more nested structure of leader-follower (LF) relations and a clear hierarchy *to mitigate* the damage of individual drive to group cohesion”.

These claims imply that the swarms leverage the MS to achieve certain goals, but there is no evidence that this phenomenon is taking place, as far as I can tell from the results. In the correlational section, for example, the results indicate that swarms with higher/lower MS have different structural characteristics, but this does not fully prove that MS is the causal reason behind the differences. As the authors recognize in the discussion, the ecological contexts and other factors could play a role.

The only exception perhaps is the collective evacuation experiment. However given the specific nature of the setup and lack of detailed mechanisms, it is unlikely that an overly general conclusion on the role of the MS can be inferred from this single experiment.

(C) Nestedness results.

The authors followed my advice to compare raw nestedness metrics with the levels of nestedness observed in randomized networks of increasing level of constraint. But the results seem to show that even under the loosest null models, the nestedness is rarely significant, as the z scores rarely exceed +2 (as a side note, using p-values instead of z-scores would be more suitable in the absence of clear evidence that the nestedness metrics follow a normal distribution in the randomized networks). This lack of significance seems to suggest that the observed nestedness of the binarized networks simply results from heterogeneity in interaction frequencies, which is not discussed in the manuscript. Perhaps a weighted nestedness metric, such as the spectral radius of the weighted network, might produce significant results and interesting insights.

(D) Readability.

The manuscript remains very dense, with many definitions, abbreviations, and results briefly mentioned but not fully explained. The text is written in a “non linear” way, and the reader has to go back and forth from the main text to methods and back to try to understand the details. Some definitions (e.g., individual leadership) can only be inferred by the reader through combining pieces of information from far-away paragraphs. Besides, the text seems more like an initial draft, it is not

polished at various points of the manuscript and reply letter, which sometimes prevents the reader from fully understanding the point the authors wish to convey.

Reviewer #2 (Remarks to the Author):

The authors did a great job revising the manuscript, I find the new version highly improved in context and clarity and all my concerns have been tackled. I would thus happily recommend its acceptance for publication.

Reviewer #3 (Remarks to the Author):

The authors have addressed my comments in the latest revision of the manuscript titled "Perception of Motion Saliency Shapes the Emergence of Collective Motions". I believe that these changes have improved the quality of the manuscript.

Response to Reviewer #1

Point 1.0: The authors have performed a substantial revision to the manuscript, and they have done extensive work to respond to the reviewers' concerns, which has improved the manuscript. I believe that the manuscript, in its current stage, remains interesting exploratory work which takes on the ambitious challenge to connect observational evidence, model-based simulations, and implementation in the laboratory with actual swarm experiments. I still believe that the manuscript is an impressive effort whose results contain some interesting insights.

However, given the substantial nature of the major issues, I cannot unfortunately recommend publication in Nature Communications.

Specifically, I remain very concerned about the lack of clear mechanisms behind the observed results, whether the correlational results are non-trivial, the significance of the results and their contribution to the literature. Other shortcomings concern the nestedness-related results and the readability of the manuscript. I consider it unlikely that these major issues could be solved without performing substantially more research.

I report below my major concerns, in decreasing order of importance. I hope the authors will find my comments helpful to further develop their paper.

Response: We thank Reviewer #1 very much for reviewing our paper again. Following her/his constructive comments and suggestions, we have carefully addressed each concern to improve the quality of our work. We summarized the major revisions as follows:

- 1) incorporating the new individual-level empirical finding as a separate subsection titled "Individuals will accelerate convergence of velocity with neighbors who have higher MS".
- 2) reorganizing all MS-related definition and interpretation into a new subsection titled "Perception of motion salience measures the relative motion changes" to enhance the clarity and effectiveness of defining MS.
- 3) rewriting the Abstract, Introduction, and Discussion sections in main text to address our motivations and contributions of this study.
- 4) removing (or deleting) certain contents from main text to SI that are not directly relevant to the key findings.

Next, we address each of the reviewer's comments in detail.

Point 1.1: (A) Lack of clear mechanisms

A.1. Where does the MS come from?

The authors define a quantity, the perception of motion salience (MS, eq. 1), which is found to correlate with individual leadership. In my previous report, I mentioned that no hint at potential mechanisms was provided, and worried that the result could be confounded by the birds' velocities. The authors did a comprehensive analysis to show that the correlations between MS and leadership are not explained by velocity, but they did not clarify or validate alternative mechanisms behind their results.

Some sentences that potentially point to a mechanism are:

“The difference between MS and Speed could mainly be attributed to the scope of their respective definitions. From the definition, MS directly or indirectly takes into account the variations in position, heading, speed and acceleration over a period of flock “

“It is evident that speed, when compared to MS, is not a comprehensive characteristic for distinguishing the modes of collective motion, especially for extremely agile flocks.”

However, these statements remain vague. It is left unclear *how* the MS directly or indirectly takes into account all those properties. Perhaps more importantly, without detailed empirical validation at the individual level, it is unclear if the MS is the actual property that is perceived by the birds and drives their movement.

Response: We thank Reviewer #1 very much for carefully reviewing our manuscript and consistently offering highly constructive feedback. The MS metric does not directly involve velocity and acceleration, but it reflects the variations of collective motions. In the previous manuscript, we did address this aspect, but didn't perform specific empirical validation at the individual level. We apologized for that.

Here, we reported two kinds of correlation analysis to validate our statements (see **Fig.R1a** for the workflow):

- 1) **correlation analysis between MS and the average velocity consensus (AVC) of each pair of birds within a time period $[t - \tau, t]$.** AVC, denoted as $\phi_{ij}(t, \tau) = \langle \mathbf{v}_i(t) \cdot \mathbf{v}_j(t) \rangle_{\tau}$, directly describes the consensus of two birds' velocities over $[t - \tau, t]$. A higher ϕ_{ij} value indicates a greater velocity consensus within a bird pair. Then we computed the Spearman correlation coefficient of two vectors consisting of $M_{ij}(t, \tau)$ and $\phi_{ij}(t, \tau)$ of all bird pairs. For a given flock, we expanded the correlation analysis between MS and AVC for various combinations of t and τ , as illustrated in **Fig.R1b**. We found that, in the flock depicted in **Fig.R1a**, the Spearman correlation between MS and AVC across different combinations of t and τ predominantly exhibited negative values. Similarly, the correlation analysis between MS and AVC in 3 bird-flocking datasets consistently demonstrates a prevalence of negative correlation (**Fig.R1d-f**). The negative correlation between MS and AVC aligns with our intuition, as increased motion salience within a bird pair would likely result in decreased velocity consensus.
- 2) **correlation analysis between MS and the distance of temporal speed (DS) of each bird pair within a time period $[t - \tau, t]$.** Here, the DS quantifies the distance of two vectors from two birds' temporal speed within $[t - \tau, t]$, i.e., $\mathcal{D}_{ij}(t, \tau) = \text{Distance}(\text{vec}(v_i(t)), \text{vec}(v_j(t)))$, as shown in **Fig.R1a(ii)**. If two birds closely align within a time interval $[t - \tau, t]$, their temporal speeds, particularly the pattern of speed variation are expected to be similar, resulting in a reduced DS value, e.g., see inset of **Fig.R1a(ii)**. Therefore, we calculated the Spearman correlation coefficient of two vectors consisting of $M_{ij}(t, \tau)$ and $\mathcal{D}_{ij}(t, \tau)$ for all bird pairs. For simplicity, the DS value $\mathcal{D}_{ij}(t, \tau)$ is calculated by the Euclidean distance. For a given flock, we expanded the correlation analysis between MS and DS for various combinations of t and τ , as illustrated in **Fig.R1c**. We found that, different from the correlation between MS and AVC as shown in **Fig.R1b**, the correlation between MS and DS predominantly exhibits positive values. We then performed the correlation analysis for all the three bird-flocking datasets,

finding similar results (see **Fig.R1g-i**). These results suggest that a higher MS value for a bird pair corresponds to a greater difference in temporal speed. Of course, this positive correlation also accords with our common sense, because larger relative motion changes in a bird pair are associated with greater changes in speed patterns.

The aforementioned correlation analysis highlights that although the MS equation does not directly incorporate variations of velocity and speed, it effectively incorporates these variations by considering the relative position changes between the starting and ending timestamps. This is due to the fact that the relative position changes between two timestamps could indicate the variations in heading, speed, and acceleration over a period of flock.

In the revised manuscript, we have thoroughly improved the wording and incorporated the above results to clarify the definition of MS more effectively:

- We have incorporated the above results into SI as a separate subsection titled “**5. Perception of motion salience reflects variations in heading, speed, and acceleration over a period of flock**” (see **SI: pages 10-11, lines 323-360**). We also have emphasized this point in the revised main text (see **main text: page 7, lines 219-222**) as follows:
 “**Note that although MS equation does not directly incorporate variations of velocity and speed, we found that it effectively incorporates these variations by considering the relative position changes between the starting and ending timestamps (see detailed analysis in Supplementary Sec.5).**”
- We have included Fig.R1 as Supplementary Fig.23 in the revised SI (see **SI: page 48**).

Point 1.2: The interpretation of the three factors in Eq. 1 is clear, but why to take their product? Is it the result of a principled way to combine these properties or simply a heuristic attempt that, somehow, “works”?

Response: We thank Reviewer #1 for this critical comment. We admitted that Eq.1 is a heuristic attempt to quantitatively reflect the relative motion changes between a bird pair. This attempt facilitates the discrete-data analysis of bird flocks and its straightforward implementation in swarm robotics.

Eq.1 takes the focal individual- i 's first-person view to represent three key components of MS:

- 1) the angle between two vectors of relative position at two timestamps $\angle\left(\frac{\hat{\mathbf{x}}_{ij}(t), \hat{\mathbf{x}}_{ij}(t-\tau)}{\tau}\right)$;
- 2) anisotropic effect of motion perception $\left(\frac{1+\hat{\mathbf{v}}_i(t) \cdot \hat{\mathbf{x}}_{ij}(t)}{2}\right)$ simulating forward-oriented preference of perception;
- 3) anisotropic factor (α) controlling the fact of forward-oriented preference.

The anisotropic effect of motion perception reflects the forward-oriented preference of perceiving neighbor- j 's motion from the focal individual- i 's first-person view (**Fig.R2b**). For the focal individual- i , its perception capability diminishes as the first-person view transitions from the front to the back. Here, the first-person view of individual- i aligns with itself heading at time t . For example, if $\angle(\hat{\mathbf{v}}_i, \hat{\mathbf{x}}_{ij}) = 0$, it means that neighbor- j locates directly ahead of the focal individual- i 's movement direction, and the focal individual could fully perceive the neighbors because $(1 + \hat{\mathbf{v}}_i \cdot \hat{\mathbf{x}}_{ij})/2$ is 1. Once $\angle(\hat{\mathbf{v}}_i, \hat{\mathbf{x}}_{ij})$ deviates from 0 to π or $-\pi$, it represents that the neighbor- j gradually moves from the front to back of the focal individual- i , and assumes the ability of motion perception could increasingly diminish since $(1 + \hat{\mathbf{v}}_i \cdot \hat{\mathbf{x}}_{ij})/2$ decreases from 1 to 0.

We incorporated the anisotropic factor $\alpha \geq 0$ in Eq.1 to control the fact of forward-oriented preference in biological perception (**Fig.R2c**). If $\alpha = 0$, $M_{ij}(t, \tau)$ ignores the blind area of perception because the last two components in Eq.(1) always equal to 1 regardless of the relative positions of neighbors. Increasing α in Eq.1 will amplify the anisotropic effect of motion perception, that is, the ability of individual perceiving movements of around neighbors gradually shrinks to the front vision only. For instance, $\alpha = 10$ controls $M_{ij}(t, \tau) \approx 0$ when the neighbors' relative positions are beyond the horizontal sight $(-\pi/2, \pi/2)$ (**Fig.R2c**).

In the revised manuscript, this issue has been addressed as follows:

- We have thoroughly reorganized all MS-related definition and interpretation into a new subsection titled “**Perception of motion salience measures the relative motion changes**” (see **main text: pages 6-7, lines 182-239**).
- As MST (maximal motion salience time) is not directly relevant to investigate interaction rules derived from real bird flocks, we have deleted the related materials from the revised manuscript. While MST is an intriguing metric for assessing the degree to which the focal individual aligns with its neighbors or follows its own drive, it is not directly relevant to investigating interaction rules derived from real bird flocks. Thus, we have included Fig.R2 as Fig.2 in the revised main text, which solely features the diagram defining MS (see **Fig.2 in main text: page 26**).

Point 1.3: The swarm model section proposes the hypothesis that “AMS interactions could be responsible for shaping the emergence of nested and hierarchical LF relations in collective motions”, and then tests it through simulations. But it remains unexplained, also in the model-based results presented, how the AMS causes the emergence of LF relations.

What are possible mechanisms? If there are, do they apply equally to birds, particles in the self-propelled model, and robots in the lab? How to validate them in the data?

Response: We thank Reviewer #1 for pointing out this issue and prompting us to reconsider the validation of AMS at the individual level using the empirical datasets.

In this study, the idea of AMS was inspired from our finding about “*the predominant positive correlation between MS and LF in real bird-flocking datasets*”, that is, “individuals with higher MS tend to occupy the higher leading role”. We then translated this finding to an adaptive MS-based interaction (AMS) as $\mathbf{v}_i(t+1) = \hat{\mathbf{v}}_i(t) + \sum_{j \in \mathcal{S}_i} w_{ij}(t) \hat{\mathbf{v}}_j(t)$ with $w_{ij}(t) = \frac{M_{ij}(t, \tau)}{\sum M_{ij}(t, \tau)}$. Thanks to reviewer’s critical comment, we realized that prior to translating the empirical finding to mathematical modeling, further research at the individual level is needed to address why individuals with higher MS tend to occupy the higher leading role. To fill in this gap, we considered how the focal individual- i calculates the perception of its neighbor’s MS within the past time period $[t - \tau_{\text{pre}}, t]$ to analyze the future trend of motion changes between the focal individual and its neighbors in the subsequent time period of $[t, t + \tau_{\text{next}}]$. We used two kinds of metrics to describe the future trends of motion changes between a bird pair:

- 1) **The average of temporal velocity consensus within the future period** $\phi_{ij}(t, \tau_{\text{next}}) = \langle \mathbf{v}_i(t) \cdot \mathbf{v}_j(t) \rangle$

One of the future trends of motion changes could be represented by the average of temporal velocity consensus $\phi_{ij}(t, \tau_{\text{next}}) = \langle \mathbf{v}_i(t) \cdot \mathbf{v}_j(t) \rangle$. The higher value $\phi_{ij}(t, \tau_{\text{next}})$ means the higher velocity consensus between individual- i and j . Therefore, for the focal individual-1, **Fig.R3b** shows its neighbors' $M_{1j}(t, \tau_{\text{pre}})$ (and the normalization $w_{1j}(t, \tau_{\text{pre}}) = \frac{M_{1j}(t, \tau)}{\sum M_{1j}(t, \tau)}$) perceived by individual-1 within the past interval $[t - \tau_{\text{pre}}, t]$ (colored by blue in **Fig.R3a**), and **Fig.R3c** shows the average of temporal velocity consensus ($\phi_{12}(t, \tau_{\text{next}}) = \langle \mathbf{v}_1(t) \cdot \mathbf{v}_2(t) \rangle$) between bird-1 and 2 within the subsequent interval $[t, t + \tau_{\text{next}}]$ (colored by red in **Fig.R3a**). Then we performed the correlation analysis between two vectors composed of $w_{1j}(t, \tau_{\text{pre}})$ and $\phi_{1j}(t, \tau_{\text{next}})$ respectively (**Fig.R3d**). It is intriguing that across all individuals present in the flock, negative correlations between $w_{ij}(t, \tau_{\text{pre}})$ and $\phi_{ij}(t, \tau_{\text{next}})$ are consistently prevalent for different combinations of τ_{pre} and τ_{next} (**Fig.R3e-g**). Moreover, we also noticed the persistence of negative correlations between $w_{ij}(t, \tau_{\text{pre}})$ and $\phi_{ij}(t, \tau_{\text{next}})$ across all birds in three bird-flocking datasets (**Fig.R3h-j**).

This result agrees well with our intuition, as seen from the perspective of the focal individual- i . When perceiving its neighbor- j who moved with significant motion changes (reflected in the higher value of $w_{ij}(t, \tau_{\text{pre}})$) in the past period $[t - \tau_{\text{pre}}, t]$, it could lead to a less synchronized velocity consensus (indicated by a lower value of $\phi_{ij}(t, \tau_{\text{next}})$) in the subsequent period $[t, t + \tau_{\text{next}}]$.

2) The average slope of temporal velocity consensus within the future period $k_{ij}(t, \tau_{\text{next}}) = \langle \text{slope of } \mathbf{v}_i(t) \cdot \mathbf{v}_j(t) \rangle$

Another method to describe the future trends of motion changes is calculating the average slope of temporal velocity consensus within the future period, denoted as $k_{ij}(t, \tau_{\text{next}}) = \langle \text{slope of } \mathbf{v}_i(t) \cdot \mathbf{v}_j(t) \rangle$.

From the definition, unlike $\phi_{ij}(t, \tau_{\text{next}})$ that measures the degree of velocity consensus in the future period, $k_{ij}(t, \tau_{\text{next}})$ directly depicts the change rate of the convergence or divergence of velocity consensus. It is worth noting that the value of $\phi_{ij}(t, \tau_{\text{next}})$ calculated from the future period could be influenced by the past $M_{ij}(t, \tau_{\text{pre}})$. For example, higher $M_{ij}(t, \tau_{\text{pre}})$ results in less velocity consensus between a bird pair by the end of $[t - \tau_{\text{pre}}, t]$, and consequently leads to a low level of $\mathbf{v}_i(t) \cdot \mathbf{v}_j(t)$ at the beginning of $[t, t + \tau_{\text{next}}]$. However, $k_{ij}(t, \tau_{\text{next}})$ overcomes this issue by solely indicating the rate of changes of $\mathbf{v}_i(t) \cdot \mathbf{v}_j(t)$ within the interval $[t, t + \tau_{\text{next}}]$, regardless of whether $\mathbf{v}_i(t) \cdot \mathbf{v}_j(t)$ is high or low at the beginning of this time interval.

Clearly, a larger value of $k_{ij}(t, \tau_{\text{next}})$ implies that two individuals are aiming to align with each other more rapidly in the future. **Fig.R4d** shows the correlation analysis between two vectors composed of $w_{1j}(t, \tau_{\text{pre}})$ and $k_{1j}(t, \tau_{\text{next}})$ respectively. Interestingly, **Fig.R4e-g** show that among all individuals, positive correlations between $w_{ij}(t, \tau_{\text{pre}})$ and $k_{ij}(t, \tau_{\text{next}})$ prevail consistently across different combinations of τ_{pre} and τ_{next} . Furthermore, we also observed the enduring presence of positive correlations between $w_{ij}(t, \tau_{\text{pre}})$ and $k_{ij}(t, \tau_{\text{next}})$ among all birds in three bird-flocking datasets (**Fig.R4h-j**).

To ensure a thorough analysis of the results, we also performed the correlation analysis between w_{ij} and k_{ij} (or ϕ_{ij}), both of which are calculated from the period of $[t - \tau_{\text{pre}}, t]$ (**Fig.R5a**). **Fig.R5b-d** show that $\phi_{ij}(t, \tau_{\text{pre}})$ continues to exhibit a predominance of negative correlation with

$w_{ij}(t, \tau_{\text{pre}})$. It makes sense because significant changes occurring in the relative positions of a bird pair naturally lead to a reduced level of synchronized velocity consensus. However, upon redirecting our attention to $k_{ij}(t, \tau_{\text{pre}})$, unlike the emergence of positive correlations between $w_{ij}(t, \tau_{\text{pre}})$ and $k_{ij}(t, \tau_{\text{next}})$ in a sequential period, we could not observe a prevalence of positive correlations between $w_{ij}(t, \tau_{\text{pre}})$ and $k_{ij}(t, \tau_{\text{pre}})$ (**Fig.R5e-g**). The difference of correlation results derived from $k_{ij}(t, \tau_{\text{next}})$ and $k_{ij}(t, \tau_{\text{pre}})$ respectively indicates the presence of time delay for a bird to adjust its heading after perceiving its neighbors' motion.

We further calculated the correlation between $w_{ij}(t, \tau_{\text{pre}})$ and $k_{ij}(t, \tau_{\text{next}})$ for the swarm trajectories generated by three interactions employed in the collective evacuation experiments. **Fig.R6** shows that only AMS indicates the prevalence of positive correlation between $w_{ij}(t, \tau_{\text{pre}})$ and $k_{ij}(t, \tau_{\text{next}})$, whereas we could not observe this in the experiments using ATHD or average interaction. (Please note that the positive correlation in AMS is not highly significant, as its peak of the correlation distribution locates at $\rho_{w_{ij}-k_{ij}} \approx 0.15$. It is possible that, except the alignment, collective evacuation experiments involve various types of interactions, such as collision avoidance against inter-agent or agent-wall, as well as guidance for agents to navigate towards the exit.)

Based on the above analysis from real bird flocks to robotics experiments, **we have concluded that the prevalence of positive correlations between $w_{ij}(t, \tau_{\text{pre}})$ and $k_{ij}(t, \tau_{\text{next}})$ demonstrates that individuals will accelerate the convergence of velocity with neighbors who have higher MS.** For example, if individual- i perceives neighbor- j with higher MS in the past, it will adjust its heading to align with neighbor- j more rapidly in the future. This empirical finding in real bird flocks indicates two points:

- **It perfectly accords with the meaning of AMS interaction.** In AMS, the coefficient w_{ij} signifies that the impact of neighbor- j 's velocity is directly proportional to M_{ij} . If individual- i observes that neighbor- j has a higher MS compared to other neighbors, individual- i will consider much more influences of neighbor- j to adjust its heading in the next step. Consequently, it results in a better alignment between individual- i and j . In contrast to the commonly used average interaction, AMS causes the homogenous interaction (e.g., $w_{ij} = 1$) to transition into a kind of heterogenous interaction. In the revised manuscript, we have emphasized this point (see **main text: page 11, lines 372-378**):
*“Finally, this empirical finding motivates us to introduce the concept of adaptive MS-based (AMS) interaction in swarm model, such as, $\mathbf{v}_i(t + 1) = \hat{\mathbf{v}}_i(t) + \sum_{j \in \mathcal{S}_i} w_{ij}(t) \hat{\mathbf{v}}_j(t)$ and $w_{ij}(t) = \frac{M_{ij}(t, \tau)}{\sum M_{ij}(t, \tau)}$ (**Fig.5a and Methods**). In AMS, the coefficient w_{ij} signifies that the impact of neighbor- j 's velocity is directly proportional to M_{ij} . If individual- i observes that neighbor- j has a higher MS compared to other neighbors, individual- i will consider much more influences of neighbor- j to adjust its heading in the next step. Consequently, it results in a better alignment between individual- i and j .”*
- **It explains why individuals with higher MS tend to occupy the higher leading tier.** As we know, the LF relation is derived from the maximal value of the curve of $\langle C_{ij} \rangle(\tau) = \langle \hat{\mathbf{v}}_i(t) \cdot \hat{\mathbf{v}}_j(t + \tau) \rangle$. Based on the positive correlations between $w_{ij}(t, \tau_{\text{pre}})$ and $k_{ij}(t, \tau_{\text{next}})$, it could be interpreted that if neighbor- j is perceived to have a higher MS perceived by the focal individual- i (equals to higher $w_{ij}(t, \tau_{\text{pre}})$), the focal one will be quicker to align with neighbor- j 's velocity (equals to higher $k_{ij}(t, \tau_{\text{next}})$). This perhaps leads to a time delay for individual- i to converge with the heading of neighbor- j . Coincidentally, the time delay accords with definition

of an LF relation from neighbor- j to individual- i in this study. In the revised manuscript, we have addressed this point (see **main text: pages 10-11, lines 349-355**):

“This phenomenon may assist us in comprehending why individuals with higher MS tend to occupy the higher leading tier, because the positive correlations could be interpreted that if neighbor- j is perceived to have a higher MS by the focal individual- i (equals to higher $w_{ij}(t, \tau_{pre})$), the focal one will be quicker to align with neighbor- j 's velocity (equals to higher $k_{ij}(t, \tau_{next})$). This perhaps leads to a time delay for individual- i to converge with the heading of neighbor- j . Coincidentally, the time delay accords with definition of an LF relation from neighbor- j to individual- i in this study.”

Overall, the above individual-level empirical findings between the past perception of MS and future trends of motion changes not only elucidate why individuals with higher MS tend to occupy higher leading tier, but also reinforce the rationale behind translating bio-inspired mechanisms revealed from real bird flocks into mathematical models in swarm robotics.

In the revised manuscript, we have incorporated the above results into main text as a separate subsection titled “**Individuals will accelerate convergence of velocity with neighbors who have higher MS**” (see **main text: pages 9-11, lines 308-378**), a new Fig.4 (see **main text: pages 28-29**), and new Supplementary Figs.34-36 (see **SI: pages 59-61**).

Point 1.4: A.2. Are the correlations non-trivial?

One of the main correlational claims of the manuscript is the correlation between LF and MS. In my initial report, I already expressed the concern that the high correlation might be already implied by the definition. I remain not persuaded that the correlation is non-trivial. The MS depends on the velocities and positions of the birds at times t and $t-\tau$ (averaged over columns as explained in line 275). The results show that it correlates with another quantity that depends on the relative velocities of the birds at times t and $t-\tau$ (i.e., the individual leadership whose definition can be inferred from lines 140, 153, and 276 + Methods).

The average MS rewards birds toward which many other birds tend to get closer especially when there is a large angle between the relative positions of the birds at times t and $t-\tau$, which seems conceptually related to the concept of leadership encapsulated in the LF definitions.

Response: We thank Reviewer #1 for this critical comment. Upon a comprehensive analysis from three bird-flocking datasets, we are more convinced of non-trivial correlations between MS and LF, and the significance of MS in understanding the collective behaviors in animal flocks.

We speculated that the reviewer's concern may stem from the fact that in previous manuscript we only presented the observation of positive correlations between LF and MS, without conducting a further analysis to elucidate the individual-level reasons behind this phenomenon. As pointed out by the reviewer, while we could vaguely suggest a connection between LF and MS at a conceptual level, there is still a lack of clear empirical evidence at the individual level regarding how individuals utilize MS to cause the emergence of LF relations.

Therefore, leveraging the new individual-level empirical findings, we aimed to address this concern from two perspectives.

1) The empirical finding regarding the predominance of positive correlations between $w_{ij}(t, \tau_{pre})$ and $k_{ij}(t, \tau_{next})$ elucidates why MS positively correlates with LF at the individual level.

We fully agreed the reviewer that “The average MS rewards birds toward which many other birds tend to get closer especially when there is a large angle between the relative positions of the birds at times t and $t-\tau$, which seems conceptually related to the concept of leadership encapsulated in the LF definitions.” Based on our response to **Point 1.3**, the predominance of positive correlations between $w_{ij}(t, \tau_{pre})$ and $k_{ij}(t, \tau_{next})$ strongly supports this assertion with explicit empirical evidence, thereby reinforces the justification of adaptive MS-based (AMS) interaction in the field of bio-inspired swarm robotics.

2) The role of velocity in MS only reflects the forward-oriented preference of the focal individual.

We apologized for not making this point clear in the previous manuscript. Instead of LF relation being directly derived from velocity, the role of velocity in MS only reflects the forward-oriented preference of the focal individual.

Actually, we emphasize that MS, i.e., $M_{ij}(t, \tau) = \frac{\angle(\hat{x}_{ij}(t), \hat{x}_{ij}(t-\tau))}{\tau} \left(\frac{1+\hat{v}_i(t) \cdot \hat{x}_{ij}(t)}{2} \right)^\alpha \left(\frac{1+\hat{v}_i(t-\tau) \cdot \hat{x}_{ij}(t-\tau)}{2} \right)^\alpha$, primarily depends on the difference of relative position of a bird pair at times t and $t - \tau$. As noted by the reviewer, the part $\frac{\angle(\hat{x}_{ij}(t), \hat{x}_{ij}(t-\tau))}{\tau}$ measures the angle between the **relative positions** of the birds at times t and $t - \tau$. The expression $\left(\frac{1+\hat{v}_i(t) \cdot \hat{x}_{ij}(t)}{2} \right)^\alpha$ or $\left(\frac{1+\hat{v}_i(t-\tau) \cdot \hat{x}_{ij}(t-\tau)}{2} \right)^\alpha$ only reflects the forward-oriented preference of perceiving neighbor- j 's motion from the first-person view of focal individual- i . Here, the first-person view of individual- i aligns with itself heading at time t . For example, the diagram in **Fig.R2b** shows that the value of $\frac{1+\hat{v}_i(t) \cdot \hat{x}_{ij}(t)}{2}$ decreasing from 1 to 0.5 to 0 due to the relative positive between individual- i and j .

Therefore, the last two components of MS equation mean that, for the focal individual- i , its perception capability diminishes as the first-person view transitions from the front to the back. Here, the first-person view of individual- i aligns with itself heading at time t , and α (only taking $\alpha \geq 0$) controls the anisotropic effect of motion perception. If $\alpha = 0$, $M_{ij}(t, \tau)$ ignores the blind area of perception because the last two components in MS equation always equal to 1 regardless of the relative positions of neighbors. Increasing α could make MS equation amplify the anisotropic effect of motion perception, that is, the ability of individual perceiving movements of around neighbors gradually narrows to the front vision.

In the revised manuscript, we have refined the following sentences to enhance clarity (see **main text: page 7, lines 223-227**):

“To mimic the anisotropic effect of motion perception in real birds, the last two components on the right side of the MS equation in Eq.(1) reflects the forward-oriented preference of perceiving neighbor- j 's motion from the first-person view of focal individual- i (**Fig.2b**). It simulates that, for the focal individual- i , its perception capability diminishes as the first-person view transitions from the front to the back. Here the first view of individual- i aligns with itself heading at time t .”

Point 1.5: A.3. Contribution and relevance of the results.

As in the previous version, by reading the introduction, it remains very hard for the reader to infer the actual challenge being solved by the manuscript. Many broad problems are mentioned, but the introduction fails to give a clear sense of what is being solved and why it is significant. A broad spectrum of limitations of existing studies are mentioned — e.g., the use of simplified interaction rules in existing models, the use of phenomenological interactions, lack of integration of the perception stream.

But the only solution offered by the manuscript to this broad array of limitations seems to be the use of the MS, which does not seem to fully solve the mentioned problems. For example, as the MS seems to be a heuristic quantity, and the MS mechanism is not investigated or validated in detail, it seems again that we have a “phenomenological” model (not derived from first principles) similarly to previous work. The model by the authors makes parsimonious assumptions on individual behavior, without the interaction mechanism being detailed validated in the data. Many other issues mentioned in the introduction are, in the end, not fully addressed by the present submission.

Similarly, in the Discussion, the authors define the status quo in the study of collective motions (line 479), but it remains unclear whether the manuscript constitutes a substantial shift in the status quo, given its limitations. For example, in the absence of a clear mechanism behind the observed correlations and experimental results, it is unclear if the obtained results would generalize beyond the analyzed flock datasets and the specific evacuation experiments performed.

In sum, it is hard to conclude that the authors have indeed achieved a “comprehensive research chain from biological observation to bionic mechanisms to bio-inspired robotics”. The manuscript presents interesting ideas and initial insights in three areas, but seems far from achieving a clear link between the three aspects without doing substantially more research on the individual-level mechanisms behind their findings.

Response: We thank Reviewer #1 for these critical comments. We believe our new individual-level analysis on the past perception of MS and future trends of motion changes further strengthens the rationale for AMS with explicit empirical evidence. Thus, we have rewritten the **Abstract** (see main text: page 2, lines 18-41), **Introduction** (see main text: pages 3-4, lines 44-107), and **Discussion** (see main text: pages 14-16, lines 483-555) sections to address our contributions of this study from 3 points:

- 1) **Start point of the modeling framework.** We highlighted that in order to go beyond the most popular averaging-type interaction, it is important to consider the first-person perspective of individuals within the group to further investigate the interaction rules in real flocks. This is natural: at the individual level, the macro-phenomena emerging from group-level, such as hierarchical leading relations in pigeon flock, stem from individuals adjusting their movements based on information perceived from the surrounding environment and the motions of neighboring individuals. Therefore, in this work, we tried to propose a heuristic metric to measure the relative motion changes between an individual pair from the focal one’s first-person view. Then we investigated the correlation between individual perception and leader-follower relations from three bird-flocking datasets, as the emergence of leading relation is intrinsically connected to adjustment of movement upon perceiving neighbors’ motions.
- 2) **Methodology of comprehensive research chain.** We highlighted that this study aims to establish a promising route to integrate the comprehensive research from real flocking data analysis to applications of swarm robotics. Once this promising route is well-established, it

could not only validate the effectiveness of self-organization underlying collective behaviors revealed from real flocks but also provide substantial guidance and improvement on practical design and distributed control strategies for swarm robots. In this study, by leveraging three large bird-flocking datasets, our goal is to establish a comprehensive research chain that begins with studying interactions among neighbors from real bird flocks, translating them into bio-inspired swarm models with explicit empirical evidence, and ultimately applying them in swarm robotics to examine the efficacy of self-organization in collective tasks.

- 3) **Surprise of transferring bio-inspired mechanisms to swarm robotics.** We highlighted the importance of applying bio-inspired mechanisms for swarm robotics. Moving beyond the mathematical modeling and numerical simulations in the field of collective behaviors, the physical swarm robotics, as an arbitrarily customizable test-bed, not only could examine the efficacy of bio-inspired mechanisms in collective tasks but also could extend the potential applications of self-organization for group tasks. For instance, we were highly impressed by the significant advantage of AMS in enhancing the self-organization of the swarm in collective evacuation experiments: the swarm spontaneously emerge three kinds of collective behaviors in a sequence — contracting away from the narrow exit, then facing towards the center of narrow exit, and ultimately evacuating smoothly as a cohesive unit.

Point 1.6: (B) Causal claims not supported by the results.

Causal sentences have been reduced in the manuscript, but some key claims of the manuscript are still causal, e.g., the MS has the “ability to bridge perception and motion”, the AMS “empowers the swarm to promptly respond to the transient perturbation”; in the abstract, “flocks with higher maneuverable motions (i.e., mobbing and circling) tend to exhibit a more nested structure of leader-follower (LF) relations and a clear hierarchy *to mitigate* the damage of individual drive to group cohesion”.

These claims imply that the swarms leverage the MS to achieve certain goals, but there is no evidence that this phenomenon is taking place, as far as I can tell from the results. In the correlational section, for example, the results indicate that swarms with higher/lower MS have different structural characteristics, but this does not fully prove that MS is the causal reason behind the differences. As the authors recognize in the discussion, the ecological contexts and other factors could play a role.

The only exception perhaps is the collective evacuation experiment. However given the specific nature of the setup and lack of detailed mechanisms, it is unlikely that an overly general conclusion on the role of the MS can be inferred from this single experiment.

Response: We thank Reviewer #1 for this critical comment.

As the responses to this concern is closely related to the individual-level empirical findings and wording of the manuscript, we have reported the details in **Point 1.3** and **Point 1.5**, respectively.

- We have incorporated the individual-level empirical findings into main text as a separate subsection titled “**Individuals will accelerate convergence of velocity with neighbors who have higher MS**” (see main text: pages 9-11, lines 308-378), a new Fig.4 (see main text: pages 28-29), and new Supplementary Figs.34-36 (see SI: pages 59-61).

- We have rewritten the **Abstract** (see main text: page 2, lines 18-41), **Introduction** (see main text: pages 3-4, lines 44-107), and **Discussion** (main text: pages 14-16, lines 483-555) sections to avoid the ambiguous assertions without a clear cause.

Besides, we also made the revisions of effect (or advantages) of AMS on simulations and collective evacuation experiments.

- We have included a paragraph and a new Supplementary Figure to further strength that AMS could effectively capture the fundamental characteristics observed in real bird flocks (see main text: page 12, lines 399-410 and **Supplementary Fig.37** in SI: page 62):

“To deeply understand the effect of AMS, we performed the hybrid simulations by introducing AMS to ATHD (or average interaction). For example, Supplementary Fig.37 shows that firstly we set all the individuals follow ATHD (or average interaction), and then we tuned the percent of individual utilizing AMS from 0% to 100% (x -axis of Supplementary Fig.37a,b). Lastly, we compared the nestedness of LF networks and distribution of LF-MS correlation across the different percent of individual using AMS. Obviously, Supplementary Fig.37a,b shows the nestedness of LF networks rises as the percentage of individuals utilizing AMS increased, and Supplementary Fig.37c,d indicates that as more individuals used AMS, the LF-MS correlation across different combinations of t and τ gradually shifted towards a predominance of positive correlation. The results confirm that AMS interaction may play a significant role in shaping the emergence of nested and hierarchical LF relationships in collective motions, as well as reproducing the pattern of predominant positive correlation between MS and LF.”

- We have included a paragraph and an updated Supplementary Figure detailing the swarm evacuation process using AMS, along with an explanation of the advantages of AMS in evacuation experiments (see main text: page 13, lines 438-461 and **Supplementary Fig.43** in SI: page 68):

“Surprisingly, AMS interaction completely surpasses this limitation by spontaneously inducing three distinct collective behaviors in sequence: first, the swarm undergoes a contraction process away from the narrow exit; then, it aligns directly with the center of the narrow exit; and finally, it evacuates smoothly as a cohesive unit (**Fig.6c** and Supplementary Videos 5-7). In particular, we observed that in order to spare space for swarm contraction, some individuals temporarily sacrifice their evacuation time to move towards the opposite direction of the exit (highlighted by red boxes in Supplementary Fig.42). To gain a deeper understanding of self-organization of AMS, we focused on comparing the dot product between desired alignment heading $\hat{\mathbf{v}}_{al,i}$ and real moving heading \mathbf{v}_i . Note that the other four parts of velocity components except $\hat{\mathbf{v}}_{al,i}$ remain consistent across AMS, ATHD, and average interaction in the evacuation experiments (see Supplementary Sec.9.1 for details). After analyzing the evacuation process using AMS, ATHD and average interaction shown in Fig.6a-c, we found that the dot product between $\mathbf{v}_i(t)$ and $\hat{\mathbf{v}}_{al,i}(t)$ in descending order is AMS, ATHD and Average, and AMS has the least variance of dot product (Supplementary Fig.43). It demonstrates that throughout the evacuation process, AMS, compared to other interactions, enables each individual to achieve maximum consistency between aligning with neighbors ($\hat{\mathbf{v}}_{al,i}$) and following the actual direction (\mathbf{v}_i , resulted from various types of interactions with surrounding environments and neighbors) of movement. Given that all individuals are potentially guided towards the exit, if each individual maintains maximum alignment consensus with neighbors and the actual direction of movement, the swarm should be well-organized to efficiently evacuate through the narrow exit. Moreover, we also tested the correlation between $w_{ij}(t, \tau_{pre})$ and $k_{ij}(t, \tau_{next})$ for the swarm trajectories generated by three interactions (Supplementary Fig.43). The results show that only AMS indicates the prevalence of positive correlation between $w_{ij}(t, \tau_{pre})$ and $k_{ij}(t, \tau_{next})$, whereas we could not observe this in the other two interactions.”

- We have removed “collective following experiments” to SI (see **SI Sec.9.2** and **9.3: pages 17-20**), as its significance cannot be adequately compared to the evacuation experiments.

Point 1.7: (C) Nestedness results.

The authors followed my advice to compared raw nestedness metrics with the levels of nestedness observed in randomized networks of increasing level of constraint. But the results seem to show that even under the loosest null models, the nestedness is rarely significant, as the z scores rarely exceed +2 (as a side note, using p-values instead of z-scores would be more suitable in the absence of clear evidence that the nestedness metrics follow a normal distribution in the randomized networks). This lacks of significance seems to suggest that the observed nestedness of the binarized networks simply results from heterogeneity in interaction frequencies, which is not discussed in the manuscript. Perhaps a weighted nestedness metric, such as the spectral radius of the weighted network, might produce significant results and interesting insights.

Response: We thank Reviewer #1 for this constructive comment. Upon carefully examining the z-score calculation in the previous revision, we realized that we actually calculated the z-scores for 100 NODF values of random binary matrices (corresponding to 100 randomizations for each of 4 null models), instead of determining the z-score of real NODF value within the distribution of random NODF values. You could locate the caption of Fig.R2,R3 in the first-round response letter and it wrote as follows: “*For the mobbing dataset, the boxplot for z-scores of average NODF of 4 null models. Each box represents a mobbing flock, and is calculated from the corresponding null model of 100 independent randomizations.*” We apologized for this mistake.

In this revision, we performed the z-score calculation as follows: (1) calculating the NODF of a real LF network; (2) calculating the NODF of 100 random binary matrices generated from one of 4 null modes; (3) calculating the z-score of real NODF as

$$z_score = (\text{real_NODF_value} - \text{mean}(\text{random_NODF_values})) / \text{std}(\text{random_NODF_values}).$$

We have extensively reviewed the tutorial on “statistical tests” in the BiMat package (Flores et al, Methods in Ecology and Evolution, 2016), and utilized the built-in function `StatisticalTest.TEST_NESTEDNESS(matrix,100,@NullModels.EQUIPROBABLE,@NestednessNODF)`; to validate the correction of new z-score of real NODF compared with the null models.

Fig.R7 shows the new z-score of real NODF value within the distribution of random NODF values generated from 4 null models. We found that except Null-4 (both the connectance and the degree distribution of each node are preserved), more nested LF networks (representing higher NODF) exhibit higher z-scores with respect to their randomized counterparts.

In the revised SI, we have rectified the error in SI Sec.4.3 (see **SI: page 10, lines 318-322**) and Supplementary Figs.18,19 (see **SI: pages 43-44**):

“**Moreover, Supplementary Fig.18f-i (or Supplementary Fig.19f-i) show the z-score of real NODF (or real normalized Spectral Radius) value within the distribution of random NODF (or random normalized Spectral Radius) values generated from 4 null models. We found that except Null-4, more nested LF networks (representing higher NODF or normalized Spectral Radius) exhibit higher z-scores with respect to their randomized counterparts.**”

Point 1.8: (D) Readability.

The manuscript remains very dense, with many definitions, abbreviations, and results briefly mentioned but not fully explained. The text is written in a “non linear” way, and the reader has to go back and forth from the main text to methods and back to try to understand the details. Some definitions (e.g., individual leadership) can only be inferred by the reader through combining pieces of information from far-away paragraphs. Besides, the text seems more like an initial draft, it is not polished at various points of the manuscript and reply letter, which sometimes prevents the reader from fully understanding the point the authors wish to convey.

Response: We thank Reviewer #1 for this critical comment.

To avoid the ‘non-linear’ narrative type in the manuscript, and considering MS as the central concept in this study, we have gathered all MS-related definitions and interpretations into a distinct subsection within the Results section, titled “**Perception of motion salience measures the relative motion changes**” (see **main text: pages 6-7, lines 182-239**). Besides, since leadership is a well-established and frequently employed concept in the many modeling papers of collective behaviors, also considering the density of content in the main text, perhaps introducing this concept and interpretation in the Methods section would help conserve space in the main body of the Results section.

In this round of revisions, we have meticulously reviewed and revised certain sentences, rephrased sections (or subsections), as well as incorporated new results in the main text to better articulate our findings and increase the readability of our manuscript. We reported the major revisions following the order of the content as it appears in the manuscript.

- The motivation and contribution of this study (see responses to Point 1.5). We have rewritten the sections of
 - **Abstract** (see **main text: page 2, lines 18-41**),
 - **Introduction** (see **main text: pages 3-4, lines 44-107**),
 - **Discussion** (see **main text: pages 14-16, lines 483-555**).
- The definition and interpretation of MS (see responses to Point 1.1 and Point 1.2)
 - We have thoroughly reorganized all MS-related definition and interpretation into a new subsection titled “**Perception of motion salience measures the relative motion changes**” (see **main text: pages 6-7, lines 182-239**).
 - We have incorporated the new results into SI as a separate subsection titled “**5. Perception of motion salience reflects variations in heading, speed, and acceleration over a period of flock**” (see **SI: pages 10-11, lines 323-360**).
 - We have deleted the contents related to MST (maximal motion salience time), such as, paragraphs in the main text, panels in Fig.2, and some Supplementary Figures. Now, the new Fig.2 in main text only contains the diagram of definition of MS (see **Fig.2 in main text: page 26**).
- The empirical findings between the past perception of MS and future motion trend at the individual-level (see responses to Point 1.3)
 - We have incorporated the individual-level empirical findings into main text as a separate subsection titled “**Individuals will accelerate convergence of velocity with neighbors who have higher MS**” (see **main text: pages 9-11, lines 308-378**), a new Fig.4 (see **main text: pages 28-29**), and new Supplementary Figs.34-36 (see **SI: pages 59-61**).

- The revisions of effect (or advantages) of AMS on simulations and collective evacuation experiments (see responses to Point 1.6)
 - We have included a paragraph and a new Supplementary Figure to further strength that AMS could effectively capture the fundamental characteristics observed in real bird flocks (see **main text: page 12, lines 399-410** and **Supplementary Fig.37 in SI: page 62**).
 - We have included a paragraph and an updated Supplementary Figure detailing the swarm evacuation process using AMS, along with an explanation of the advantages of AMS in evacuation experiments (see **main text: page 13, lines 438-461** and **Supplementary Fig.43 in SI: page 68**).
 - We have removed “collective following experiments” to SI (see **SI Sec.9.2** and **9.3: pages 17-20**), as its significance cannot be adequately compared to the evacuation experiments.

Finally, we thank Reviewer #1 again for her/his very insightful and constructive comments/suggestions. We hope our responses above have addressed those very legitimate issues/concerns in a satisfactory manner.

Response to Reviewer #2

The authors did a great job revising the manuscript, I find the new version highly improved in context and clarity and all my concerns have been tackled. I would thus happily recommend its acceptance for publication.

Response: We thank Reviewer #2 very much for reviewing our paper again. We are very pleased to know that s/he is now happy with the revised version.

Response to Reviewer #3

The authors have addressed my comments in the latest revision of the manuscript titled "Perception of Motion Saliency Shapes the Emergence of Collective Motions". I believe that these changes have improved the quality of the manuscript.

Response: We thank Reviewer #3 very much for reviewing our paper again. We are very pleased to know that s/he is now happy with the revised version.

Response Figure

Fig.R1 | The correlation analysis between MS and other two metrics quantifying the motion differences by velocity and speed. **a**, For a period $[t - \tau, t]$ highlighted by red, we performed the correlation analysis for all pairs of birds between MS and other two metrics: (i) average velocity consensus (AVC) of each pair of birds, denoted as $\phi_{ij}(t, \tau) = \langle \mathbf{v}_i(t) \cdot \mathbf{v}_j(t) \rangle_\tau$; (ii) distance of temporal speed (DS) of each pair of birds, denoted as $\mathcal{D}_{ij}(t, \tau) = \text{Distance}(\text{vec}(v_i(t)), \text{vec}(v_j(t)))$. For simplicity, the DS value $\mathcal{D}_{ij}(t, \tau)$ is calculated by the Euclidean distance of two vectors composed of v_i and v_j within the period $[t - \tau, t]$. From the flock shown in panel a, the heatmap of Spearman correlation of MS-AVC (**b**) and MS-DS (**c**) under various combinations of t and τ . The distribution of Spearman correlation of MS-AVC (**d-f**) and MS-DS (**g-i**) from three bird-flocking datasets. In panels b-i, for each flock, we took $\alpha = 0$ to calculate MS, 21 time stamps for t , and 22 time points for τ .

Fig.R2 | Perception of motion salience in collective motions. **a**, Diagram of MS to quantify the relative movement changes of neighbor- j from the focal individual- i 's first-person view. **b**, The anisotropic effect of motion perception in Eq.(1) mimics the idea that the perception capability diminishes as the focal individual's first-person sight extends from the front to the back. **c**, The anisotropic factor $\alpha \geq 0$ in MS controls the fact of forward-oriented preference in biological perception. The x -axis indicates the heading between two vectors $\hat{\mathbf{v}}_i$ and $\hat{\mathbf{x}}_{ij}$, and y -axis corresponds to the second or third term of right side of Eq.(1). If $\alpha = 0$, we ignored the blind area of motion perception. With increasing $\alpha > 0$, we assumed that the ability of individual perceiving movements of around neighbors gradually narrows to the front vision.

Fig.R3 | The correlation analysis between MS and average of temporal velocity consensus at the individual level. We used the flock within the period $[t - \tau_{\text{pre}}, t]$ (highlighted by blue in **a**) and the period $[t, t + \tau_{\text{next}}]$ (highlighted by red in **a**) to generate $M_{ij}(t, \tau_{\text{pre}})$ (and the normalization $w_{ij}(t, \tau_{\text{pre}}) = \frac{M_{ij}(t, \tau)}{\sum M_{ij}(t, \tau)}$) (**b**) and the average of temporal velocity consensus $\phi_{ij}(t, \tau_{\text{next}}) = \langle \mathbf{v}_1(t) \cdot \mathbf{v}_j(t) \rangle$ (**c**). **d**, Then for individual-1, we could perform the correlation analysis between two vectors composed of $w_{1j}(t, \tau_{\text{pre}})$ and $\phi_{1j}(t, \tau_{\text{next}})$ respectively. In panels **e-g**, across all individuals present in the flock, negative correlations between $w_{ij}(t, \tau_{\text{pre}})$ and $\phi_{ij}(t, \tau_{\text{next}})$ are consistently prevalent for different combinations of τ_{pre} and τ_{next} . Note that one step corresponds to 1/60 second in mobbing and transit datasets, and 1/30 second in circling dataset. In panels **h-j**, we also noticed the persistence of negative correlations between $w_{ij}(t, \tau_{\text{pre}})$ and $\phi_{ij}(t, \tau_{\text{next}})$ across all birds in three bird-flocking datasets. In this figure we took $\alpha = 0$ to calculate MS.

Fig.R4 | The correlation analysis between MS and average slope of temporal velocity consensus at the individual level. We used the flock within the period $[t - \tau_{\text{pre}}, t]$ (highlighted by blue in **a**) to generate $M_{ij}(t, \tau_{\text{pre}})$ (and the normalization $w_{ij}(t, \tau_{\text{pre}}) = \frac{M_{ij}(t, \tau)}{\sum M_{ij}(t, \tau)}$) (**b**). Based on the curve of temporal velocity consensus $\phi_{ij}(t, \tau_{\text{next}}) = \langle \mathbf{v}_i(t) \cdot \mathbf{v}_j(t) \rangle$ (inset in panel **c**), we could calculate the average slope of $\phi_{ij}(t, \tau_{\text{next}})$ within the future period $[t, t + \tau_{\text{next}}]$, denoted as $k_{ij}(t, \tau_{\text{next}}) = \langle \text{slope of } \mathbf{v}_i(t) \cdot \mathbf{v}_j(t) \rangle$ (**c**). A larger value of $k_{ij}(t, \tau_{\text{next}})$ signifies that two individuals are aiming to align with each other more rapidly in the future. **d**, Then for individual-1, we could perform the correlation analysis between two vectors composed of $w_{1j}(t, \tau_{\text{pre}})$ and $k_{1j}(t, \tau_{\text{next}})$ respectively. In panels **e-g**, across all individuals present in the flock, positive correlations between $w_{ij}(t, \tau_{\text{pre}})$ and $k_{ij}(t, \tau_{\text{next}})$ prevail consistently for different combinations of τ_{pre} and τ_{next} . Note that one step corresponds to 1/60 second in mobbing and transit datasets, and 1/30 second in circling dataset. In panels **h-j**, we also observed the enduring presence of positive correlations between $w_{ij}(t, \tau_{\text{pre}})$ and $k_{ij}(t, \tau_{\text{next}})$ among all birds in three bird-flocking datasets. In this figure we took $\alpha = 0$ to calculate MS.

Fig.R5 | The correlation analysis between two vectors composed of M_{ij} and ϕ_{ij} (or k_{ij}) from the same period $[t - \tau_{pre}, t]$. **a**, Different with Fig.R2 (or Fig.R3), both w_{ij} and k_{ij} are derived from the same period of $[t - \tau_{pre}, t]$. In three bird-flocking datasets, $\phi_{ij}(t, \tau_{pre})$ continues to exhibit a predominance of negative correlation with $w_{ij}(t, \tau_{pre})$ during the same period (**b-d**). However, unlike the emergence of positive correlations between $w_{ij}(t, \tau_{pre})$ and $k_{ij}(t, \tau_{next})$ in a sequential period, we could not observe a prevalence of positive correlations between $w_{ij}(t, \tau_{pre})$ and $k_{ij}(t, \tau_{pre})$ (**e-g**). In this figure we took $\alpha = 0$ to calculate MS.

Fig.R6 | The correlation analysis between $w_{ij}(t, \tau_{pre})$ and $k_{ij}(t, \tau_{next})$ for the swarm trajectories generated by three interactions employed in the collective evacuation experiments. For the given interaction (e.g., AMS, ATHD or average) and exit width, we have 10 independent swarm trajectories to perform this correlation analysis.

Fig.R7 | The z-score of real nestedness value within the distribution of random nestedness values generated from 4 null models. a, For the mobbing dataset, the z-score of real NODF value within the distribution of random NODF values generated from 4 null models respectively. Each point represents a mobbing flock, the x -axis is real NODF value calculated from the LF matrix of this flock, and the y -axis is z-score of this real NODF value within 100 random NODF values. **b,** For the mobbing dataset, the z-score of real n-Spectral Radius value within the distribution of random n-Spectral Radius values generated from 4 null models respectively. Each point represents a mobbing flock, the x -axis is real n-Spectral Radius value calculated from the LF matrix of this flock, and the y -axis is z-score of this real n-Spectral Radius value within 100 random n-Spectral Radius values.

REVIEWERS' COMMENTS

Reviewer #1 (Remarks to the Author):

The manuscript has further improved. The new results bring the manuscript closer to a publishable version. In particular, the authors clarified that the MS metric is heuristic variable, they corrected the nestedness results, and they moved an initial step toward understanding the individual-level mechanism behind their correlational results. Besides, they characterized more in depth the dynamics in their laboratory experiments.

There are still a few areas that need attention and improvements, which I list below in order of importance.

(1) Readability and wording.

The introduction has improved. The motivation of the work looks now clearer. I feel it could be further improved, by first presenting all what is not there in the literature, and then concluding with the contributions of this paper. Currently, the introduction goes back and forth between the two, which makes it confusing to read. I would suggest also highlighting more the laboratory validation, as it seems to be a strong contribution of this work.

I feel the abstract too can improved. The revised version is quite long, and I believe that the results can be presented more concisely.

When examining individual sentences throughout the text, it seems that significant improvements are needed in the write-up.

– There are many grammatically incorrect sentences, even in the abstract, which indicates that a major proofreading is needed.

– I would suggest avoiding or reducing substantially the use of strong words like “surprisingly, obviously, completely, insufficient, deficiency, naturally“, which are currently used extensively, often distracting the reader and leading to confusion. For example, in the experimental section, the authors write that “surprisingly, the AMS interaction completely surpasses this limitation ...”. But at

this stage of the manuscript, shouldn't it be expected in light of the previous results? Or if it is surprising, it is just a lucky result without theoretical understanding? "Surprisingly" also appears in the discussion section.

Overall, I strongly recommend that the authors pay much more attention to word usage. Perhaps seeking advice from colleagues can be helpful to polish the text and ensure clarity.

(2) Precision in formulas and notations.

– One could perhaps define a perception variable $P_{ij}(t) = (1 + v_i \cdot x_{ij})/2$, where dot is the scalar product. And use it in the MS definition, eq. 1, and novel Fig. 2.

– The equation of w_{ij} in line 317 seems incorrect or incomplete. Is the sum over j ? Should τ be τ_{pre} ?

– If my understanding is correct, w_{ij} is simply a normalized version of the MS. It would be ideal to not introduce a different symbol; one could use m_{ij} instead of w_{ij} , where m is a normalized version of M .

– Equation of k_{ij} incomplete in line 334, how is "slope" defined? Please add the precise definition.

Response to Reviewer #1

Point 1.0: The manuscript has further improved. The new results bring the manuscript closer to a publishable version. In particular, the authors clarified that the MS metric is heuristic variable, they corrected the nestedness results, and they moved an initial step toward understanding the individual-level mechanism behind their correlational results. Besides, they characterized more in depth the dynamics in their laboratory experiments.

There are still a few areas that need attention and improvements, which I list below in order of importance.

Response: We thank Reviewer #1 very much for reviewing our paper again. Following her/his constructive comments and suggestions, we have carefully addressed each concern to improve the quality of our work.

(1) Readability and wording.

Point 1.1: The introduction has improved. The motivation of the work looks now clearer. I feel it could be further improved, by first presenting all what is not there in the literature, and then concluding with the contributions of this paper. Currently, the introduction goes back and forth between the two, which makes it confusing to read. I would suggest also highlighting more the laboratory validation, as it seems to be a strong contribution of this work.

Response: We thank Reviewer #1 very much for carefully reviewing our manuscript. Followed this valuable suggestion, we have restructured the Introduction section and rewritten a part of paragraphs (see **main text: page 3-4, lines 33-113**).

Point 1.2: I feel the abstract too can improved. The revised version is quite long, and I believe that the results can be presented more concisely.

When examining individual sentences throughout the text, it seems that significant improvements are needed in the write-up.

– There are many grammatically incorrect sentences, even in the abstract, which indicates that a major proofreading is needed.

Response: We thank reviewer's critical suggestion. Here we have shortened the abstract and presented the results more concisely in about 160 words (see **main text: page 2, lines 20-31**):

“Despite the profound implications of self-organization in animal groups for collective behaviors, understanding the fundamental principles and applying them to swarm robotics remains incomplete. Here we propose a heuristic measure of perception of motion salience (MS) to quantify relative motion changes of neighbors from first-person view. Leveraging three large bird-flocking datasets, we explore how this perception of MS relates to the structure of leader-follower (LF) relations, and further perform an individual-level correlation analysis between past perception of MS and future change rate of velocity consensus. We observe prevalence of the positive correlations in real flocks, which demonstrates that individuals will accelerate the convergence of velocity with neighbors who have higher MS. This empirical finding motivates us to introduce the concept of adaptive MS-based (AMS) interaction in swarm model. Finally, we implement AMS in a swarm of $\sim 10^2$ miniature

robots. Swarm experiments show the significant advantage of AMS in enhancing self-organization of the swarm for smooth evacuations from confined environments.”

Point 1.3: – I would suggest avoiding or reducing substantially the use of strong words like “surprisingly, obviously, completely, insufficient, deficiency, naturally”, which are currently used extensively, often distracting the reader and leading to confusion. For example, in the experimental section, the authors write that “surprisingly, the AMS interaction completely surpasses this limitation ...”. But at this stage of the manuscript, shouldn’t it be expected in light of the previous results? Or if it is surprising, it is just a lucky result without theoretical understanding? “Surprisingly” also appears in the discussion section.

Response: We thank Reviewer #1 for this critical comment. We have carefully reviewed the entire text and eliminated (or rephrased) the strong adjective or adverbs that convey subjective feelings.

Point 1.4: Overall, I strongly recommend that the authors pay much more attention to word usage. Perhaps seeking advice from colleagues can be helpful to polish the text and ensure clarity.

Response: We thank Reviewer #1 for this very valuable suggestion. During this round of revisions, we meticulously examined and refined the wording and expressions used throughout the manuscript. Following reviewer’s suggestion, our focus is on ensuring clarity, precision, and coherence in conveying the intended concepts. Please review the Abstract, Introduction, Discussion and Results sections in the new version.

Point 1.5: (2) Precision in formulas and notations.

– One could perhaps define a perception variable $P_{ij}(t) = (1 + \hat{v}_i \cdot \hat{x}_{ij})/2$, where dot is the scalar product. And use it in the MS definition, eq. 1, and novel Fig. 2.

Response: We thank Reviewer #1’s suggestion. We think that the MS equation of Eq.(1) clearly illustrates the three components required for calculating MS, and it may be unnecessary to introduce a new symbol $P_{ij}(t)$ to demonstrate the MS definition more effectively. Besides, $(1 + \hat{v}_i(t) \cdot \hat{x}_{ij}(t))/2$ only appears in the definition of MS. Considering that this work already incorporates numerous symbols, we think that introducing a new symbol $P_{ij}(t) = (1 + \hat{v}_i(t) \cdot \hat{x}_{ij}(t))/2$ to define a perception variable may be unnecessary.

– The equation of w_{ij} in line 317 seems incorrect or incomplete. Is the sum over j ? Should τ_{pre} be τ_{pre} ?

Response: We thank Reviewer #1 for pointing this out. Yes, the correct equation is

$$w_{1j}(t, \tau_{pre}) = \frac{M_{1j}(t, \tau_{pre})}{\sum_{j \in S_1} M_{1j}(t, \tau_{pre})}$$
. We have corrected this in the new version. It is sum over neighbor-

j of the focal individual-1, because $M_{1j}(t, \tau_{pre})$ means the neighbors’ MS which is perceived by the focal individual-1.

– If my understanding is correct, w_{ij} is simply a normalized version of the MS. It would be ideal to not introduce a different symbol; one could use m_{ij} instead of w_{ij} , where m is a normalized version of M .

Response: We thank Reviewer #1’s suggestion. Yes, w_{ij} represents a normalized version of the MS. We consider that the weighted coefficient is always denoted as w_{ij} in swarm model, such as $\mathbf{v}_i(t + 1) = \hat{\mathbf{v}}_i(t) + \sum_{j \in \mathcal{S}_i} w_{ij}(t) \hat{\mathbf{v}}_j(t)$. In the manuscript, w_{ij} is consistently used to represent the weighted coefficient or normalization from real data analysis to swarm model to swarm experiments. Besides, we clearly state that w_{ij} is the normalization of MS (see **main text: page 10, lines 314-315**):

“For the mobbing flock shown in Fig.4a, Fig.4b shows the perception of neighbors’ $M_{1j}(t, \tau_{pre})$ (and the normalization $w_{1j}(t, \tau_{pre}) = \frac{M_{1j}(t, \tau_{pre})}{\sum_{j \in \mathcal{S}_1} M_{1j}(t, \tau_{pre})}$) from the focal bird-1 within the past period $[t - \tau_{pre}, t]$ ”

Moreover, considering that this work already incorporates many symbols, we think it may be unnecessary to introduce a new symbol, m_{ij} , to represent the normalization of MS.

– Equation of k_{ij} incomplete in line 334, how is “slope” defined? Please add the precise definition.

Response: We thank Reviewer #1 for pointing this out. We have revised this point in the main text (see **main text: page 10, lines 331-333**):

“Another method is calculating the average slope of temporal velocity consensus within the future period, which is denoted as $k_{ij}(t, \tau_{next}) = \langle \text{slope of the curve } \mathbf{v}_i(t) \cdot \mathbf{v}_j(t) \text{ at time } t \rangle$ (Fig.4c) and the slope at time t is calculated as $\mathbf{v}_i(t) \cdot \mathbf{v}_j(t) - \mathbf{v}_i(t - 1) \cdot \mathbf{v}_j(t - 1)$.”